# On the Dynamics & Transferability of Latent Generalization during Memorization

**Simran Ketha**                                                                    *p20200021@hyderabad.bits-pilani.ac.in*
*Anuradha & Prashanth Palakurthi Centre for Artificial Intelligence Research,*
*Department of Computer Science & Information Systems,*
*Birla Institute of Technology & Science Pilani, Hyderabad 500078, India.*

**Venkatakrishnan Ramaswamy**                                                       *venkat@hyderabad.bits-pilani.ac.in*
*Anuradha & Prashanth Palakurthi Centre for Artificial Intelligence Research,*
*Department of Computer Science & Information Systems,*
*Birla Institute of Technology & Science Pilani, Hyderabad 500078, India.*

**Reviewed on OpenReview:** *https://openreview.net/forum?id=tO24ZmOtKF*

## Abstract

Deep networks have been known to have extraordinary generalization abilities, via mechanisms that aren't yet well understood. It is also known that upon shuffling labels in the training data to varying degrees, deep networks, trained with standard methods, can still achieve perfect or high accuracy on this corrupted training data. This phenomenon is called *memorization*, and typically comes at the cost of poorer generalization to true labels. Our recent work has demonstrated, surprisingly, that the internal representations of such models retain significantly better latent generalization abilities than is directly apparent from the model. In particular, it has been shown that such latent generalization can be recovered via simple probes (called MASC probes) on the layer-wise representations of the model. However, the origin and dynamics over training of this latent generalization during memorization is not well understood. Here, we track the training dynamics, empirically, and find that latent generalization abilities largely peak early in training, with model generalization. Next, we investigate to what extent the specific nature of the MASC probe is critical for our ability to extract latent generalization from the model's layerwise outputs. To this end, we first examine the mathematical structure of the MASC probe and show that it is a quadratic classifier, i.e. is non-linear. This brings up the question of the extent to which this latent generalization might be linearly decodable from layerwise outputs. To investigate this, we designed a new linear probe for this setting. We find cases where this linear probe outperforms MASC, and other cases, where the opposite happens, notably for many instances of ResNet-18 trained on CIFAR-10. Next, we consider the question of whether it is possible to transfer latent generalization to model generalization by directly editing model weights. To this end, we devise a way to transfer the latent generalization present in last-layer representations to the model using the new linear probe. This immediately endows such models with improved generalization in many cases, i.e. without additional training. We also explore training dynamics, when the aforementioned weight editing is done midway during training. Our findings provide a more detailed account of the rich dynamics of latent generalization during memorization, provide clarifying explication on the specific role of the probe in latent generalization, as well as demonstrate the means to leverage this understanding to directly transfer this generalization to the model. Our code is available at: **https://github.com/simranketha/Dynamics_during_training_DNN**.

# 1  Introduction

Overparameterized deep neural networks have seen widespread deployment in many fields, due to their remarkable generalization abilities. However, we still don't have a clear understanding of the mechanisms underlying their ability to generalize so well to unseen data. It has also been shown (Zhang et al., 2017; 2021) that overparameterized deep networks are capable of achieving high or even perfect training accuracy on datasets, wherein a subset of training data have their labels randomly shuffled. Such models typically have poor generalization performance, i.e. poorer accuracy on test data with correct labels – a phenomenon that has been called *memorization*. It is known (Arpit et al., 2017) that during training, models trained with such corrupted datasets exhibit better generalization during the initial phases of training; however generalization progressively deteriorates as training accuracy improves subsequently.

Label noise is also a practical issue in Deep Learning. The success of Deep Learning has depended on large, carefully annotated datasets, which are costly to obtain. To reduce labeling effort, non-expert sources such as Amazon Mechanical Turk and web-derived annotations are often used, but they frequently introduce unreliable labels (Paolacci et al., 2010; Cothey, 2004; Mason & Suri, 2012; Scott et al., 2013). Labeling can also be difficult even for domain experts (Frénay & Verleysen, 2013; RV, 2004), and may be intentionally corrupted through adversarial attacks (Xiao et al., 2012). Such corrupted annotations, referred to as noisy labels, have been estimated to constitute 8.0%–38.5% of real-world datasets (Xiao et al., 2015; Li et al., 2017; Lee et al., 2018; Song et al., 2019) and are known to degrade performance more severely than other noise types, such as input noise (Zhu & Wu, 2004). Consequently, a better understanding of learning under label noise and the representations that drive it, not only has foundational relevance, but could also have downstream implications for our ability to design more effective techniques for learning in the presence of label noise.

Our recent study (Ketha & Ramaswamy, 2026) has shown that while deep networks trained on datasets having corrupted labels tend to exhibit poor generalization, their intermediate-layer representations retain a surprising degree of latent generalization ability. This ability can be recovered from such trained networks by using a simple probe – the Minimum Angle Subspace Classifier (MASC) – that utilizes the subspace geometry of the corrupted training dataset representations, to this end. Our findings suggest that generalizable features are present in the layer-wise representations of such networks, even when the model fails to utilize them sufficiently. However, the origin and evolution of this latent generalization ability during training are not well understood. In particular, it is unclear whether model generalization and latent generalization arise from the same underlying mechanisms, and whether the specific nature of the probe (MASC) is critical for extracting latent generalization from layer-wise representations. Moreover, it is not even clear whether this latent generalization can be directly transferred to the model. That is, whether it is, in principle, possible to directly modify model weights, so it overtly acquires this latent generalization performance, is not known. On the one hand, this question is conceptually important because it establishes conclusively that the latent generalization manifested by the probe is also within reach of the model, with exactly the information that the model was provided during training, namely the corrupted training data. On the other hand – and more pragmatically – it also suggests the possibility of "repairing" a trained model that has memorized, without requiring expensive retraining from scratch. Here, we address these questions.

Our main contributions are listed below.

- We empirically characterize the evolution of latent generalization[1], as manifested by MASC, over epochs of training, in comparison to model generalization over training.

- We observe that MASC is a non-linear classifier. This brings up the question of the extent to which the improved generalization performance of the probe (i.e. MASC) is attributable to the effectiveness of the non-linear nature of the probe itself that may not easily be linearly decodable. To address this point, we introduce a simple linear alternative – Vector Linear Probe Intermediate-layer Classifier (VeLPIC). On the one hand, we find cases where VeLPIC achieves superior latent generalization performance in comparison to MASC, with this happening especially for higher degrees of label

---

[1]We provide precise operational definitions of latent generalization in Subsection 3.1

corruption. On the other hand, there are cases, e.g. with ResNet-18[2], where the best-performing layers with MASC outperform those of VeLPIC.

- As a baseline, we analyze latent generalization during training using a linear probe (logistic regression) trained by minimizing a crossentropy loss & compare its performance with MASC & VeLPIC.

- By leveraging the linear probe (VeLPIC), we devise a way to directly edit the pre-softmax weights of such deep networks, that immediately transfers to the model, the latent generalization performance of VeLPIC (as applied to the last layer).

- We study the dynamics of model generalization and linear probe behavior (VeLPIC) during training by introducing a targeted weight intervention at a specific training epoch.

## 2 Related work

In influential work, (Zhang et al., 2017; 2021) showed that deep networks can achieve perfect training accuracy even with randomly shuffled labels, accompanied by poor generalization. In follow-up work, (Arpit et al., 2017) find that in the memorization regime, networks learn simple patterns first during training. Their work provides a detailed account of the early dynamics of training. More recently, our work (Ketha & Ramaswamy, 2026) in fact show that in spite of the fall in model generalization later on in training, the layerwise representations of the model retain significant latent generalization ability. (Ketha et al.) used probes from (Ketha & Ramaswamy, 2026) to investigate the role of internal representations in adversarial setting.

Analyzing intermediate representations in deep networks has been previously explored using kernel-PCA (Montavon et al., 2011) and linear classifier probes (Alain & Bengio, 2018). Notably, (Alain & Bengio, 2018) state that they deliberately did not probe deep networks in the memorization setting since they thought that such probes would inevitably overfit. On the contrary, (Ketha & Ramaswamy, 2026) demonstrate that probes on deep networks in the memorization setting, can have enhanced generalization. (Stephenson et al., 2021) show evidence suggesting that memorization occurs in the later layers. Li et al. (2020) show that in the memorization regime, gradient descent with early stopping is provably robust to label noise and that there is substantial deviation from initial weights after the early stopping point, which drives overfitting.

Several training paradigms have been proposed to enhance generalization performance when learning from corrupted datasets. For example, MentorNet (Jiang et al., 2018) introduces a framework wherein a mentor network guides the learning process of a student network by guiding the student model to focus on likely clean labels. Likewise, Co-Teaching (Han et al., 2018) trained two peer networks simultaneously, each selecting small-loss examples to update its counterpart. Early-Learning Regularization (ELR) (Liu et al., 2020) augmented the training objective with a regularization term, which implicitly prevents learning of incorrect labels.

Gradient descent on separable data has been shown to exhibit an implicit bias toward specific solutions, even in the absence of explicit regularization, helping to explain generalization in overparameterized models (Soudry et al., 2018). This view was further refined by characterizing implicit bias through optimization geometry, linking solution selection to the structure of the loss landscape (Gunasekar et al., 2018).

Saxe et al. (2013) offer theoretical explanations on generalization for deep linear networks and Lampinen & Ganguli (2018) offer theoretical explanations in the memorization regime. Methodologies such as Canonical Correlation Analysis (Raghu et al., 2017; Morcos et al., 2018) and Centered Kernel Alignment (Kornblith et al., 2019) have been used to characterize training dynamics and network similarity. Representational geometry and structural metrics provide further insights into learned representation properties (Chung et al., 2016; Cohen et al., 2020; Sussillo & Abbott, 2009; Farrell et al., 2019; Bakry et al., 2015; Cayco-Gajic & Silver, 2019; Yosinski et al., 2014). The Neural Tangent Kernel (NTK) framework provides a kernel-based

---

[2]ResNet-18 is the most modern network tested here and, in particular, for the best performing layers, MASC has significantly better accuracy than VeLPIC. Furthermore, there are multiple early layers where MASC on that layer outperforms VeLPIC on the same layer for many degrees of label corruption. However, even with ResNet-18, we find that, for later layers, VeLPIC outperforms MASC later in training, when there is high label noise.

perspective on learning dynamics in infinitely wide networks, connecting neural network training to kernel regression and offering insights into generalization(Jacot et al., 2018).

Recent advances have highlighted limitations of classical generalization theory. In particular, the double descent phenomenon challenges the traditional bias–variance trade-off by showing that model test risk can decrease even after achieving zero training error as model capacity grows, thereby extending the classical U-shaped risk curve (Belkin et al., 2019). This insight has important implications for understanding generalization in high-capacity models that are often trained on large, imperfect datasets. In parallel, robust learning in the presence of label noise has emerged as a critical research direction, since mislabeled data can significantly impair generalization performance and is pervasive in real-world applications. The survey in (Song et al., 2022) systematically reviews existing approaches for learning under noisy labels and highlights the remaining challenges in achieving robustness to label corruption.

## 3 Preliminaries & experimental setup

### 3.1 Preliminaries

We study a $C$-class classification problem defined over an unknown data distribution $\mathcal{D}$ on $\mathcal{X} \times \mathcal{Y}$, where $\mathcal{X} \in \mathbb{R}^n$ and $\mathcal{Y} \in \{1, \dots, C\}$.

Let $f_\theta : \mathcal{X} \to \mathbb{R}^C$ denote a model, i.e. a neural network. The corresponding classifier is defined as

$$h_\theta(x) = \arg\max_{c \in \mathcal{Y}} f_\theta(x)_c$$

We assume that the network admits a decomposition

$$f_\theta = g_\theta \circ \phi_\theta$$

where $\phi_\theta : \mathcal{X} \to \mathbb{R}^d$ denotes the map from the input to the latent representation at a chosen hidden layer, and $g_\theta : \mathbb{R}^d \to \mathbb{R}^C$ denotes the mapping performed by the remainder of the network.

To study generalization properties inherent to the representation from $\phi_\theta$, we consider a probe $f^{\mathrm{lat}} : \mathbb{R}^d \to \mathcal{Y}$ that operates solely on latent representations and the corresponding latent classifier is

$$h^{\mathrm{lat}}(x) = \arg\max_{c \in \mathcal{Y}} f^{\mathrm{lat}}(x)_c$$

In practice, let $T_{\mathrm{test}} = \{(x_i, y_i)\}_{i=1}^m$ denote a test dataset of $m$ samples, where $y_i \in \mathcal{Y}$ are the corresponding true labels. For each test input $x_i \in \mathcal{X}$, we define its latent representation as

$$z_i = \phi_\theta(x_i) \in \mathbb{R}^d.$$

The empirical latent generalization of the chosen hidden layer is the test accuracy of $h^{\mathrm{lat}}$ evaluated on the latent representations $z_i$ with respect to their true labels $y_i$.

$$\widehat{\mathcal{A}}_{\mathrm{lat}}(\theta) = \frac{1}{m} \sum_{i=1}^m \mathbf{1}\{h^{\mathrm{lat}}(z_i) = y_i\}.$$

The empirical model generalization is the test accuracy of $f_\theta$ evaluated on $x_i$ with respect to their true labels $y_i$.

$$\widehat{\mathcal{A}}(\theta) = \frac{1}{m} \sum_{i=1}^m \mathbf{1}\{h_\theta(x_i) = y_i\}.$$

Let $T = \{(x_i, y_i)\}_{i=1}^{t} \sim \mathcal{D}^t$ be an i.i.d. training dataset drawn from $\mathcal{D}$. For a corruption degree[3] $p$, we construct a modified training dataset $\widehat{T}_p$ by changing the labels uniformly at random with probability $p$. $\widehat{T}_p$ is used to train $f_\theta$; we call such models memorized models[4]. The corresponding $z_i$'s and labels are used to construct probes on the layer in question. In this paper, we study the latent generalization of specific classes of probes during model training, and examine their empirical latent generalization relative to the empirical model generalization of the models at the corresponding epoch of model training.

## 3.2 Experimental setup

We demonstrate results for the same set of models and datasets as presented in (Ketha & Ramaswamy, 2026). Specifically, we use Multi-Layer Perceptrons (MLPs) trained on the MNIST (Deng, 2012) and CIFAR-10 (Krizhevsky, 2009) datasets; Convolutional Neural Networks (CNNs) trained on MNIST, Fashion-MNIST (Xiao et al., 2017), and CIFAR-10; AlexNet (Krizhevsky et al., 2012) trained on Tiny ImageNet dataset (Moustafa, 2017) and ResNet-18 (He et al., 2016) trained on CIFAR-10. Additional details on the model architectures and training are available in Section A.

Each model was trained under two distinct schemes: (i) using training data with true labels, referred to as "generalized models," and (ii) using training data with labels randomly shuffled to varying degrees (referred to as "memorized models" Zhang et al. (2021). Similar to Ketha & Ramaswamy (2026), we train the aforementioned models using corruption degrees of 0%, 20%, 40%, 60%, 80%, and 100%. Training with a *corruption degree c* implies that, with probability $c$, the label of a training datapoint is changed with a randomly selected label drawn uniformly from the set of possible classes. This may result in the label remaining the same after the change as well. All models were trained either until achieving high training accuracy (99% or 100%) or for a maximum of 500 epochs, whichever occurred first.

To study the dynamics of the training process, we conducted the experiments on model checkpoints saved at various stages of training. Specifically, we began with the randomly initialized model (corresponding to epoch 0), followed by checkpoints saved at every second epoch up to the 20th epoch. Beyond epoch 20, results are shown at intervals of five epochs for the MLP, CNN and ResNet-18 models, and at intervals of ten epochs for the AlexNet model. The reported results are averaged over three independent training runs, with shaded regions in the plots indicating the range across instances.

Ketha & Ramaswamy (2026) investigate the organization of class-conditional subspaces using the training data at various layers of deep networks. These subspaces are estimated via Principal Components Analysis (PCA), specifically, ensuring that they pass through the origin. To probe the layerwise geometry without relying on subsequent layers, we propose a new probe – the Minimum Angle Subspace Classifier (MASC). For a given test input, MASC projects the layer output onto each class-specific subspace, and computes the angles between the original and projected vectors, for each subspace. The label predicted by MASC corresponds to the class whose subspace yields the projected vector with the smallest such angle. We provide a detailed summary of the working of MASC in Section B. We have used 99% as the percentage of variance explained by the principal components that form the class-specific subspaces used by MASC, similar to experiments conducted in Ketha & Ramaswamy (2026).

## 4 Training dynamics of latent generalization using MASC

As shown in Ketha & Ramaswamy (2026), for most models trained with corrupted labels, there exists at least one layer where MASC exhibits better generalization than the corresponding trained model. However, the origin & evolution of this latent generalization across training isn't well understood.

Here, we empirically study the behavior of latent generalization, as manifested by MASC, during training. MASC testing accuracy during training for MLP trained on MNIST, MLP trained on CIFAR-10, CNN trained on MNIST, CNN trained on Fashion-MNIST, CNN trained on CIFAR-10, AlexNet trained on Tiny

---

[3]Please see Section 3.2 (2nd paragraph) for the definition of *corruption degree*.

[4]Following convention of prior work. The phenomenon of such models having poor empirical model generalization has been called *memorization* — a convention we continue to follow here.

ImageNet and ResNet-18 trained on CIFAR-10 are shown in Figure 1. Results with 0% and 100% corruption degrees are shown in Figure 10 in Section C.

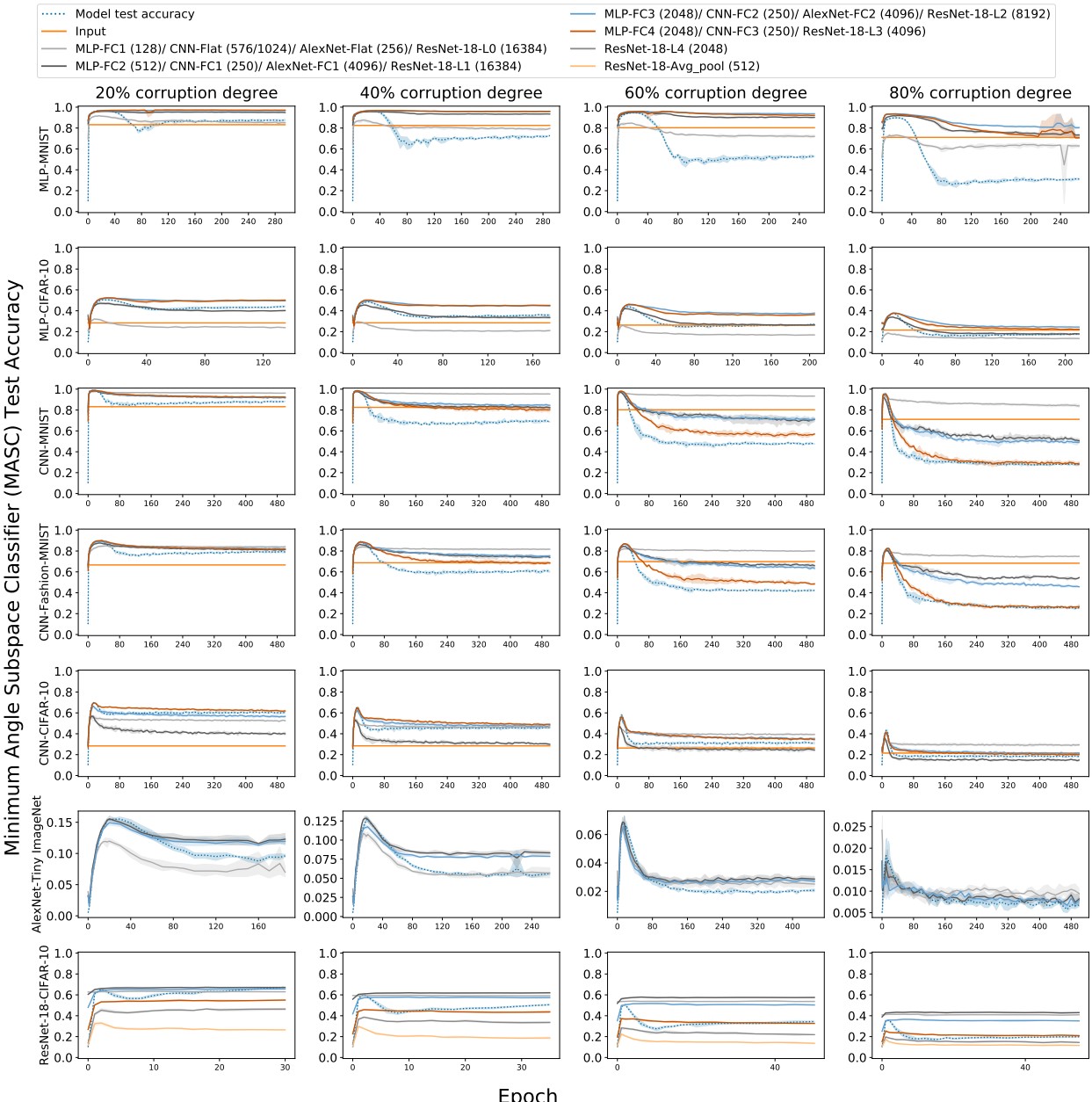

Figure 1: Minimum Angle Subspace Classifier (MASC) test accuracy over epochs of training for multiple models/datasets, where test data is projected onto class-specific subspaces constructed at each epoch from corrupted training data with the indicated label corruption degree. The plots display MASC accuracy across different layers of the network. For reference, the evolution of test accuracy of the corresponding model (blue dotted line) over epochs of training is also shown. FC denotes fully connected layers with *ReLU* activation, and Flat refers to the flatten layer without *ReLU*.

It is known (Arpit et al., 2017) that for models trained with label noise for corruption degrees less than 100%, there is an early rise in model test accuracy that culminates in a peak, subsequent to which the

model test accuracy falls. For various non-zero degrees of corruption, in most cases (except those with 100% degrees of corruption), the MASC test accuracy largely follows the rise in the model's test accuracy up to its peak. However, beyond the peak, while the model's test accuracy declines significantly, the drop in MASC accuracy for the best-performing layers is usually less steep and plateaus at a higher level[5] over the epochs. A notable exception to these observations occurs in the early layers of ResNet-18 trained on CIFAR-10, which we discuss next.

An interesting point about the training dynamics is indeed what occurs prior to the start of training, i.e. just at the random initialization, which we plot as Epoch 0 in Figures 1 and 10. At Epoch 0, the model always has roughly chance-level test accuracy. However, MASC demonstrates[6] test accuracies that are significantly above chance; there is variation in the values depending on the layer in question for each model. However, for most models (except ResNet-18 trained on CIFAR-10), the rise of test accuracies over the initial epochs happens at rates comparable to that of MASC. For this reason, in those models, the model test accuracy as well as the MASC test accuracies appear to be largely overlaid on each other early in training, prior to the peak in test accuracy. For ResNet-18 trained on CIFAR-10 however, MASC test accuracies seem to improve somewhat slower over early epochs. Furthermore, in case of some early layers (i.e. L0, L1), for higher corruption degrees, MASC in fact beats the best early stopping accuracy of the model at Epoch 0, and the MASC test accuracies end up being at levels markedly above model test accuracies all through training. This may have to do with the significantly higher ambient dimensionality (16,384) of those layers. Also, L0 MASC test accuracy appears largely flat[7] over training, whereas L1 MASC test accuracy shows a slight rise early in training before being largely flat.

One question that arises is about why there is a marked difference in MASC test accuracies across layers of the network. A definitive answer likely requires a deeper understanding of the principles used by deep networks to organize layerwise representations and their evolution across layers – an understanding that we don't yet have, as a field. That said, one observes that MASC layer performance appears to be correlated with the ambient dimensionality of the layer, in most cases. That is, the best-performing layers tend to be those with the highest dimensionality. A notable exception is with CNN-CIFAR-10 for lower corruption degrees, where there exist layers that outperform the CNN-Flat (1024) layer. In many models (MLPs, AlexNet, ResNet-18), multiple layers have the highest dimensionality. Here, layer performance differs, albeit not significantly. Furthermore, it does not seem to be determined by their relative position of those layers in the deep network.

Our results represent progress in clarifying the origin & evolution of latent generalization by MASC, during training. In particular, given that model generalization & latent generalization often show a concurrent initial rise, it suggests the possibility of common mechanisms that drive both in the early phases of training, for those models. The subsequent divergence between model generalization & latent generalization is an intriguing phenomenon, whose mechanisms merit future investigation.

## 5 Non-linearity of MASC

Classically (Alain & Bengio, 2018), linear probes have been used to probe layers of deep networks. However, (Ketha & Ramaswamy, 2026) do not use the standard linear probe from (Alain & Bengio, 2018). Linear probes are simple and interpretable, due to which they have been widely deployed. Furthermore, a linear probe demonstrating good performance indicates that the representations of the layer that the probe operates on, contain information that is linearly decodable. This implies that such information is, in principle easily decodable by downstream layers. While MASC is a elegant probe with a nice geometric interpretation, (Ketha & Ramaswamy, 2026) do not consider the question of whether it is a linear probe. Below, we prove that MASC (Ketha & Ramaswamy, 2026) is in fact a non-linear classifier. In particular, it is quadratic in the layerwise output of the layer that it is applied to. This means that it uses a quadratic decision surface.

---

[5]Indeed, as reported in (Ketha & Ramaswamy, 2026), there are layers whose MASC performance is worse than the model performance.

[6]Indeed, the performance of MASC at Epoch 0 for a subset of models has also been reported in (Ketha & Ramaswamy, 2026).

[7]We note that L0 does not have residual connections, unlike subsequent layers, although it is unclear if this contributes to this phenomenon.

**Proposition 1.** *MASC is a quadratic classifier.*

*Proof.* Let $\boldsymbol{x_l}$ denote the output of the layer $l$ of the deep network when it is given input $\boldsymbol{x}$. Let $\boldsymbol{p_1^c}, \boldsymbol{p_2^c}, \ldots, \boldsymbol{p_k^c}$ be an orthonormal basis[8] of the subspace $\mathcal{S}_c$ corresponding to class $c$. Let $\boldsymbol{x_l^c}$ be the projection of $\boldsymbol{x_l}$ on $\mathcal{S}_c$. We have

$$\boldsymbol{x_l^c} = (\boldsymbol{x_l} \cdot \boldsymbol{p_1^c}) \; \boldsymbol{p_1^c} + \ldots + (\boldsymbol{x_l} \cdot \boldsymbol{p_k^c}) \; \boldsymbol{p_k^c} \tag{1}$$

Now, MASC on layer $l$ predicts[9] the label of $\boldsymbol{x}$ as

$$\arg\max_c (\boldsymbol{x_l} \cdot \boldsymbol{x_l^c}) = \arg\max_c ((\boldsymbol{x_l} \cdot \boldsymbol{p_1^c})^2 + \ldots + (\boldsymbol{x_l} \cdot \boldsymbol{p_k^c})^2) \tag{2}$$

which is quadratic in $\boldsymbol{x_l}$. This establishes that MASC is a quadratic classifier. $\square$

# 6 Vector Linear Probe Intermediate-layer Classifier (VeLPIC): A new linear probe

Given that MASC is inherently a non-linear classifier as proved above, a natural question is if its remarkable ability to decode generalization from hidden representations of memorized networks is largely a consequence of its non-linearity. In other words, if the quadratic nature of MASC is indeed responsible for its effectiveness, then a corresponding linear probe would be expected to perform substantially worse. It raises the question of the extent to which the latent generalization reported in (Ketha & Ramaswamy, 2026) is linearly decodable from the layerwise representations of the network.

To investigate this, we build a linear probe analogous to MASC. We sought to retain the same broad idea, namely determine an instance of a mathematical object per class and measure closeness of the layerwise output of an incoming datapoint to these objects with the prediction corresponding to the class whose object was closest in this sense. In contrast to (Alain & Bengio, 2018), where parameters of their linear probe are learned iteratively by minimizing a cross-entropy loss, we seek to determine the linear probe parameters directly via the geometry of the class-conditional training data. We choose to simply use a vector[10] as this mathematical object and measure closeness in the angle sense. We call this probe the Vector Linear Probe Intermediate-layer Classifier (VeLPIC). As we discuss subsequently, we find, surprisingly, that this choice is often more effective than MASC. Secondly, we show that we can use the parameters of the probe as applied to the last layer, to modify the model weights to immediately confer the corresponding generalization to the model.

We now discuss how the vector corresponding to each class in VeLPIC is constructed. Each class vector is determined using only the top principal component from PCA run on augmented[11] class-conditional corrupted training data. However, the first principal component can manifest in two opposite directions (i.e. the vector or its negative). This is important here[12] because incoming data vectors can be "close" to this class vector, even though their angles are obtuse and closer to $180°$. VeLPIC resolves this directional issue by aligning the class vector based on the sign of the projection of the training data mean; if the mean of the training data projected on this principal component is negative, the direction of the principal component is flipped to obtain the class vector; otherwise, it is retained as is.

Formally, for a given test data point $\boldsymbol{x}$, let $\boldsymbol{x_l}$ denote its activation at layer $l$ obtained in the forward pass of $\boldsymbol{x}$ through the deep network until the output of layer $l$. For layer $l$, let $\{\mathcal{P}_m\}_{m=1}^{M}$ be the top principal component vectors, one each per class, of the class-conditional corrupted training data and $\{T_m\}_{m=1}^{M}$ be its corresponding[13] projection means, where $M$ is the number of classes. Let $\{\mathcal{V}_m\}_{m=1}^{M}$ be unit vectors

---

[8]which is typically estimated via PCA, where $k$ is the number of principal components.

[9]This is equivalent to the formulation of MASC in (Ketha & Ramaswamy, 2026), where we maximize a cosine similarity. See Section B.1 for a proof of equivalence.

[10]We note that previous work (Das et al., 2007) has used the idea of using classwise-PCA for classification in the setting of EEG data, although they use a Bayes classifier to perform the classification.

[11]We augment class training data points with their negative, so as to obtain a 1-D subspace, rather than a 1-D affine space, along the lines of the subspace construction procedure for MASC.

[12]Observe that this isn't an issue with MASC, since it is a quadratic classifier.

[13]i.e. $T_i$ is the mean of projecting training data points on $\mathcal{P}_i$.

representing VeLPIC class vectors. VeLPIC uses $\{\mathcal{V}_m\}_{m=1}^M$ to predict the label of $\boldsymbol{x_l}$ based on its maximum projection among these class vectors[14], as outlined in Algorithm 1.

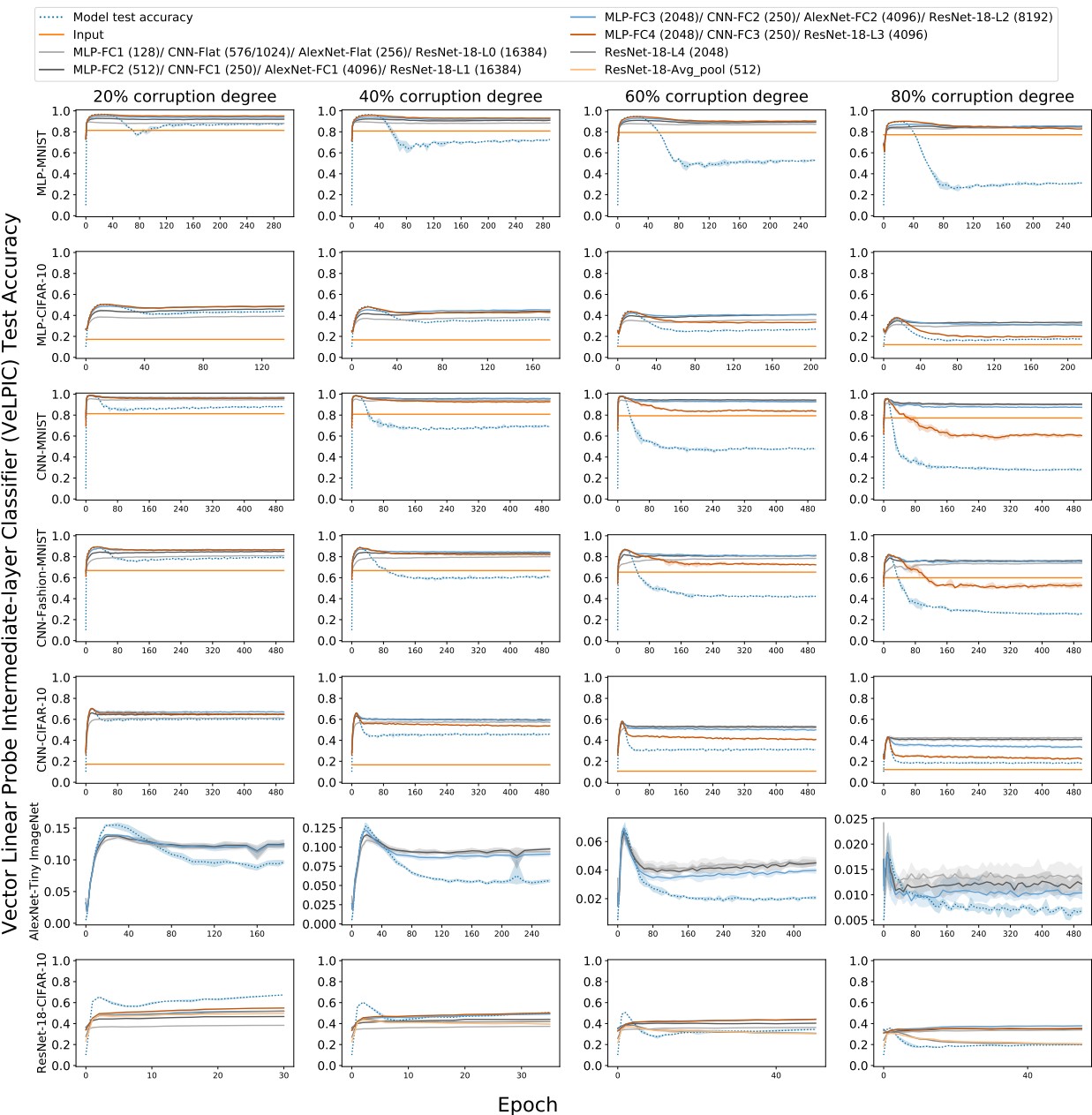

Figure 2: Vector Linear Probe Intermediate-layer Classifier (VeLPIC) test accuracy during training of the network, where test data is projected onto class vectors constructed at each epoch from training data with the indicated label corruption degrees. The plots display VeLPIC accuracy across different layers of the network for various model–dataset combinations. For reference, the test accuracy of the models (blue dotted line) over epochs of training is also shown. FC denotes fully connected layers with *ReLU* activation, and Flat refers to the flatten layer without *ReLU*.

---

[14]This is equivalent to minimum angle to the VeLPIC class vectors.

---

**Algorithm 1 Vector Linear Probe Intermediate-layer Classifier (VeLPIC)**

---

**Input:** Principal component vectors $\{\mathcal{P}_m\}_{m=1}^M$, projection training means $\{T_m\}_{m=1}^M$, layer $l$ output $\boldsymbol{x_l}$, class labels $\{C_m\}_{m=1}^M$.
**Output:** Predicted label $y(\boldsymbol{x_l})$.

1: **for** each class $m = 1, \ldots, M$ **do**
2:    **if** $T_m < 0$ **then**
3:       $\mathcal{V}_m \leftarrow -\mathcal{P}_m$
4:    **else**
5:       $\mathcal{V}_m \leftarrow \mathcal{P}_m$
6:    **end if**
7: **end for**
8: **for** each class $m = 1, \ldots, M$ **do**
9:    $\boldsymbol{x_{lm}} \leftarrow$ Projection of $\boldsymbol{x_l}$ onto $\mathcal{V}_m$
10: **end for**
11: $y(\boldsymbol{x_l}) \leftarrow C_j$ where $j = \arg\max_m \boldsymbol{x_{lm}}$
12: **Return:** $y(\boldsymbol{x_l})$

---

## 6.1 Training dynamics of the linear probe

Here, we examine the performance of VeLPIC, in contrast to that of MASC during training, to assess the extent of linear decodability of latent generalization. It is worth mentioning that favorable performance of VeLPIC implies high linear decodability; however, poor performance of VeLPIC does not rule out the possibility of the existence of another linear probe with superior performance.

VeLPIC test accuracy during training for MLP-MNIST, MLP-CIFAR-10, CNN-MNIST, CNN-Fashion-MNIST, CNN-CIFAR-10, AlexNet-Tiny ImageNet and ResNet-18-CIFAR-10 are shown in Figure 2. The results with 0% and 100% corruption degrees are shown in Figure 11 in Section D. The difference between VeLPIC test accuracy and MASC test accuracy are shown in Figure 3. Results for 0% and 100% corruption degrees are available in Section D.1.

Unexpectedly, the performance of VeLPIC is often, but not always, better than that of MASC. For representations from many layers, VeLPIC is able to extract significantly better latent generalization performance than MASC and our results show that, for these layers, VeLPIC's performance plateaus at significantly higher levels than MASC. There also exist many cases, where MASC does indeed outperform VeLPIC. In particular, in ResNet-18[15] trained on CIFAR-10, MASC substantially outperforms VeLPIC's test accuracy in layers L0 and L1, which were the best performing layers for MASC in this model. This suggests the possibility that indeed in these layers, MASC utilizes its nonlinearity in ways that VeLPIC is not able to. There is also the possibility of other[16] linear probes that may be able to do better for this model. Similarly, in multiple other cases (e.g. early layers of CNN-MNIST, CNN-Fashion-MNIST, later layers of MLP-MNIST, MLP-CIFAR-10), MASC outperforms VeLPIC.

One factor that appears to be correlated with poor linear decodability as manifested by VeLPIC, is the ambient dimensionality of layerwise outputs. Recall that, in the previous section, we observed that MASC appears to be quite effective when the ambient dimensionality of the layer in question is high. The results in this section suggest that VeLPIC performance is concomitantly lower in such layers, in comparison to MASC. This hypothesis is especially supported by results from layers L0 and L1 of ResNet-18, which have the highest layerwise dimensionality of 16,384. However, results from CNN-CIFAR-10 run contrary to this hypothesis. Here, VeLPIC outperforms MASC in the CNN-Flat layer, with dimensionality 1024, which has the highest dimensionality of all layers tested in this model. Therefore, while ambient dimensionality may

---

[15]Which is arguably, the most modern network tested. We do note, however, that AlexNet is a larger model, by the measure of number of trainable parameters.

[16]As we show in the next section, the only other baseline linear probe tested, in fact shows worse performance than VeLPIC for these two layers, for higher corruption degrees.

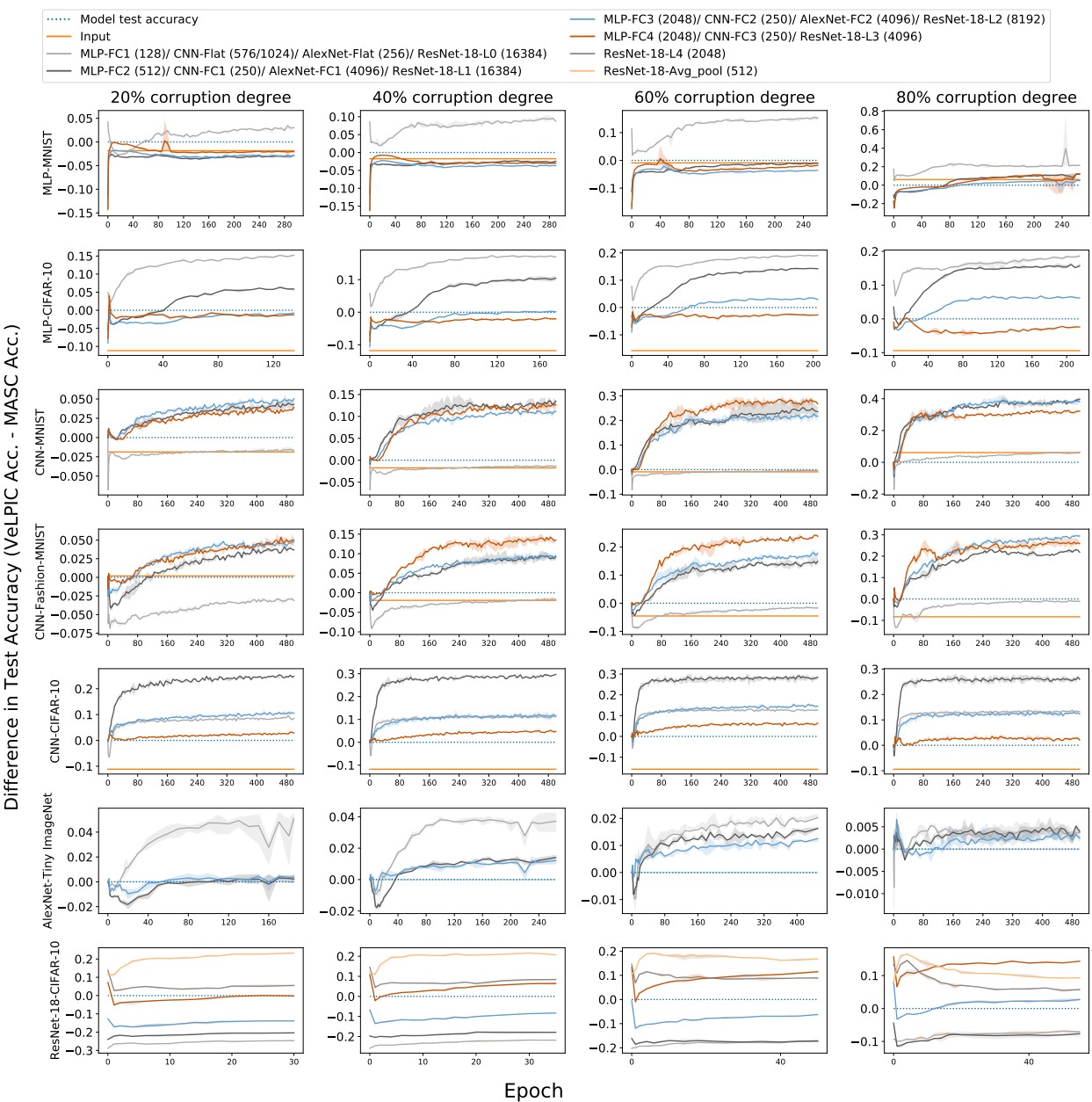

Figure 3: Difference in test accuracy (VeLPIC Accuracy - MASC Accuracy) during training of the network, where test data is projected onto class vectors constructed at each epoch from training data with the indicated label corruption degrees. The plots display difference in accuracy across different layers of the network for various model–dataset combinations. For reference, the test accuracy of the models (blue dotted line) over epochs of training is also shown, which would be 0.

be an important factor, the extent of linear decodability or non-decodability of latent generalization during memorization might involve an interplay of multiple factors that require careful investigation.

Interestingly, with some CNN models (e.g. CNN-Fashion-MNIST), MASC decodes good generalization in early layers but not later layers; however, VeLPIC is able to extract comparable generalization from later layers as well.

We compare the test accuracy of the model, MASC, and VeLPIC for both randomly initialized and trained models across varying corruption degrees in Section H. Furthermore, we study the impact of dropout as a regularizer in latent generalization (MASC & VeLPIC) during memorization in Section I.

In closing, our results with the new linear probe indicate that there are multiple cases where one can linearly decode latent generalization to a greater extent than reported with MASC. Here, as well, it is unclear if there exist other quadratic classifiers that can extract better latent generalization than VeLPIC[17]. On the other hand, there are important exceptions where MASC still significantly outperforms this linear probe, especially for ResNet-18, which is the most modern network architecture tested. It is unclear if the underperformance of VeLPIC is because a significant measure of latent generalization in these cases is not linearly decodable. Alternatively, there could exist other linear probes that could match or outperform MASC in these cases.

## 7 Comparing MASC & VeLPIC with a baseline linear probe

While VeLPIC is indeed a linear probe, it differs from standard linear probes (Alain & Bengio, 2018) which are trained by iteratively minimizing a suitable loss function, unlike VeLPIC whose parameters are directly derived from the class-conditional geometry of internal representations. In order to compare the relative performance of MASC and VeLPIC with a classic linear probe, here, we study the evolution of latent generalization during training using a logistic regression probe (LR probe).

For a given model and network layer, we train a logistic regression probe on the outputs of the layer for 20 epochs using cross-entropy loss with the Adam optimizer (for learning rate $1 \times 10^{-3}$). The test accuracy of the logistic regression probe during training of the memorized models, for all models and corruption degrees ranging from 20% to 80%, is shown in Figure 4. Corresponding results for 0% corruption degree is presented in Figure 13. For the non-zero corruption degrees, we find interestingly that in most cases, there is at least a layer or two, whose LR probe tends to largely follow the dynamics of the model's test accuracy over training, including after the early peak, with ResNet-18 being a notable exception. Secondly, in many cases where the test accuracy of the LR probe does not follow that of the model (e.g. CNN-CIFAR-10 for higher corruption degrees), the test accuracies of the LR probe after the early peak tend to plateau at a lower level than is the case with MASC and VeLPIC probes. Thirdly, for many layers, the accuracy of the LR probe tends to underperform[18] both MASC and VeLPIC.

We compare the performance of the logistic regression probe with that of the MASC probe in Figures 5 and 14. We find that MASC outperforms LR for many layers in multiple models; however, there are several layers where the opposite effect is true. There is also some regularity in the identity of layers where this happens. For some models (MLPs, CNN-CIFAR-10 and AlexNet-Tiny ImageNet) the LR probe outperforms MASC in the early layers and vice versa in the later layers. For other models (CNN-MNIST, CNN-Fashion-MNIST, ResNet-18-CIFAR-10) one sees the opposite effect.

We further compare the logistic regression probe results with the VeLPIC probe in Figures 6 and 15. Here, we find that especially for higher corruption degrees and later during model training, VeLPIC almost always outperforms the LR probe. Even for low corruption degrees, in most layers (with ResNet-18-CIFAR-10 on 20% corruption degree being a notable exception), VeLPIC outperforms the LR probe.

A natural question that arises here is about why the logistic regression probe underperforms VeLPIC with respect to generalization to true labels, even though both of them are linear classifiers. We speculate that this might have to do with the fact that classic linear probes seek to minimize a loss which corresponds to doing well on the corrupted training set, which can be antithetical to doing well on the test set which has true labels. However, techniques such as MASC and VeLPIC have an unsupervised flavor to them that does not seek to optimize training set performance explicitly, even though they operate on the same corrupted training set. This view requires careful future investigation.

---

[17]A general quadratic classifier, in principle, will do as well as VeLPIC, by keeping only its linear terms and setting its quadratic terms to zero.

[18]The corresponding differences are plotted in Figures 5 and 6.

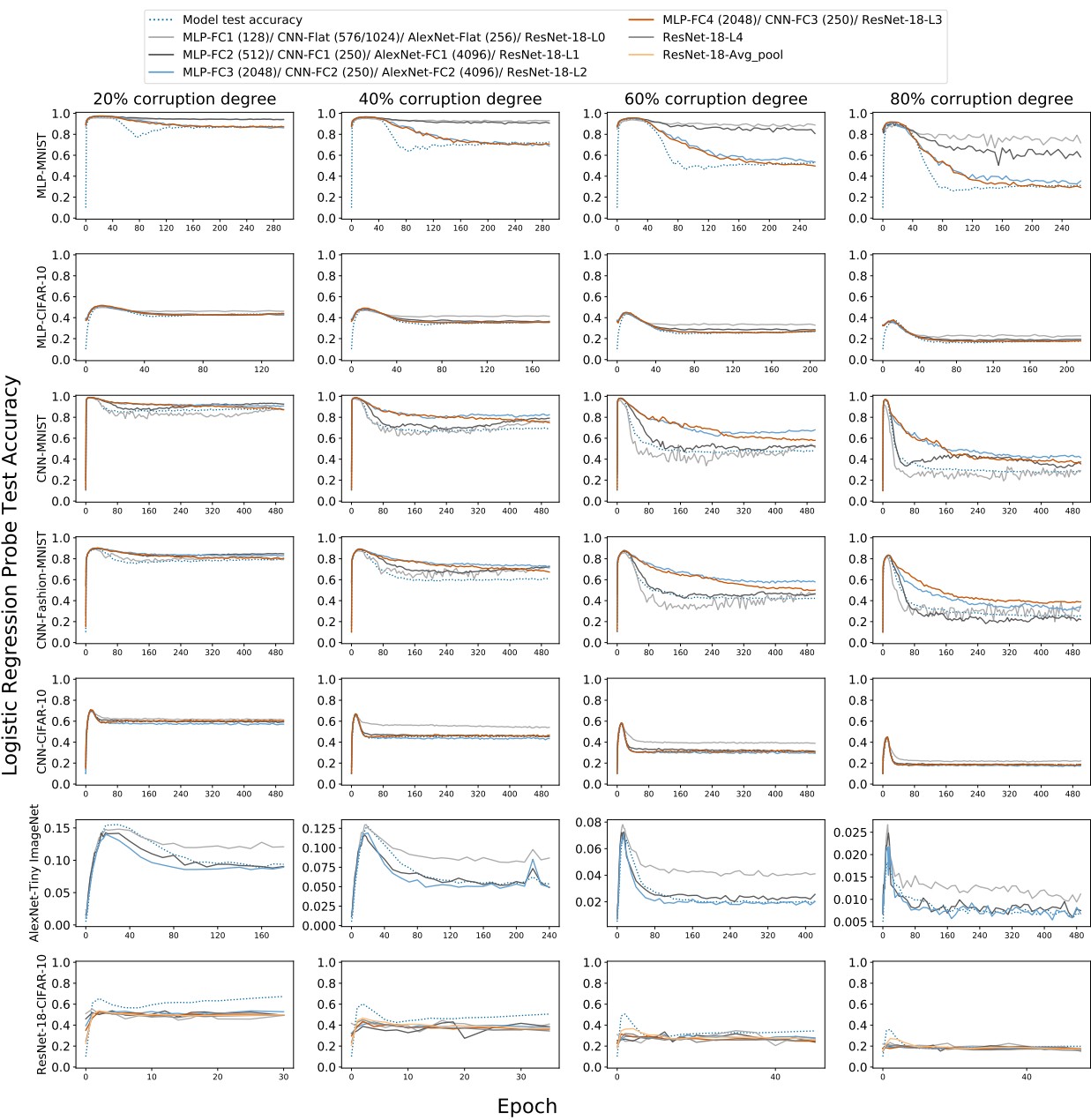

Figure 4: Logistic regression probe's test accuracy over epochs of training for multiple models/datasets. The plots display logistic regression probe's accuracy across different layers of the network. For reference, the evolution of test accuracy of the corresponding model (blue dotted line) over epochs of training is also shown.

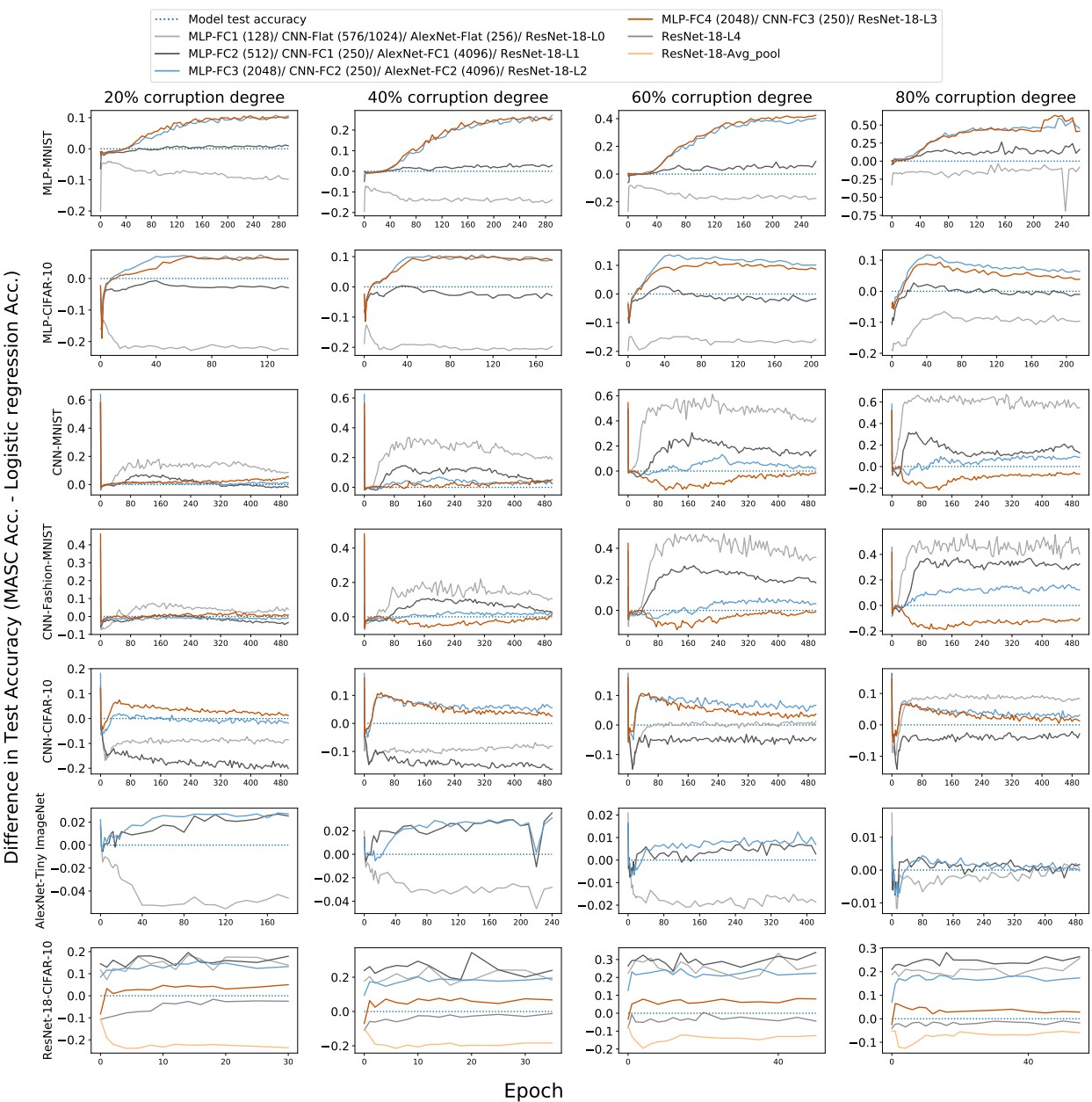

Figure 5: Difference in test accuracy (MASC Accuracy - Logistic regression probe Accuracy) during training of the network, where for MASC test data is projected onto class vectors constructed at each epoch from training data with the indicated label corruption degrees. The plots display difference in accuracy across different layers of the network for various model–dataset combinations. For reference, the test accuracy of the models (blue dotted line) over epochs of training is also shown, which would be 0.

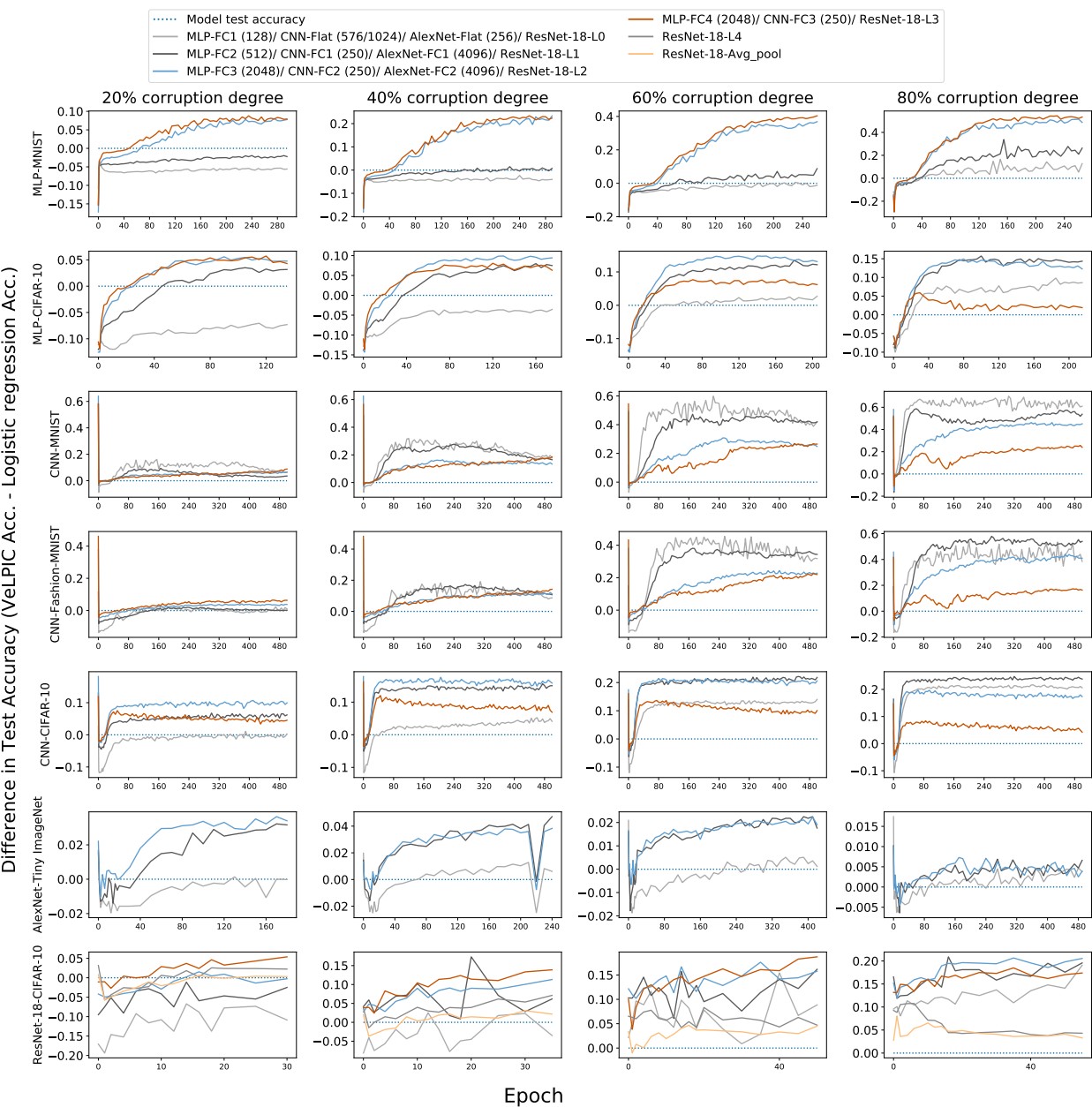

Figure 6: Difference in test accuracy (VELPIC Accuracy - Logistic regression probe Accuracy) during training of the network, where for VELPIC test data is projected onto class vectors constructed at each epoch from training data with the indicated label corruption degrees. The plots display difference in accuracy across different layers of the network for various model–dataset combinations. For reference, the test accuracy of the models (blue dotted line) over epochs of training is also shown, which would be 0.

## 8    Transferring latent generalization to model generalization

Given that latent generalization often exceeds the models' generalization ability, a natural question is whether this hidden latent generalization can be transferred directly to the model by appropriately modifying its weights. An affirmative answer to this question has implications that the phenomenon of latent generalization is not mainly driven by the remarkable effectiveness of the probe (MASC) used in our prior work (Ketha & Ramaswamy, 2026). Conceptually, this question is significant because it demonstrates that the probe's latent generalization is inherently accessible to the model using only the corrupted data it was trained on. Yet, the model during the act of standard training does not end up generalizing to the same extent, for reasons that we don't understand well. Practically, it raises the possibility of repairing memorized model using the latent representations, without the cost of retraining from scratch.

Here, we ask if the latent generalization in models that memorize, can be directly transferred to the model, in order to immediately improve its generalization. To this end, it turns out that the class vectors of VeLPIC applied to the last layer can be directly substituted in the pre-softmax layer of the model as an intervention that transfers VeLPIC's generalization performance to the model, without further training. We elaborate below on how this is so.

Consider a model whose last layer (i.e. the layer preceding the pre-softmax layer) consists of $d$ units. Let $\boldsymbol{v}_j \in \mathbb{R}^d$ be the VeLPIC class vector for class $j$. The new pre-softmax weight matrix $\mathbf{W}_{\text{pre-softmax}} \in \mathbb{R}^{M \times d}$ is constructed as:

$$\mathbf{W}_{\text{pre-softmax}} = \left( \begin{bmatrix} \boldsymbol{v}_1 & \boldsymbol{v}_2 & \cdots & \boldsymbol{v}_M \end{bmatrix} \right)^{\top} \tag{3}$$

This weight matrix $\mathbf{W}_{\text{pre-softmax}}$ replaces the original pre-softmax weights, and all biases are set to zero. It is straightforward to see that this substitution results in the model making the same predictions as VeLPIC applied to the last layer. While this substitution is fairly straightforward, it needs the linear probe, i.e. VeLPIC, for it to work. In particular, we note that it is unclear if transferability can happen with MASC alone and therefore the development of VeLPIC was an enabling factor in helping us answer this question.

During model training, we replace the pre-softmax weights with VeLPIC vectors, as indicated above and evaluate the model's performance on the test dataset at each epoch. Figure 7 presents these results for MLP-MNIST, MLP-CIFAR-10, CNN-MNIST, CNN-Fashion-MNIST, CNN-CIFAR-10, AlexNet-Tiny ImageNet and ResNet-18-CIFAR-10. Results for models with 0% and 100% corruption levels, for all model-dataset pairs are presented in Figure 16 in Section F.

We observe that, in most cases, the weight intervention that replaces pre-softmax weights with the VeLPIC vectors leads to an immediate & significant improvement in generalization performance in every epoch of the latter phase of training, matching that of the linear probe, & in particular, without any further training. A notable exception is for ResNet-18, where, especially for lower corruption degrees the edited model has worse generalization performance. For this model, for none of the probes tested, is it the case that later layers manifest better latent generalization than the model, due to which this weight editing technique does not lead to better model generalization either. An interesting question is if one could develop weight editing techniques that can transfer exactly or approximately probe performance on early layers as well, to the model. In closing, we establish in this section that the latent generalization in memorized models can be directly harnessed in many cases to enhance their test performance, even in the presence of label noise.

## 9    Weight intervention during training using VeLPIC

Having shown that latent generalization can be directly transferred into the model by editing its weights, a natural question arises: What happens if we inject/transfer this latent generalization to the model at a specific epoch and continue standard training thereafter?

We ask whether intervening during training – by updating the model weights at a specific epoch using information derived from its latent generalization – can enhance model generalization. Additionally, we investigate how latent generalization (VeLPIC) evolves across different layers throughout training, when we do so.

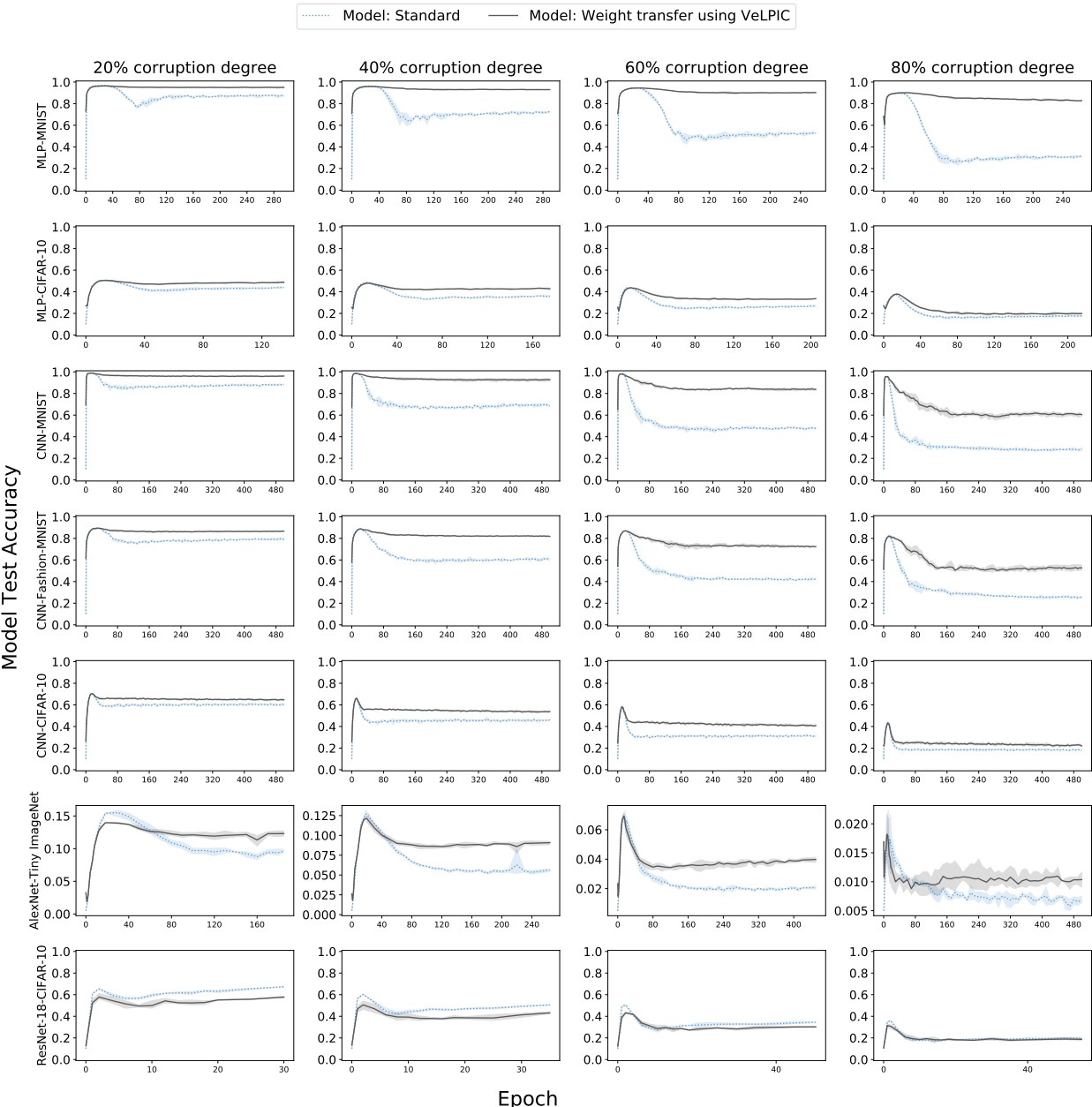

Figure 7: Model test accuracy when the weight intervention is applied to the epoch in question during training. The test accuracy of the model with standard training without weight intervention (blue dotted line) is overlaid for comparison.

To address these questions, the model is trained with corrupted data for the first 40 epochs using standard training. The intervention is performed at the 40th epoch by replacing the pre-softmax weights with VeLPIC vectors (last layer), as done in the previous section. Standard training is performed for the next 60 epochs using corrupted training data. Model test accuracy on true labels is shown in Figure 8, overlaid with the case of no intervention, for comparison. VeLPIC test accuracy during training when this intervention is applied at the 40th epoch for different corruption degrees in Figure 9 and specifically for 0% corruption degree in Figure 19.

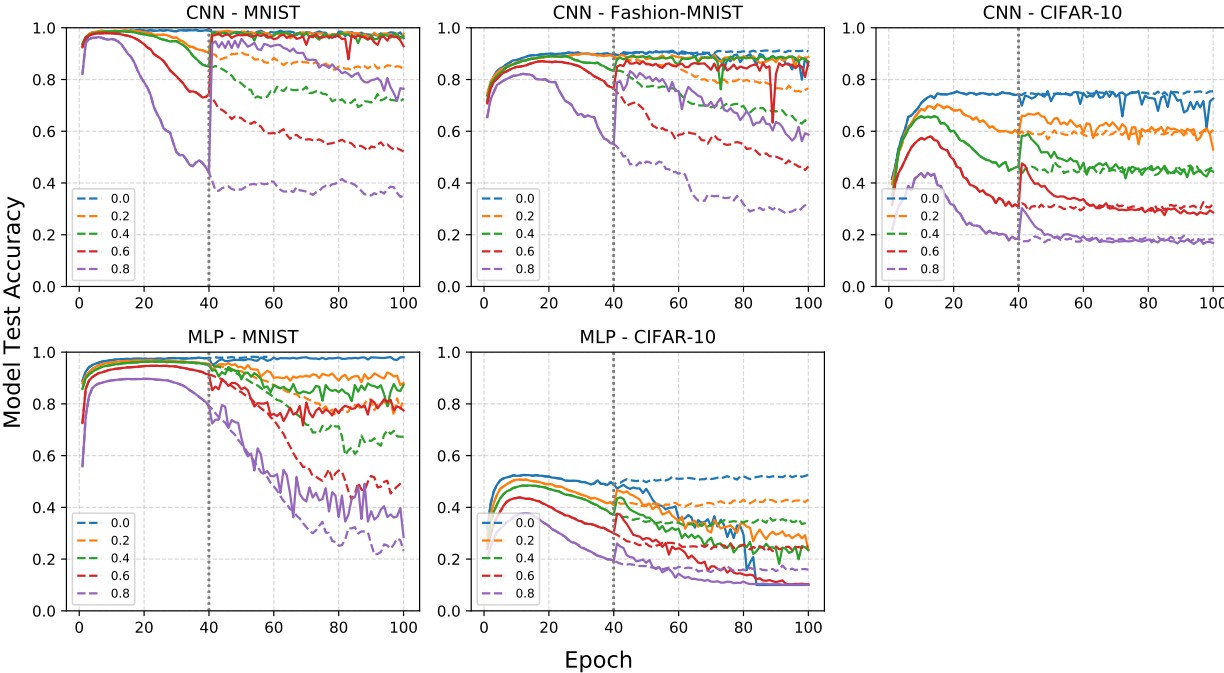

Figure 8: Model test accuracy with true labels during training when a weight intervention that involves replacing pre-softmax weights with the VeLPIC vector, is performed at the 40th epoch and standard training is performed thereafter for 60 epochs. The corresponding test accuracies for the model with standard training (dotted) without intervention is overlaid for comparison. The results correspond to a single run in each case.

As expected from results in Section 6.1, upon applying the intervention at the 40th epoch, there is typically a rise in model generalization (i.e. test accuracy of the model). In many cases, this model generalization degrades subsequently over training, albeit not typically to the degree that model generalization would have degraded without the intervention; there are exceptions. One exception to this is MLP-CIFAR-10 at 80% corruption degree, wherein the model generalization is ultimately worse post-intervention than in the no-intervention case. On the contrary, CNN-MNIST and CNN-Fashion-MNIST, show little degradation post-intervention for low and modest corruption degrees. Also, e.g. with CNN-CIFAR-10, barring the transient rise in model generalization immediately after the intervention, the model generalization with or without the intervention seem to follow largely similar trajectories. We also plotted accuracies when VeLPIC was applied at each epoch over training for models during the same interventions. Here again results are mixed and largely follow trends of the dynamics of model generalization, post-intervention. However, by-and-large, it appears that applying VeLPIC with standard training, and without intervention, on a suitably chosen layer, usually yields the best generalization at any epoch over training, in comparison to alternatives considered here.

## 10  Discussion

The notion of memorization, where deep networks are able to perfectly learn noisy training data at the expense of generalization has posed a challenge to traditional notions of generalization from Statistical Learning Theory (Zhang et al., 2017; 2021). Our recent work (Ketha & Ramaswamy, 2026) demonstrating improved latent generalization in such models is a new development in our understanding of memorization and the nature of representations that drive it. Our goal here was to take a deeper dive into this phenomenon, to investigate the origin and dynamics of latent generalization and examine the possibility of directly transferring it to the model. We showed that early-on in training, latent generalization and the model's generalization

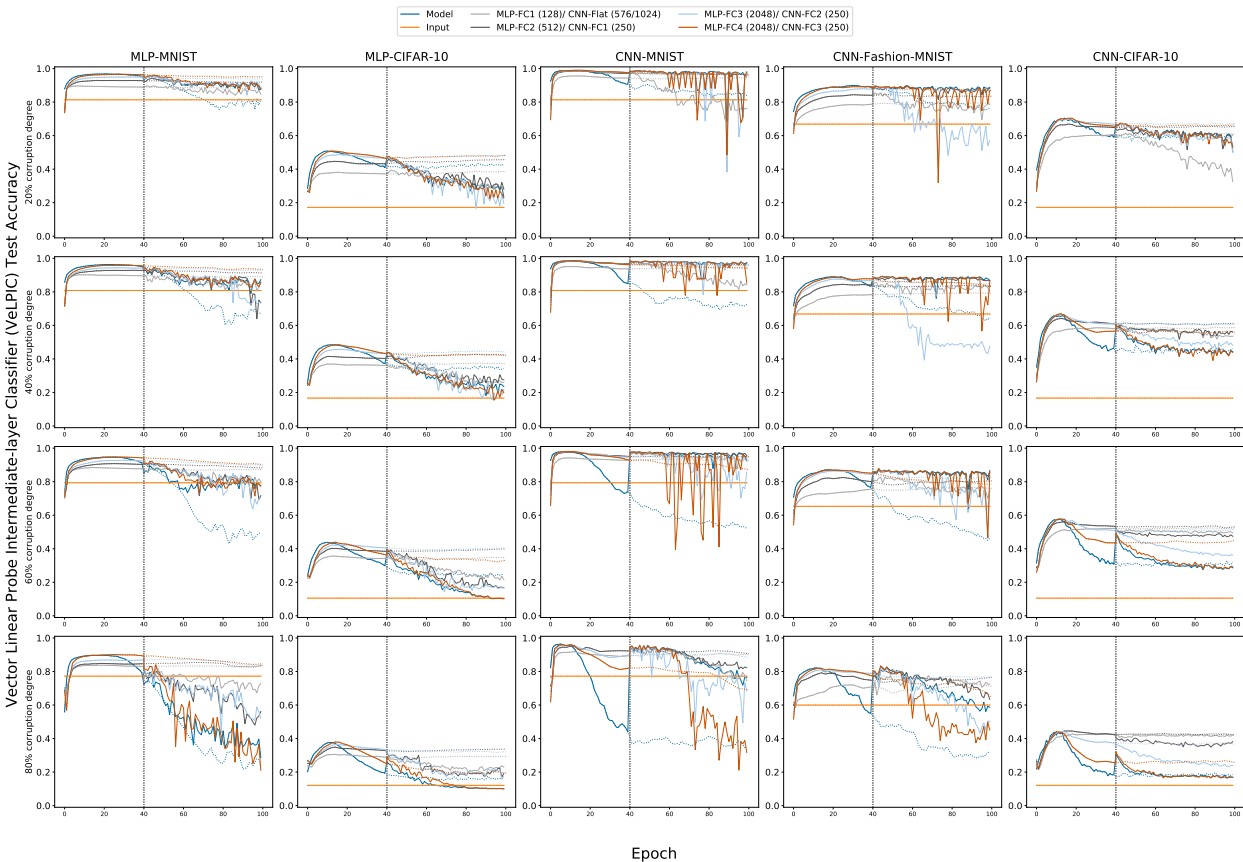

Figure 9: Vector Linear Probe Intermediate layer Classifier (VeLPIC) test accuracy during training on models when intervention is performed at 40th epoch and standard training is performed thereafter for 60 epochs. The VeLPIC test accuracy on models trained with standard training (dotted) without intervention is overlaid for comparison. Model test accuracy for models trained with and without intervention is overlaid for comparison. The results correspond to a single run in each case.

closely follow each other in many cases, suggesting the possibility of common mechanisms that contribute to both. However, later in training, there is a divergence, with the model often retaining significant latent generalization ability in one or more layers, while sacrificing overt model generalization to a greater degree. Next, we showed that MASC (Ketha & Ramaswamy, 2026) is a quadratic classifier and built a new linear probe (VeLPIC). We found cases where this linear probe outperforms MASC and also found other cases, in particular, with ResNet-18, where MASC appears to be able to extract significantly more latent generalization from early layers than VeLPIC. We also ran comparisons of MASC and VeLPIC to a baseline linear logistic regression (LR) probe trained using standard techniques and find that MASC and VeLPIC outperform the LR probe in many layers. We were also interested in examining if the latent generalization could readily be translated to model generalization by directly editing model weights. We utilized the VeLPIC probe to derive a new set of model pre-softmax weights to make this so. We also briefly examined the effect of further training upon so editing the weights in the 40th epoch, and find that this yields mixed results.

There are multiple limitations in this work. While we have carried out extensive experiments on a number of models and datasets, how the phenomenon of latent generalization and its dynamics over training depends on model architecture and nature of the dataset isn't yet clear. Secondly, we have explored in detail the dynamics of MASC, when the class-conditioned subspaces explain 99% variance of training data. MASC performance for other class-conditioned subspaces remains to be explored. Thirdly, an advantage of MASC lies in its ability to capture generalization when it occurs within a subspace of dimension greater than one.

This behavior is particularly evident in the early layers of ResNet-18, where MASC generalizes significantly better than both VeLPIC and the model. In contrast, VeLPIC provides a practical technique that its generalization performance at the final layer can be directly transferred to the model, an extension that is not yet available for MASC. Finally, we haven't explored the phenomenon of latent generalization in transformer models. Memorization has been reported Zhang et al. (2017) in cases where input is corrupted instead of class labels. This is a memorization setting that we have not studied here.

This work brings up multiple new directions for investigation. While we have made some progress, the detailed mechanisms governing latent generalization during memorization remain to be investigated. It is also an open question, whether there exist other probes that can extract better latent generalization from layerwise representations, in comparison to MASC and VeLPIC. Next, it is unclear if latent generalization from representations of layers other than the last layer can be transferred towards model generalization. This can be useful to do, in cases where early or middle layers exhibit better latent generalization than the last layer. Furthermore, what causes latent generalization during memorization to be or not to be linearly decodable is an open question. To what extent does model architecture, e.g. ResNet-like vs. AlexNet-like influence the answer. Do we need fundamentally different ways to remove effects of memorized labels in these differing network architectures? Answering these questions might need a deeper understanding of representations used by deep networks during memorization. More generally, in light of these results, whether an understanding of generalization in the memorization regime can inform a better understanding of generalization for models trained with uncorrupted labels is a worthwhile direction for future investigation.

In closing, our results highlight the rich role of representations in driving generalization during memorization and how their understanding can be utilized, in many cases, to directly improve model generalization.

### Acknowledgments

Simran Ketha was supported by an APPCAIR Fellowship, from the Anuradha & Prashanth Palakurthi Centre for Artificial Intelligence Research. The work was supported in part by an Additional Competitive Research Grant from BITS to Venkatakrishnan Ramaswamy. The authors acknowledge the computing time provided on the High Performance Computing facility, Sharanga, at the Birla Institute of Technology and Science - Pilani, Hyderabad Campus.

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

# A    Model architectures and training details

**MLP Model.** The MLP architecture consists of four hidden layers with 128, 512, 2048, and 2048 units, respectively. Each layer is followed by a *ReLU* activation, and a *softmax* layer is used for classification. Models were trained SGD Qian (1999) with a learning rate of $1 \times 10^{-3}$ and momentum 0.9. A batch size of 32 was used across all experiments. Input dataset was normalized by dividing pixel values by 255.

**CNN Model.** The CNN model[19] is composed of three convolutional blocks, each containing two convolutional layers followed by a max pooling layer. The convolutional layers use 16, 32, and 64 filters, respectively, with kernel size $3 \times 3$ and stride 1. The max pooling layers have a kernel size of $2 \times 2$ and stride 1. These blocks are followed by three fully connected layers with 250 units each. ReLU activation is used after all layers except pooling, and softmax is used at the output for classification. The CNN was trained using Adam optimizer Kingma (2014) with a learning rate of 0.0002. For MNIST and Fashion-MNIST, a batch size of 32 was used, while for CIFAR-10, a batch size of 128 was used. Input data was normalized by subtracting the mean and dividing by the standard deviation of each channel.

ResNet-18 was slightly adapted for CIFAR-10 dataset and trained using *SGD* (learning rate=0.001, momentum=0.9) with a batch size of 32. Inputs were normalized using channel-wise mean and standard deviation. We analyze six layers: L0–L4, and the average-pooling (avg_pool) layer. L0 denotes the activation immediately before the first residual block, while L1–L4 correspond to the outputs of successive residual blocks.

The experiments were conducted on servers and workstations equipped with NVIDIA GeForce RTX 3080, RTX 3090, Tesla V100, and Tesla A100 GPUs. The server runs on Rocky Linux 8.10 (Green Obsidian), while the workstation uses Ubuntu 20.04.3 LTS. Memory requirements varied depending on the specific experiments and models. All model implementations were developed in Python using the PyTorch library, with torch.manual_seed set to 42 to ensure reproducibility. Accuracy served as the primary evaluation metric throughout this work.

# B    Minimum Angle Subspace Classifier (MASC)

We summarize below the Minimum Angle Subspace Classifier (MASC) from (Ketha & Ramaswamy, 2026), in order to keep the exposition here largely self-contained.

For a given deep network, MASC leverages the class-specific geometric structure of network's latent representations. For an input data point $\boldsymbol{x}$, let its activation vector at layer $l$ be denoted by $\boldsymbol{x_l}$. The objective is to classify $\boldsymbol{x_l}$ by leveraging a set of class-conditional subspaces, $\{S_k\}_{k=1}^{K}$, estimated from a training dataset $\mathcal{D} = \{(\boldsymbol{x_i}, y_i)\}_{i=1}^{m}$. To predict the class label $y(\boldsymbol{x_l})$, MASC Algorithm 2 (reproduced verbatim from (Ketha & Ramaswamy, 2026)), assigns $\boldsymbol{x_l}$ to the class whose training subspace forms the smallest angle with it.

The class-conditional subspaces $\{S_k\}_{k=1}^{K}$ are estimated from the training dataset $\mathcal{D} = \{(\boldsymbol{x_i}, y_i)\}_{i=1}^{m}$, where each $\boldsymbol{x_i} \in \mathbb{R}^d$ is paired with a label $y_i \in \{C_k\}_{k=1}^{K}$. For a given layer $l$, these subspaces are constructed following Algorithms 3 and 4 (reproduced verbatim from (Ketha & Ramaswamy, 2026)). In practice, each subspace $S_k$ is represented by its principal components, which provide a compact basis for capturing the underlying class-conditional structure.

---

[19]The convolution network were implemented following the design principals outlined in (Tran et al., 2022).

**Algorithm 2 Minimum Angle Subspace Classifier (MASC)** (reproduced verbatim from (Ketha & Ramaswamy, 2026))

---

**Input:** Training subspaces $\{S_k\}_{k=1}^K$, layer output data point $\boldsymbol{x_l}$ from layer $l$ when input $\boldsymbol{x}$ is passed through the network and classes $\{C_k\}_{k=1}^K$.

**Output:** MASC prediction class label $y(\boldsymbol{x_l})$ according to layer $l$ .

1: **for** each class $C_k$ **do**
2:    $\boldsymbol{x_{lk}} \longleftarrow$ compute the projection of $\boldsymbol{x_l}$ onto subspace $S_k$.
3:    Compute the angle $\theta(\boldsymbol{x_l}, \boldsymbol{x_{lk}})$ between $\boldsymbol{x_l}$ and $\boldsymbol{x_{lk}}$
4: **end for**
5: Assign the label $y(\boldsymbol{x_l}) = C_k$ where $k = \arg\min_k \theta(\boldsymbol{x_l}, \boldsymbol{x_{lk}})$
6: **Return:** label $y(\boldsymbol{x_l})$

---

**Algorithm 3 Subspaces Estimator for MASC**
(reproduced verbatim from (Ketha & Ramaswamy, 2026))

---

**Input:** Training dataset $\mathcal{D}\{(\boldsymbol{x_i}, y_i)\}_{i=1}^m$, where each $\boldsymbol{x_i} \in \mathbb{R}^d$ and $y_i \in \{C_k\}_{k=1}^K$ are input-label pairs, neural network, and layer $l$.

**Output:** Subspaces $\{S_k\}_{k=1}^K$ for classes $K$ and given layer $l$.

1: $\mathcal{D}_l = \phi$
2: **for** each input pair $(\boldsymbol{x_i}, y_i)$ in $\mathcal{D}$ **do**
3:    Pass $\boldsymbol{x_i}$ through the network layers to obtain the output of layer $l$, denoted as $\boldsymbol{x_l} \in \mathbb{R}^{ld}$.
4:    $\mathcal{D}_l = \mathcal{D}_l \cup \{(\boldsymbol{x_l}, \boldsymbol{y_i})\}$
5: **end for**
6: Estimated subspaces $\{S_k\}_{k=1}^K \longleftarrow$ **PCA-Based Subspace Estimation**$(\mathcal{D}_l)$
7: **Return:** Subspaces $\{S_k\}_{k=1}^K$

---

**Algorithm 4 PCA-Based Subspace Estimation**
(reproduced verbatim from (Ketha & Ramaswamy, 2026))

---

**Input:** Layer output $\mathcal{D}_l = \{(\boldsymbol{x_i}, y_i)\}_{i=1}^m$, where $\boldsymbol{x_l} \in \mathbb{R}^{ld}$ and $y_i \in \{C_k\}_{k=1}^K$.

**Output:** Subspaces $\{S_k\}_{k=1}^K$ for classes $K$.

1: $\mathcal{D}_{\text{new}} \leftarrow \mathcal{D}_l$
2: **for** each $(\boldsymbol{x_i}, \boldsymbol{y_i}) \in \mathcal{D}_l$ **do**
3:    $\mathcal{D}_{\text{new}} \leftarrow \mathcal{D}_{\text{new}} \cup \{(-\boldsymbol{x_i}, \boldsymbol{y_i})\}$
4: **end for**
5: **for** each $k \in \{1, \dots, K\}$ **do**
6:    Extract the subset of data $\mathcal{D}_{\text{new},k} = \{\boldsymbol{x_i} \mid y_i = C_k\}$
7:    $S_k = \text{PCA}(\mathcal{D}_{\text{new},k})$
8: **end for**
9: **Return:** Subspaces $\{S_k\}_{k=1}^K$

---

### B.1 Proof of equivalence of MASC and classifier used in Proposition 1 of Section 5

In Proposition 1 of Section 5, we show that MASC is a quadratic classifier. However, we used a slightly different formulation of MASC in Proposition 1. Here, we show that, that formulation is equivalent to the formulation of MASC from (Ketha & Ramaswamy, 2026).

Let $\boldsymbol{x_l}$ denote the output of the layer $l$ of the deep network when it is given input $\boldsymbol{x}$. Let $\boldsymbol{p_1^c}, \boldsymbol{p_2^c}, \ldots, \boldsymbol{p_k^c}$ be an orthonormal basis[20] of the subspace $\mathcal{S}_c$ corresponding to class $c$. Let $\boldsymbol{x_l^c}$ be the projection of $\boldsymbol{x_l}$ on $\mathcal{S}_c$. We have

$$\boldsymbol{x_l^c} = (\boldsymbol{x_l} \cdot \boldsymbol{p_1^c}) \, \boldsymbol{p_1^c} + \ldots + (\boldsymbol{x_l} \cdot \boldsymbol{p_k^c}) \, \boldsymbol{p_k^c} \tag{4}$$

$$\hat{\boldsymbol{x_l^c}} = \frac{\boldsymbol{x_l^c}}{|\boldsymbol{x_l^c}|} \tag{5}$$

MASC(Ketha & Ramaswamy, 2026) on layer $l$ predicts the label of $\boldsymbol{x}$ as

$$\arg\max_c \left( \hat{\boldsymbol{x_l}} . \hat{\boldsymbol{x_l^c}} \right) \tag{6}$$

The formulation in Proposition 1 of Section 5 predicts the label of $\boldsymbol{x}$ as

$$\arg\max_c \left( \boldsymbol{x_l} . \boldsymbol{x_l^c} \right) \tag{7}$$

**Proposition 2.** $\arg\max_c \left( \hat{\boldsymbol{x_l}} . \hat{\boldsymbol{x_l^c}} \right) = \arg\max_c \left( \boldsymbol{x_l} . \boldsymbol{x_l^c} \right)$

*Proof.* As $\boldsymbol{p_1^c}, \ldots, \cdot\boldsymbol{p_k^c}$ are orthonormal, therefore

$$|\boldsymbol{x_l^c}| = \sqrt{(\boldsymbol{x_l} \cdot \boldsymbol{p_1^c})^2 + \ldots + (\boldsymbol{x_l} \cdot \boldsymbol{p_k^c})^2} \tag{8}$$

After expanding the dot product and noting that $|\boldsymbol{p_1^c}|, \ldots, |\boldsymbol{p_k^c}| = 1$, we have

$$|\boldsymbol{x_l^c}| = |\boldsymbol{x_l}| \sqrt{\cos^2 \theta_1^c + \cdots + \cos^2 \theta_k^c} \tag{9}$$

where $\theta_i^c$ is the angle between $\boldsymbol{x_l}$ and $\boldsymbol{p_i^c}$.

MASC(Ketha & Ramaswamy, 2026) on layer $l$ predicts the label of $\boldsymbol{x}$ as

$$\arg\max_c \left( \hat{\boldsymbol{x_l}} . \hat{\boldsymbol{x_l^c}} \right) = \arg\max_c \left( \frac{(\boldsymbol{x_l} \cdot \boldsymbol{p_1^c})^2 + \ldots + (\boldsymbol{x_l} \cdot \boldsymbol{p_k^c})^2)}{|\boldsymbol{x_l}||\boldsymbol{x_l^c}|} \right) \tag{10}$$

We now substitute 9 in 10

$$\arg\max_c \left( \frac{(\boldsymbol{x_l} \cdot \boldsymbol{p_1^c})^2 + \ldots + (\boldsymbol{x_l} \cdot \boldsymbol{p_k^c})^2)}{|\boldsymbol{x_l}||\boldsymbol{x_l}| \sqrt{\cos^2 \theta_1^c + \cdots + \cos^2 \theta_k^c}} \right) \tag{11}$$

After expanding the numerator and noting that $|\boldsymbol{p_1^c}|, \ldots, |\boldsymbol{p_k^c}| = 1$, we have

$$\arg\max_c \left( \frac{|\boldsymbol{x_l}|^2 \cos^2 \theta_1^c + \cdots + |\boldsymbol{x_l}|^2 \cos^2 \theta_k^c}{|\boldsymbol{x_l}|^2 \sqrt{\cos^2 \theta_1^c + \cdots + \cos^2 \theta_k^c}} \right) \tag{12}$$

---

[20]which is typically estimated via PCA, where $k$ is the number of principal components.

Simplifying, we now have

$$\arg\max_c \left( \hat{\boldsymbol{x_l}}.\hat{\boldsymbol{x_l^c}} \right) = \arg\max_c \left( \sqrt{\cos^2\theta_1^c + \cdots + \cos^2\theta_k^c} \right) \tag{13}$$

Now,

$$\arg\max_c \left( \boldsymbol{x_l}.\boldsymbol{x_l^c} \right) = \arg\max_c \left( (\boldsymbol{x_l} \cdot \boldsymbol{p_1^c})^2 + \ldots + (\boldsymbol{x_l} \cdot \boldsymbol{p_k^c})^2 ) \right) \tag{14}$$

After expanding and noting that $|\boldsymbol{p_1^c}|, \ldots, |\boldsymbol{p_k^c}| = 1$, we have

$$\arg\max_c \left( |\boldsymbol{x_l}|^2 \cos^2\theta_1^c + \cdots + |\boldsymbol{x_l}|^2 \cos^2\theta_k^c \right) \tag{15}$$

$|\boldsymbol{x_l}|^2$ is taken common and is not dependent on $c$, due to which, equivalently, we have

$$\arg\max_c \left( \boldsymbol{x_l}.\boldsymbol{x_l^c} \right) = \arg\max_c \left( (\cos^2\theta_1^c + \cdots + \cos^2\theta_k^c ) \right) \tag{16}$$

From 13 and 16, it follows that

$$\arg\max_c \left( \hat{\boldsymbol{x_l}}.\hat{\boldsymbol{x_l^c}} \right) = \arg\max_c \left( \boldsymbol{x_l}.\boldsymbol{x_l^c} \right) \tag{17}$$

$\square$

## C  Training dynamics of latent generalization using MASC

MASC testing accuracy during training with 0% and 100% corruption degrees are shown in Figure 10.

## D  Training dynamics of the linear probe: VeLPIC

A linear probe – VeLPIC – test accuracy during training for all models with 0% and 100% corruption degrees are shown in Figure 11.

### D.1  Difference between VeLPIC and MASC

Here, we present the difference between test accuracy of VeLPIC and MASC during training and for different layer of the networks. For MLP-MNIST,MLP-CIFAR-10, CNN-MNIST, CNN-Fashion-MNIST, CNN-CIFAR-10, AlexNet-Tiny ImageNet and ResNet-18-CIFAR-10, these results for 0% and 100% corruption degrees are shown in Figure 12.

## E  Additional results with linear probe (logistic regression)

For all models with 0% corruption degree, the test accuracy of the logistic regression probe during training of the memorized models, are shown in Figure 13. The results comparing the performance of logistic regression probe with MASC probe is shown in Figures 14 and with VeLPIC probe is shown in Figures 15.

## F  Transferring latent generalization to model generalization

For all models with 0% and 100% corruption degrees, model test accuracy during training when we replace the pre-softmax weights with VeLPIC vectors are shown in Figure 16. Model corrupted training accuracy for a few models-dataset-corruption are plotted in Figure 17 and Figure 18.

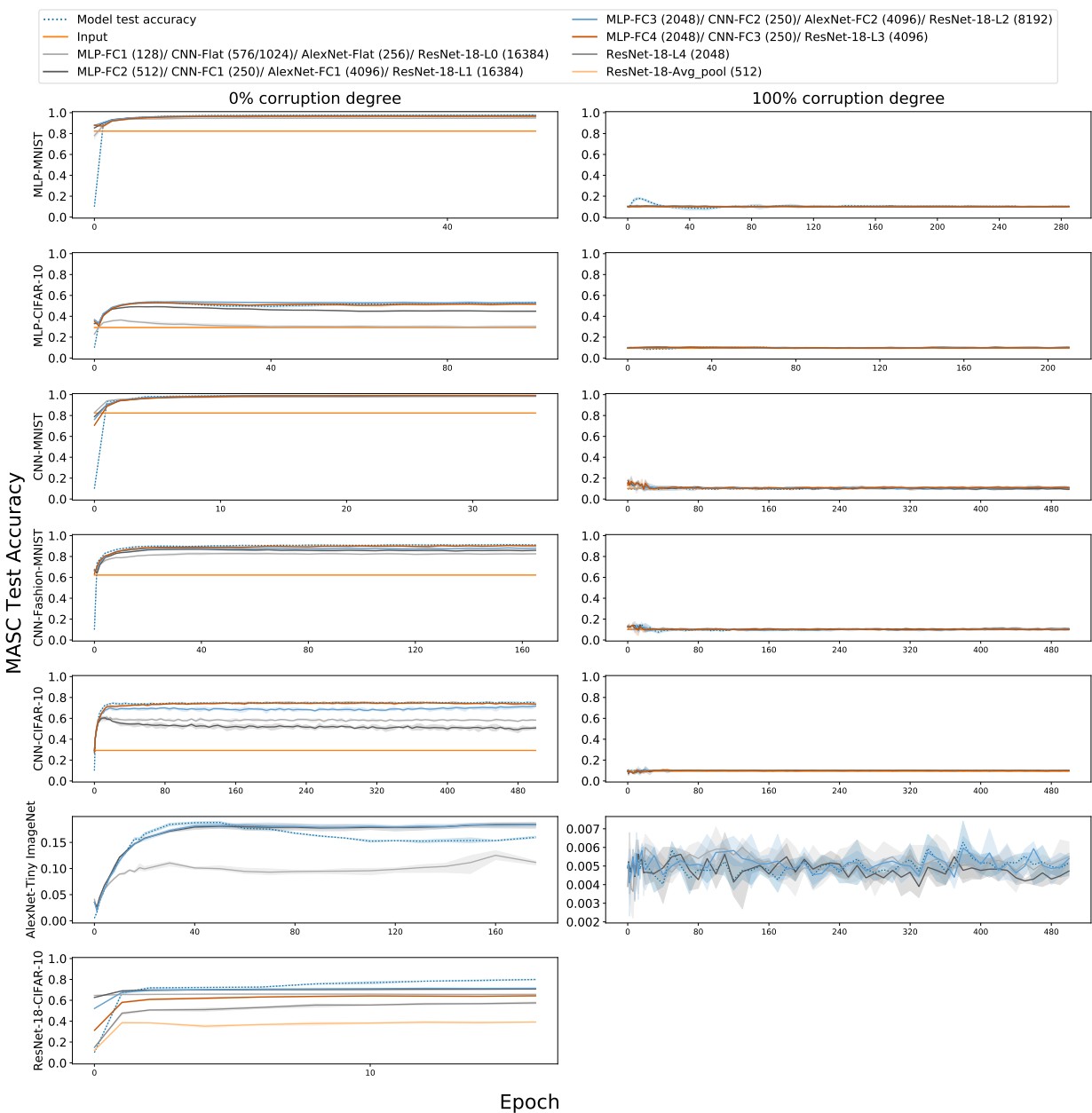

Figure 10: Minimum Angle Subspace Classifier (MASC) Test accuracy for 0% and 100% corruption degrees during training of the network, where test data is projected onto class-specific subspaces constructed from training data with the indicated label corruption degrees. The plots display MASC accuracy across different layers of the network for various model–dataset combinations. For reference, the test accuracy of the models (dotted line) is also shown. Each row corresponds to a specific corruption degree, while columns represent different models, as labeled. FC denotes fully connected layers with *ReLU* activation, and Flat refers to the flatten layer without *ReLU*.

## G   Experiment results with weight intervention

Here, we present results of the latent generalization of VeLPIC evolves across different layers during training for models trained with 0% corruption degree. The results for intervention performed at the 40th epoch is shown in Figure 19.

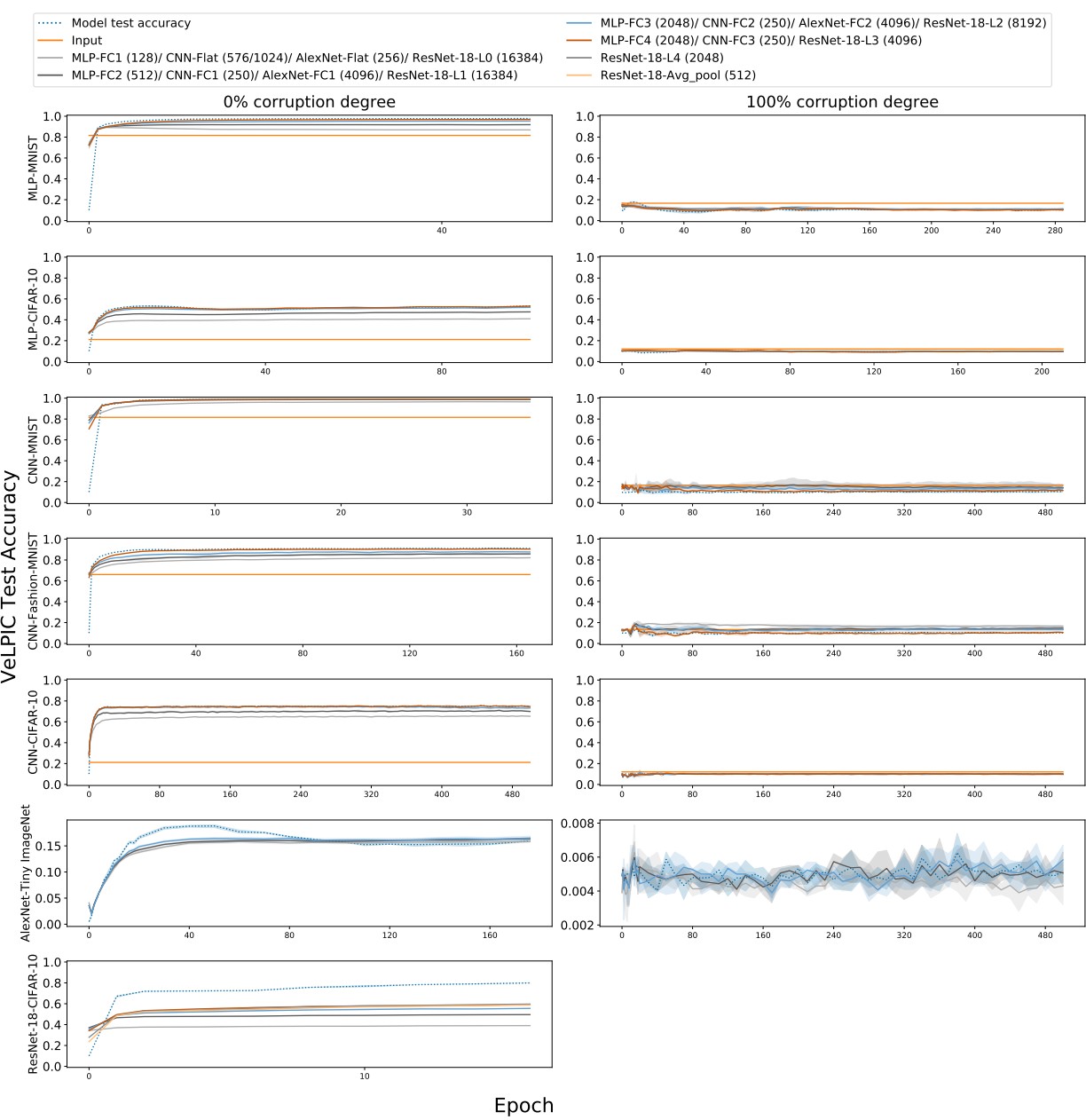

Figure 11: Vector Linear Probe Intermediate-layer Classifier (VeLPIC) test accuracy for 0% and 100% corruption degrees during training of the network, where test data is projected onto class vectors constructed at each epoch from training data with the indicated label corruption degrees. The plots display VeLPIC accuracy across different layers of the network for various model–dataset combinations. For reference, the test accuracy of the models (blue dotted line) over epochs of training is also shown. FC denotes fully connected layers with *ReLU* activation, and Flat refers to the flatten layer without *ReLU*.

## H    Comparison between randomly initialized and trained model

We compare the test accuracy of the model, MASC, and VeLPIC for both randomly initialized and trained models across varying corruption degrees. Results for all model–dataset pairs are presented in Figure 20. For MASC and VeLPIC, we report accuracies corresponding to the best-performing layer, defined as the layer achieving the highest MASC or VeLPIC test accuracy for each model–dataset pair.

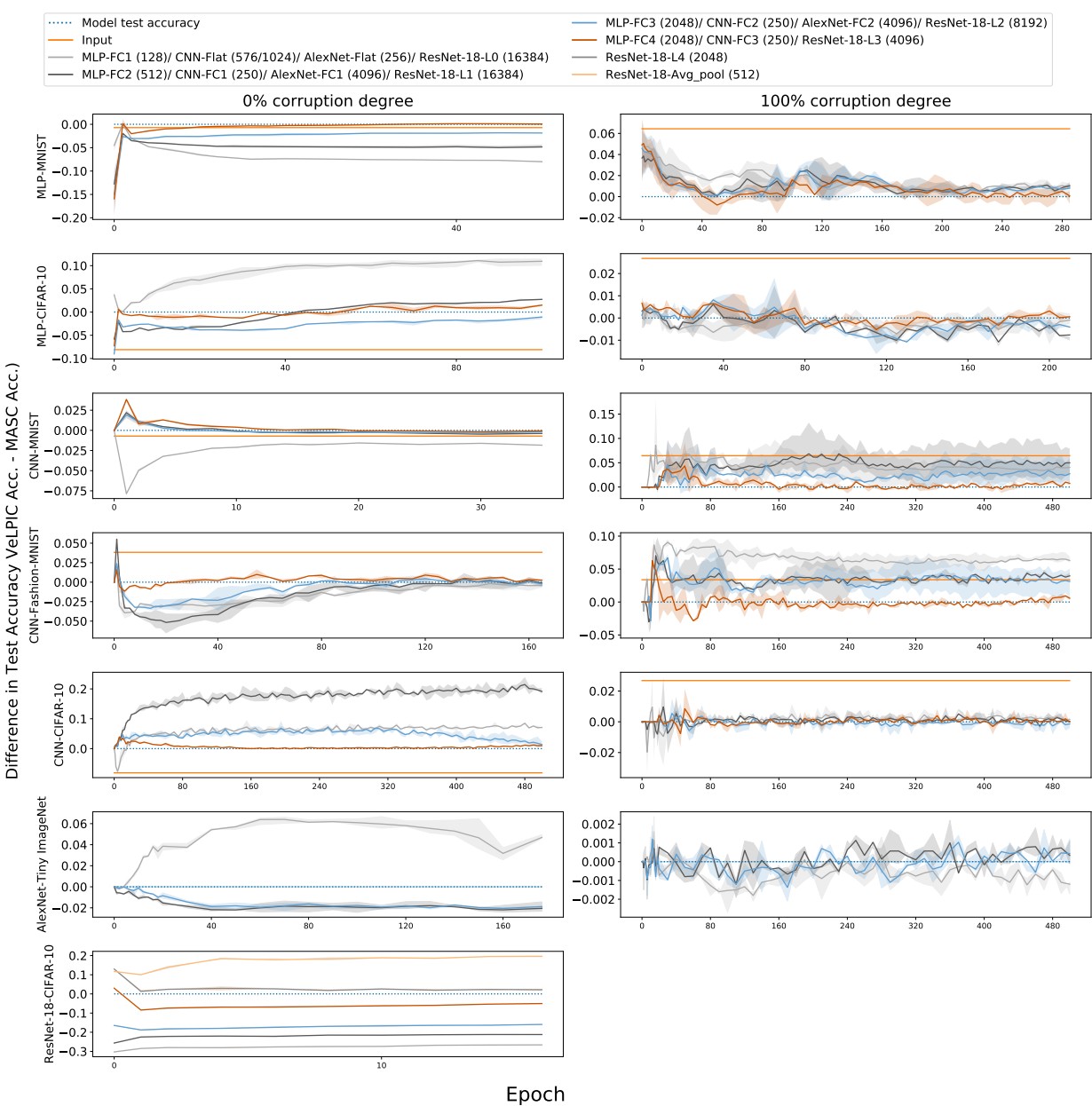

Figure 12: Difference in test accuracy (VeLPIC Accuracy - MASC Accuracy) during training of the network, where test data is projected onto class vectors constructed at each epoch from training data with the indicated label corruption degrees. The plots display difference in accuracy across different layers of the network for various model–dataset combinations. For reference, the test accuracy of the models (blue dotted line) over epochs of training is also shown, which would be 0.

At lower corruption degrees, we observe a consistent trend across all model–dataset combinations: MASC and VeLPIC evaluated on trained models outperform their counterparts evaluated on randomly initialized models when measured at the respective best layers.

At higher corruption degrees, however, a small number of exceptions emerge. Specifically, for MASC, performance on randomly initialized models exceeds that on trained models in the following cases: AlexNet-Tiny ImageNet at 60% corruption, and MLP-MNIST, MLP-CIFAR-10, and AlexNet-Tiny ImageNet at 80%

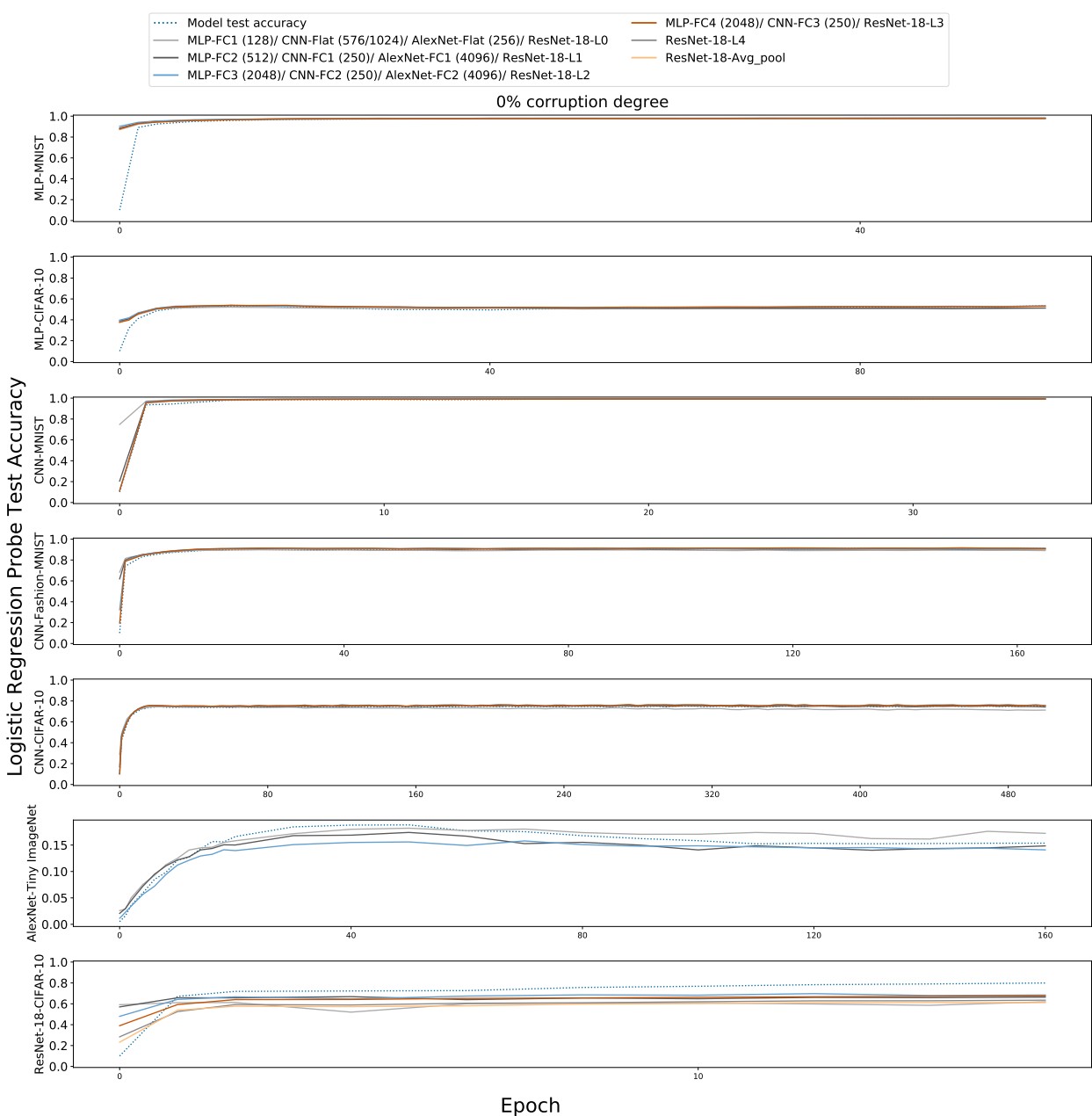

Figure 13: Logistic regression probe's test accuracy over epochs of training for multiple models/datasets. The plots display logistic regression probe's accuracy across different layers of the network. For reference, the evolution of test accuracy of the corresponding model (blue dotted line) over epochs of training is also shown.

corruption. In contrast, VeLPIC exhibits a single exception. At 80% corruption on AlexNet-Tiny ImageNet, VeLPIC evaluated on the randomly initialized model outperforms the trained model.

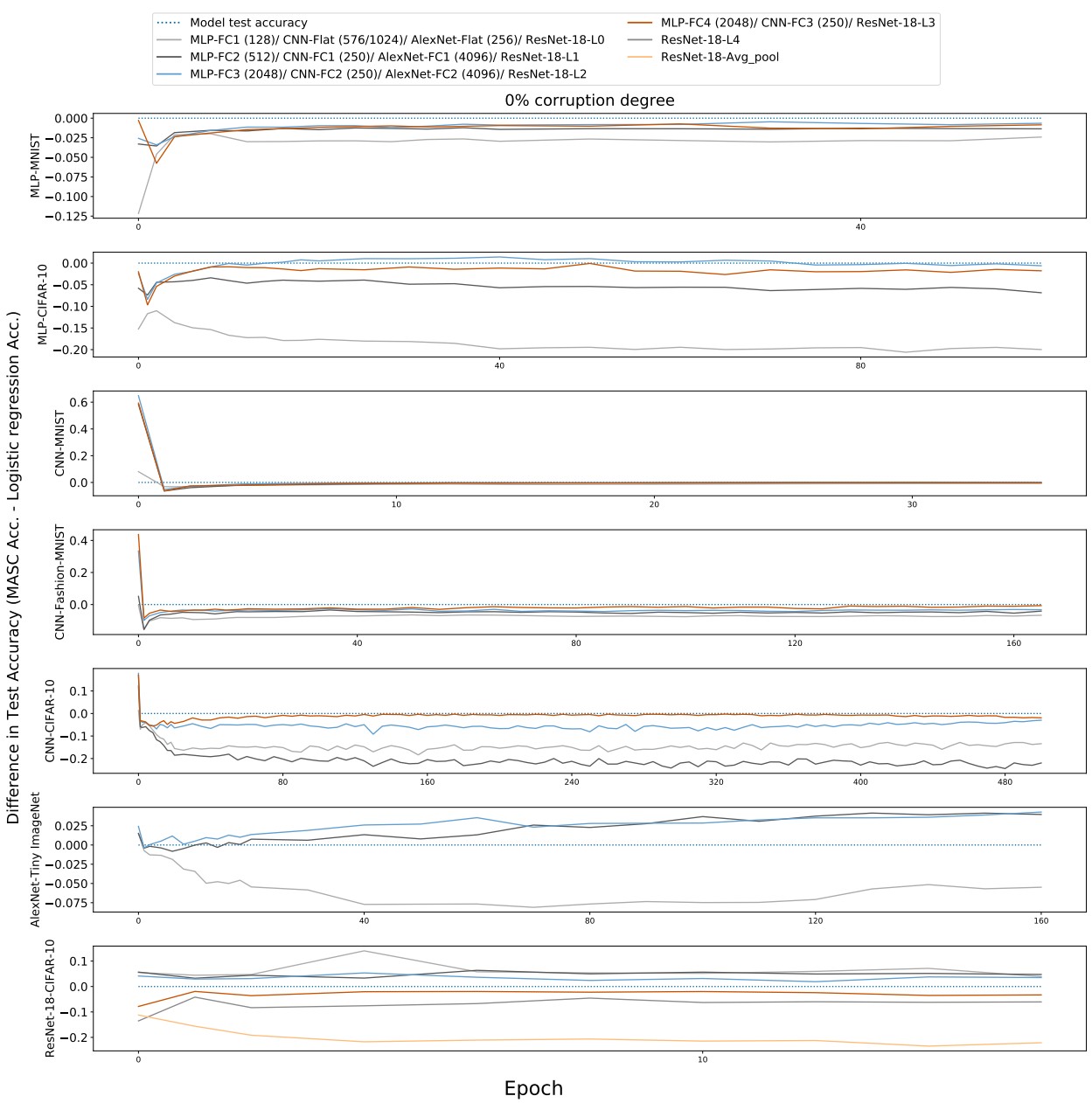

Figure 14: Difference in test accuracy (MASC Accuracy - Logistic regression probe Accuracy) during training of the network, where for MASC test data is projected onto class vectors constructed at each epoch from training data with the indicated label corruption degrees. The plots display difference in accuracy across different layers of the network for various model–dataset combinations. For reference, the test accuracy of the models (blue dotted line) over epochs of training is also shown, which would be 0.

# I    Impact of Dropout as a Regularizer

We conducted some preliminary experiments to study the effect of dropout (Srivastava et al., 2014) as a regularization technique on latent generalization during memorization. Both the CNN and ResNet-18 models were slightly modified to incorporate dropout layers. For dropout version of CNN models, a dropout layer (p=0.2) was used after every fully connected layer. The CNN models were trained on MNIST, Fashion-

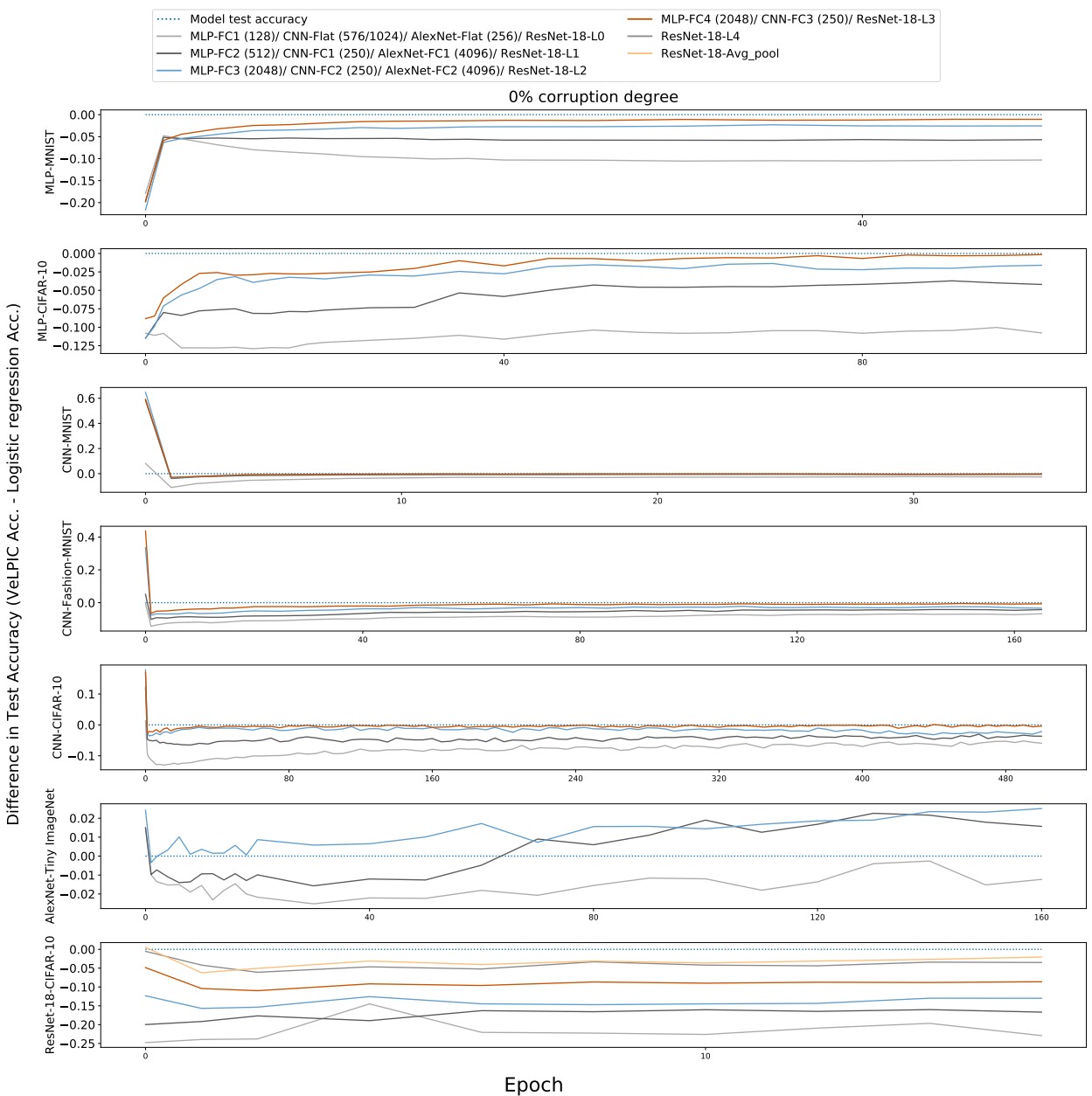

Figure 15: Difference in test accuracy (VELPIC Accuracy - Logistic regression probe Accuracy) during training of the network, where for VELPIC test data is projected onto class vectors constructed at each epoch from training data with the indicated label corruption degrees. The plots display difference in accuracy across different layers of the network for various model–dataset combinations. For reference, the test accuracy of the models (blue dotted line) over epochs of training is also shown, which would be 0.

MNIST, and CIFAR-10. For dropout version of ResNet-18, dropout layer (p=0.2) was added before the final classification layer.

MASC test accuracy on models trained with and without dropout layers are shown in Figure 21 and 22. For models trained with dropout (WD), we plot the results after the dropout layer. For models trained with dropout, the MASC dynamics remain largely similar to those of models trained without dropout. However,

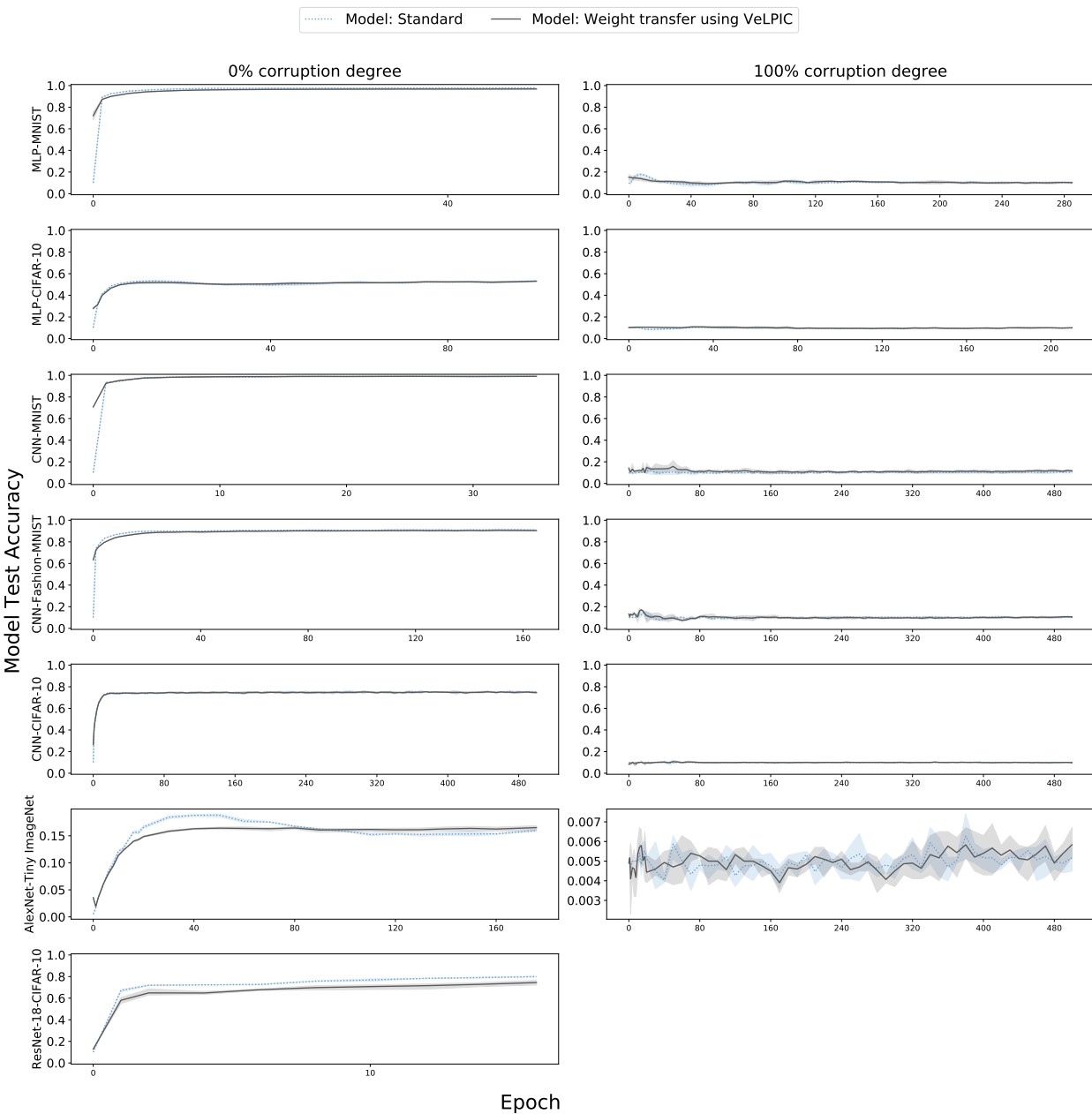

Figure 16: Comparing model test accuracy with VeLPIC transferred accuracy when the weight intervention is applied to the model at the epoch in question during training for corruption degrees 0% and 100%. The test accuracy of the model with standard training without weight intervention (blue dotted line) is overlaid for comparison.

in the later layers of CNN-WD-Fashion-MNIST and CNN-WD-MNIST, we observe a noticeable deviation, where the layer-wise behavior no longer aligns with the model's test accuracy in the early epochs of training; interestingly the corresponding VeLPIC plots in Figure 23 do not show this deviation. In most cases, MASC performance shows an initial decline followed by a rise toward a peak. In contrast, for ResNet-18 models, no significant differences are observed between the dropout and non-dropout models.

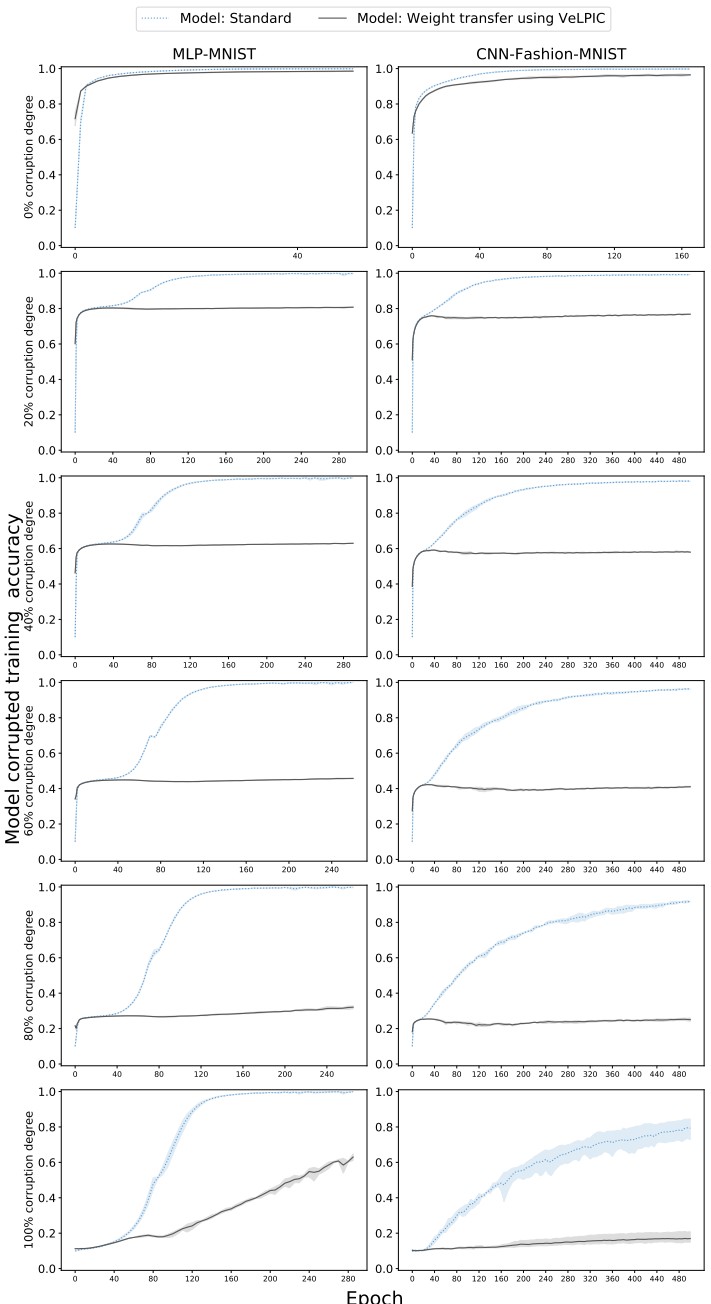

Figure 17: Model train accuracy on corrupted dataset when the VeLPIC weight intervention is applied to pre-softmax weights at the epoch in question during training. The training accuracy on corrupted dataset of the model with standard training without weight intervention (blue dotted line) is overlaid for comparison. Observe that, except for 100% corruption degree, the transferred training accuracy tends to saturate at a level largely consistent with the fraction of true training labels in the corrupted dataset.

VeLPIC test accuracy on models trained with and without dropout layers are shown in Figure 23 and 24. There is no significant difference in the dynamics of VeLPIC performance between models trained with dropout and those trained without dropout.

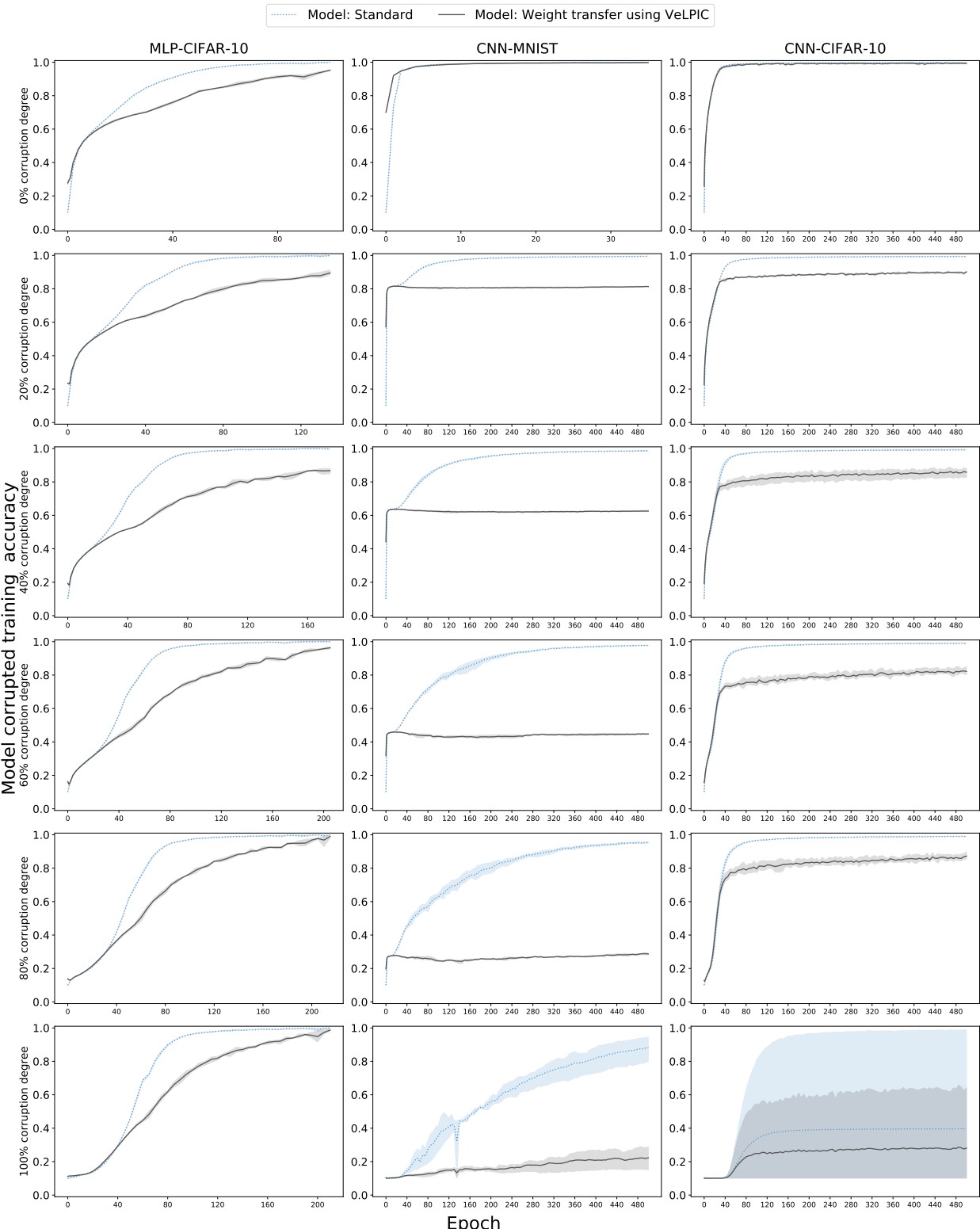

Figure 18: Model train accuracy on corrupted dataset when the VeLPIC weight intervention is applied to pre-softmax weights at the epoch in question during training. The training accuracy on corrupted dataset of the model with standard training without weight intervention (blue dotted line) is overlaid for comparison.

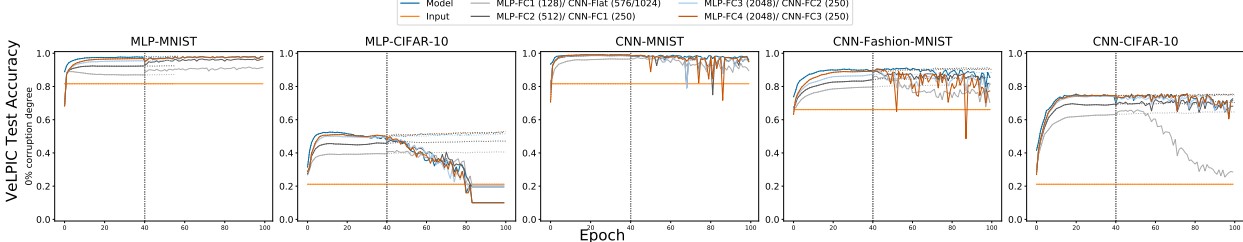

Figure 19: Vector Linear Probe Intermediate layer Classifier (VeLPIC) test accuracy during training on models when intervention is performed at 40th epoch and standard training is performed thereafter for 60 epochs for 0.0 corruption degree. The VeLPIC test accuracy on models trained with standard training (dotted) without intervention is overlaid for comparison. The results correspond to a single run in each case.

Figure 20: Model, MASC and VeLPIC test accuracy for randomly initialized and trained models across different corruption degrees and model–dataset pairs. For MASC and VeLPIC, results are reported for the best performing layer, defined as the layer achieving the highest MASC/VeLPIC test accuracy for the corresponding model–dataset pair. The test accuracies are averaged over three runs.

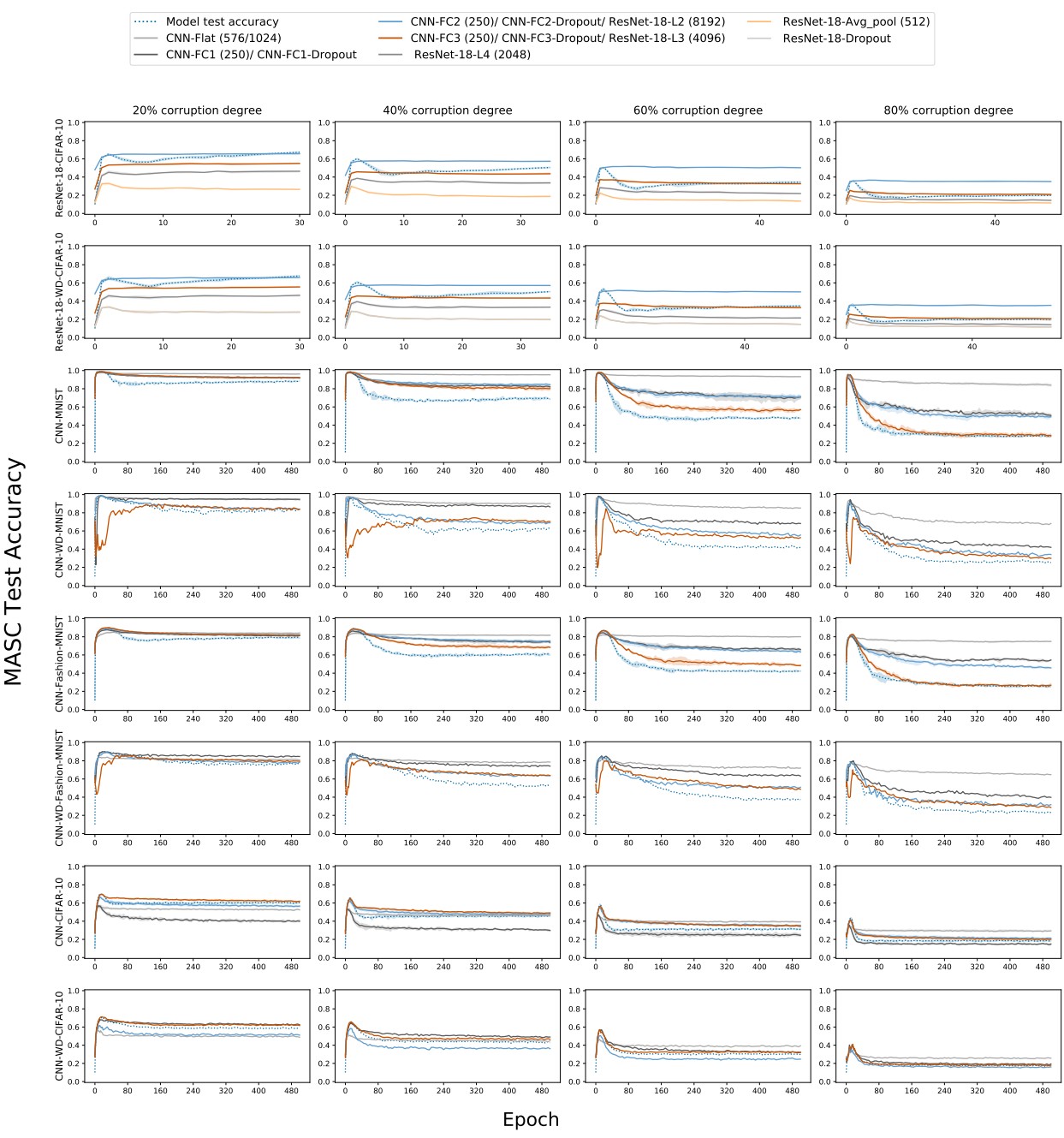

Figure 21: Minimum Angle Subspace Classifier (MASC) test accuracy over epochs of training for ResNet-18 and CNN models having trained with and with dropout, where test data is projected onto class-specific subspaces constructed at each epoch from corrupted training data with the indicated label corruption degree. The plots display MASC accuracy across different layers of the network. For reference, the evolution of test accuracy of the corresponding model (blue dotted line) over epochs of training is also shown. WD corresponding to models trained with dropout.

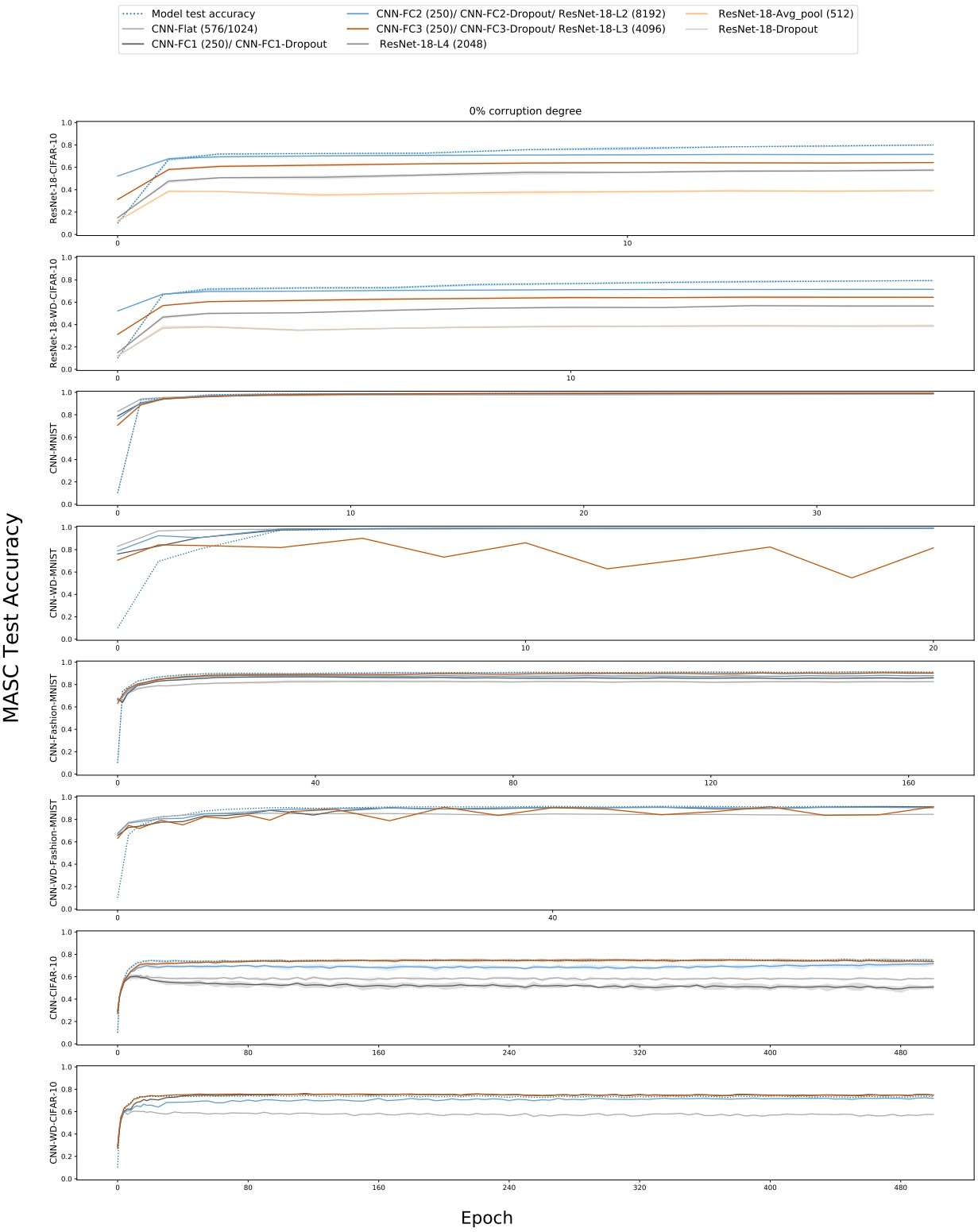

Figure 22: Minimum Angle Subspace Classifier (MASC) test accuracy for 0% corruption degrees over epochs of training for ResNet-18 and CNN models having with and with dropout, where test data is projected onto class-specific subspaces constructed at each epoch from corrupted training data. The plots display MASC accuracy across different layers of the network. For reference, the evolution of test accuracy of the corresponding model (blue dotted line) over epochs of training is also shown. WD corresponding to models trained with dropout.

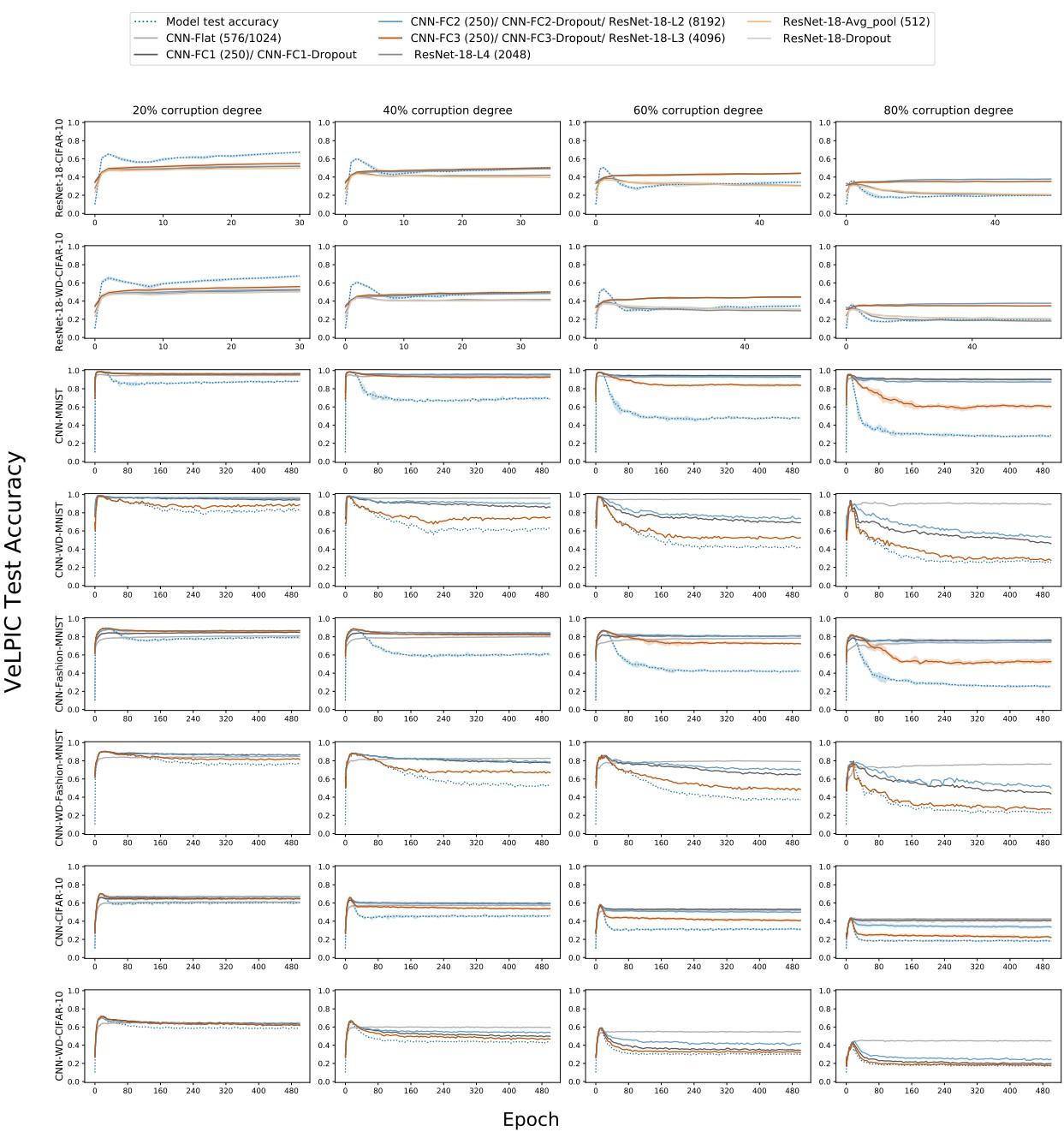

Figure 23: Vector Linear Probe Intermediate-layer Classifier (VeLPIC) test accuracy during training of ResNet-18 and CNN models with and with dropout, where test data is projected onto class vectors constructed at each epoch from training data with the indicated label corruption degrees. The plots display VeLPIC accuracy across different layers of the network for various model–dataset combinations. For reference, the test accuracy of the models (blue dotted line) over epochs of training is also shown. WD corresponding to models trained with dropout.

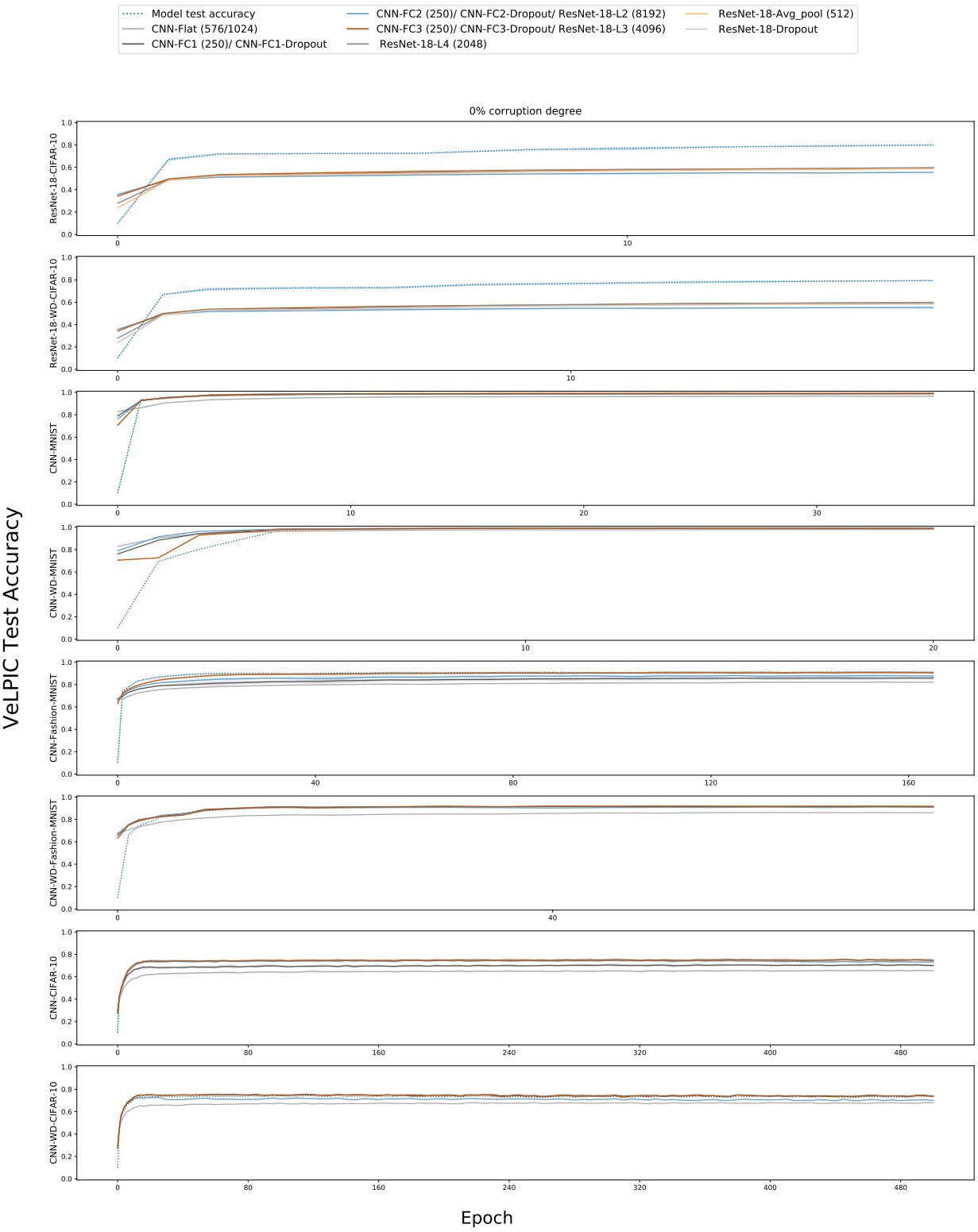

Figure 24: Vector Linear Probe Intermediate-layer Classifier (VeLPIC) test accuracy for 0% corruption degrees during training of ResNet-18 and CNN models with and with dropout, where test data is projected onto class vectors constructed at each epoch from training data. The plots display VeLPIC accuracy across different layers of the network for various model–dataset combinations. For reference, the test accuracy of the models (blue dotted line) over epochs of training is also shown. WD corresponding to models trained with dropout.

