# OpenReview forum: "On the Dynamics & Transferability of Latent Generalization during Memorization"
_TMLR — Accepted by TMLR_

### Review · Reviewer_StSB · 2026-01-06

**Summary Of Contributions:**

The paper extends the results of Ketha & Ramaswamy (2025) in two directions. First, the paper considers the dynamics of probe accuracy throughout training rather than focusing on the final weights. Second, the authors propose new, more accurate VeLPIC probes instead of MASC probes.

**Strengths**
1. The work investigates an important memorization phenomenon, relevant for understanding deep learning mechanisms.
2. The paper offers a clear improvement of Ketha & Ramaswamy's (2025) methodology.

**Weaknesses**
1. The paper mostly offers technical improvements and extensions without providing deeper insights.

**Audience:**

Yes

**Audience Explanation:**

The research question seems important. However, the significance of contributions is modest.

**Broader Impact Concerns:**

I do not feel a need for a Broader Impact Statement.

**Claims And Evidence:**

No

**Claims Explanation:**

1. I am not sure why the analysis of dynamics suggests that model generalization and latent generalization use largely the same underlying mechanisms. From what we can see, the dynamics is indeed similar at the beginning of training. However, at the end of training, the results diverge, suggesting different mechanisms for these phenomena.
2. I am not sure what authors mean by a quadratic classifier and why this distinction is important. MASC projection could easily be rewritten in a matrix form without the tensor product of PCA vectors.
3. I do not think that the inclusion of negative datapoints is a theoretically grounded way to perform PCA. In particular, in our case, this augmentation will arguably inflate a direction in which the class-conditional vector space is shifted from the origin.

**Requested Changes:**

1. Please explain in more detail the conclusions of the experiments and their limitations.
2. Please clarify what a quadratic classifier means in the context of this paper.
3. Please explain the goal of applying PCA in this setting.
4. I think the paper would benefit from a discussion about why different layers produce different results. Generally, the discussion in Section 4 seems rather short.
5. (minor) I think the paper would benefit from the inclusion of more modern vision architectures (e.g., ResNets) for FashionMNIST and CIFAR-10 experiments.

---

> ### Author Response · Authors · 2026-02-07
>
> We thank the reviewer for their careful reading and constructive review.
>
>
> We address some of the comments individually below.
>
> > 1. I am not sure why the analysis of dynamics suggests that model generalization and latent generalization use largely the same underlying mechanisms. From what we can see, the dynamics is indeed similar at the beginning of training. However, at the end of training, the results diverge, suggesting different mechanisms for these phenomena.
>
> Thank you for this point. We have rewritten some of the exposition to make it clear that similar mechanisms could be at play at the beginning of training but not subsequently. Addition of new experiments on ResNt-18 trained on CIFAR-10 add additional nuance to this view. We have accordingly modified our exposition to make this clear.
>
>
>
> > 2. I am not sure what authors mean by a quadratic classifier and why this distinction is important. MASC projection could easily be rewritten in a matrix form without the tensor product of PCA vectors.
>
> Thank you. While MASC can be written as a quadratic form in matrix notation, it is a quadratic classifier in the sense of having quadratic decision surfaces. The fact that it is a non-linear classifier could suggest that the specific nature of its nonlinearity might be crucial in its ability to extract latent generalization. We have expanded some of the exposition in the main text to make these points clearer.
>
> > 3. I do not think that the inclusion of negative datapoints is a theoretically grounded way to perform PCA. In particular, in our case, this augmentation will arguably inflate a direction in which the class-conditional vector space is shifted from the origin.
>
>
> The goal was to fit subspaces and indeed the inclusion of negative datapoints accomplishes that goal. Indeed, our doing so follows the approach from [Ketha & Ramaswamy, 2026]. Are there other alternative approaches of accomplishing this goal that the reviewer has in mind?
>
>
> >  1. Please explain in more detail the conclusions of the experiments and their limitations.
>
> We have added additional exposition to the sections that more clearly lay out the conclusions. We have also added exposition that describes limitations of our work in the Discussion section.
>
> > 2. Please clarify what a quadratic classifier means in the context of this paper.
>
> Thank you for this question. A quadratic classifier is one which has a quadratic decision surface. We have also included this point in Section 5.
>
>
> > 3. Please explain the goal of applying PCA in this setting.
>
> Broadly, the goal is to fit subspaces. We follow the broad approach from [Ketha and Ramaswamy, 2026] to this end.
>
>
> > 4. I think the paper would benefit from a discussion about why different layers produce different results. Generally, the discussion in Section 4 seems rather short.
>
> Thank you for this point. We have added an expanded discussion in Section 4 on the potential reasons for why different layers might produce different test accuracies with MASC. We have also significantly expanded the discussion in Section 4 to discuss in detail multiple aspects of our results.
>
>
> >  5. (minor) I think the paper would benefit from the inclusion of more modern vision architectures (e.g., ResNets) for FashionMNIST and CIFAR-10 experiments.
>
> Thank you for this suggestion. We have added experiments using a ResNet-18 model trained on CIFAR-10.  These results appear in Figures 1, 2, 3, and 7 of the main text and Figures 10, 11, 12 and 16 in the Appendix.
>
>
> In closing, we are thankful for your very careful and detailed comments. We have done our best in addressing many of the concerns raised, especially in the limited timeframe that was available. We would appreciate the opportunity to engage further with you during the discussion period.

---

> ### Comment · Reviewer_StSB · 2026-02-09
>
> Thank you for the clarifications. The authors addressed some of my points. However, I still have concerns about the paper's results.
>
> 1. While Section 5 argues that MASC has a quadratic decision boundary, it does not use the proper version of MASC. Ketha & Ramaswamy (2026) use cosine similarity and not the dot product for classification.
> 2. The results of Section 7, which compare MASC with a linear probe, contradict the conclusion of Section 5 of Ketha & Ramaswamy (2026) (especially for the first layer of AlexNet). Please explain the reasons for this divergence.
> 3. My point about PCA was partially about the motivation for the method: where it is expected to work well, maybe what underlying optimization problem it is trying to solve, etc. Even if the motivation is a direct comparison with MASC, the paper would benefit from a more in-depth comparison of MASC and VeLPIC's limitations.

---

> ### Author Response · Authors · 2026-02-11
>
> Thank you for the questions. We respond below.
>
> > 1. While Section 5 argues that MASC has a quadratic decision boundary, it does not use the proper version of MASC. Ketha & Ramaswamy (2026) use cosine similarity and not the dot product for classification.
>
> Thank you for bringing this up; you make a fair point, and this comes from a shortcoming in our exposition, which we have now fixed in a revision just uploaded. In practice, PCA gives us an orthonormal basis, which we should have mentioned, and therefore using the dot product is equivalent to using cosine similarity, as in [Ketha & Ramaswamy, 2026]. We have accordingly stated $p^c_1, \\ldots, p^c_k$ is an orthonormal basis and noted in a footnote that this is equivalent to the formulation in [Ketha & Ramaswamy, 2026] in the new revision. Thanks again for noticing this and pointing it out.
>
> > 2. The results of Section 7, which compare MASC with a linear probe, contradict the conclusion of Section 5 of Ketha & Ramaswamy (2026) (especially for the first layer of AlexNet). Please explain the reasons for this divergence.
>
>
> Thank you for this point. We carefully looked over the results for the earliest layer measured (AlexNet-Flat (256)) on AlexNet-Tiny ImageNet and its comparison with the LR probe. Our results are by-and-large consistent with what has been reported in [Ketha & Ramaswamy, 2026]. We just wanted to be sure that you are not inadvertently comparing the AlexNet-Tiny ImageNet results we report here with the AlexNet-CIFAR-100 results that [Ketha & Ramaswamy, 2026] report in Figure 5; in particular they have run AlexNet in that Figure on CIFAR-100 and not Tiny ImageNet, as we do. They also report comparisons of AlexNet-Tiny ImageNet  with the LR probe in Figure 10 of the Appendix of their paper. We checked and our results are by-and-large consistent. If we are missing your point, we would appreciate a pointer to the specific model/layer/corruption degree that you have in mind.
>
>
> > 3 My point about PCA was partially about the motivation for the method: where it is expected to work well, maybe what underlying optimization problem it is trying to solve, etc. Even if the motivation is a direct comparison with MASC, the paper would benefit from a more in-depth comparison of MASC and VeLPIC's limitations.
>
>
> Thank you for this. Ketha and Ramaswamy remark in a review response in TMLR that their motivation was from the Manifold Hypothesis in machine learning. Specifically, since fitting manifolds can be computationally expensive, their intent was to fit subspaces instead. We reproduce verbatim their remark below, for convenience.
> > "In building MASC, we were motivated by the manifold hypothesis in machine learning Goodfellow et al. (2016) that posits that high-dimensional data typically reside on a low-dimensional manifold. It has also been suggested (Brahma et al., 2015) that such manifolds in layerwise representations flatten across layers of deep networks. Fitting manifolds can be computationally expensive, so we were interested in examining the organization of classwise data in subspaces, even if such subspaces might be somewhat higher dimensional than the corresponding manifolds. Indeed, this view leads to the natural idea of classifying unseen data points by determining which class manifold it is closest to. MASC is simply a formalization of this idea. In particular, this classifier lends strong geometric motivation and intuition, in contrast to e.g. training a standard linear probe that iteratively minimizes a crossentropy loss. However, the reasons for the success of MASC in this setting are still largely unclear to us. The difficulty is that the principles that underlie the nature of layerwise representations in deep networks trained with standard techniques are not well understood at this time and it appears that such representations play a significant role in the success of MASC in the memorization setting. Indeed, it is even a bit surprising that the deep network does not directly leverage this structure to obtain better generalization to true labels, although that may also be because its loss function aims to maximize training accuracy which might run counter to the act of bettering generalization to true labels."
>
>
> As for MASC and VeLPIC’s limitations, a merit of MASC seems to be its ability to capture generalization when it is present in a subspace of dimension >1. This, notably, appears to be the case with the early layers of ResNet-18, where MASC generalizes significantly better than VeLPIC, as well as the model. With VeLPIC on the other hand, we have a technique to directly transfer VeLPIC’s generalization on the last layer to the model, whereas we don’t yet have a way of doing so for MASC. We have briefly included these points, in the Discussion section, while discussing limitations. We thank the reviewer for bringing this up.
>
> Thanks again for engaging. We are happy to discuss further.

---

> > ### Comment · Reviewer_StSB · 2026-02-11
> >
> > Thank you for the clarification!
> >
> > Regarding Section 5, the remark states that the two formulations are equivalent because they are normalized by $|x_l|^2$. However, the cosine similarity is not normalized by this quantity; it is normalized by $|x_l| |x_l^c|$. Since $|x_l^c|$ is different among classes, it is hard to see why the boundary is quadratic.

---

> > > ### Author Response · Authors · 2026-02-14
> > >
> > > > Regarding Section 5, the remark states that the two formulations are equivalent because they are normalized by $|x_l|^2$. However, the cosine similarity is not normalized by this quantity; it is normalized by $|x_l| |x_l^c|$. Since $|x_l^c|$ is different among classes, it is hard to see why the boundary is quadratic.
> > >
> > > Thank you for this point.
> > >
> > > We have now added a proof that establishes this equivalence in Section B.1 of the Appendix, showing how the two formulations are equivalent. We have also added a pointer from the proof of Proposition 1 in Section 5 to this proof. We believe this improves clarity and are thankful that you brought it up.

---

### Review · Reviewer_YWzM · 2026-01-16

**Summary Of Contributions:**

This paper examines the phenomenon of generalization in deep neural networks with corrupted training labels. It has been observed that DNNs trained with corrupted labels can achieve perfect training accuracy (necessarily memorizing the spurious labels), at the cost of generalization performance. However, even in this regime the model's internal representations may still contain useful, generalizable structure, which is measured by fitting a simpler predictor (a probe) to hidden representations of the trained network. The contribution of this paper follows in the tracks of recent work based around a subspace-based yet nonlinear probe to propose a linear probe, which is claimed to achieve similar performance.

**Audience:**

No

**Audience Explanation:**

The phenomenon that this paper investigates is interesting and important to understanding how deep learning works. However, I'm skeptical that its findings in the current state have much to offer, since the loose approach they explore is not new (see e.g. Li et al 2020, "Gradient descent with early stopping is provably robust to label noise for overparameterized neural networks" which examined the behavior of the classwise principle components under corruption) and it is unclear whether the findings will generalize. The clearest contribution is a conceptual improvement on MASC (Ketha & Ramaswamy 2025), so those who were aware of that work may be interested in this one, but given that was a pre-preprint with a single citation by the same authors I am not convinced that demographic is very substantial.

**Claims And Evidence:**

No

**Claims Explanation:**

There are two main points of the paper's contribution which demand experimental evidence: (i) that VeLPIC is an effective method of obtaining a functional classifier from a deep neural network trained with corrupted labels; and (ii) that VeLPIC is competitive with MASC (Ketha & Ramaswamy 2025). The most convincing evidence for both of these points is relegated to the loosely-structured appendices. To point (i), the issue here is that the story is relatively clear for the easy benchmarks (MNIST and Fashion-MNIST) but there seems to be deterioration for more difficult datasets (Tiny ImageNet and CIFAR10), where even the uncorrupted models are not achieving good generalization. To argue convincingly that the performance of VeLPIC itself generalizes to more complex tasks, experiments on larger models and datasets are necessary -- the largest network examined here is AlexNet. To point (ii), the main body contains no numerical comparison with MASC at all (although the material in the appendix does convincingly demonstrate that VeLPIC is comparable or better than MASC on the evaluations examined here).

**Requested Changes:**

* The abstract reads more like an introduction and should be trimmed down by about half to better focus on the paper's contribution.

* As I understand it, VeLPIC is needlessly convoluted by the augmentation step. Adding the negatives of a set of vectors only changes the empirical covariance matrix by an overall scalar and so does not affect the PCs or their ordering. It would then appear that VeLPIC is just linear classification with classwise PCs as class vectors, which is an old idea and should be properly referenced (e.g. Das et al, "A Classwise PCA-based Recognition of Neural Data for Brain-Computer Interfaces", IEEE 2009).

* The experiments that directly compare MASC and VeLPIC should be included in the main body.

* Toy datasets like MNIST and Fashion-MNIST should not be prioritized in the results since the results there do not seem to generalize to the more complex datasets. At minimum I recommend moving the CIFAR10 experiments to the main body.

* High-accuracy generalization is not being obtained in the 0%-noise setting for the harder datasets (Tiny ImageNet and CIFAR10). I recommend modifying models and training protocols as necessary to improve baseline generalization and then re-run the corruption experiments in that setting.

* Comparison with the baseline of VeLPIC or MASC applied to representations drawn from a randomly initialized network would also strengthen the argument that it is some generalization property of the representations rather than the strength of the probes.

---

> ### Author Response · Authors · 2026-02-07
> **Part-1 of author response**
>
> We thank the reviewer for their careful reading and constructive review.
>
>
> We address some of the comments individually below.
>
> > There are two main points of the paper's contribution which demand experimental evidence: (i) that VeLPIC is an effective method of obtaining a functional classifier from a deep neural network trained with corrupted labels; and (ii) that VeLPIC is competitive with MASC (Ketha & Ramaswamy 2025). The most convincing evidence for both of these points is relegated to the loosely-structured appendices. To point (i), the issue here is that the story is relatively clear for the easy benchmarks (MNIST and Fashion-MNIST) but there seems to be deterioration for more difficult datasets (Tiny ImageNet and CIFAR10), where even the uncorrupted models are not achieving good generalization. To argue convincingly that the performance of VeLPIC itself generalizes to more complex tasks, experiments on larger models and datasets are necessary -- the largest network examined here is AlexNet. To point (ii), the main body contains no numerical comparison with MASC at all (although the material in the appendix does convincingly demonstrate that VeLPIC is comparable or better than MASC on the evaluations examined here).
>
> Thank you for these points. We have included results of new experiments on ResNet-18 trained on CIFAR-10. In response to your suggestions, we have moved multiple figures from the Appendix to the main text, including figures that present a numerical comparison, as suggested.
>
> > The phenomenon that this paper investigates is interesting and important to understanding how deep learning works. However, I'm skeptical that its findings in the current state have much to offer, since the loose approach they explore is not new (see e.g. Li et al 2020, "Gradient descent with early stopping is provably robust to label noise for overparameterized neural networks" which examined the behavior of the classwise principle components under corruption) and it is unclear whether the findings will generalize. The clearest contribution is a conceptual improvement on MASC (Ketha & Ramaswamy 2025), so those who were aware of that work may be interested in this one, but given that was a pre-preprint with a single citation by the same authors I am not convinced that demographic is very substantial.
>
> Thank you. We remark that Li et al, 2020, do not study the effect of using probes on models whose generalization is beyond the early stopping point, and therefore do not agree that our results are anticipated by that work. We have now included results from new experiments on ResNet-18 trained on CIFAR-10 that add further nuance to our findings.
> While at the time of initial submission, (Ketha & Ramaswamy 2025) was a preprint, we note that it has now been published in TMLR, which suggests that some readers of TMLR find this topic of interest.
>
> > The abstract reads more like an introduction and should be trimmed down by about half to better focus on the paper's contribution.
>
> Thank you for the comment. We have made substantial efforts to shorten and refocus the abstract, reducing its length while emphasizing the main contributions of the work.
>
>
> > As I understand it, VeLPIC is needlessly convoluted by the augmentation step. Adding the negatives of a set of vectors only changes the empirical covariance matrix by an overall scalar and so does not affect the PCs or their ordering. It would then appear that VeLPIC is just linear classification with classwise PCs as class vectors, which is an old idea and should be properly referenced (e.g. Das et al, "A Classwise PCA-based Recognition of Neural Data for Brain-Computer Interfaces", IEEE 2009).
>
> Thank you for pointing out this reference that we were not aware of. We have cited it with a brief description in the section where we first introduce VeLPIC.
>
>
> > The experiments that directly compare MASC and VeLPIC should be included in the main body.
>
> Thank you for this suggestion. We have moved the direct comparison results between MASC and VeLPIC for corruption levels ranging from 20% to 80% into the main text. These results for all models are now presented in Figure 3 of the main paper. The corresponding results for the cases of 0% and 100% corruption are provided in Section D.1 (Figure 12) of the Appendix.

---

> ### Author Response · Authors · 2026-02-07
> **Part-2 of author response**
>
> > Toy datasets like MNIST and Fashion-MNIST should not be prioritized in the results since the results there do not seem to generalize to the more complex datasets. At minimum I recommend moving the CIFAR10 experiments to the main body.
>
> Thank you. Following this suggestion, we have moved the results for all models to the main text. To accommodate the larger number of models, we have suitably adjusted the orientation of the figures. In addition, we have included results for a ResNet-18 model trained on CIFAR-10. These changes are reflected in Figures 1-3, and 7 of the main text and Figures 10-12, and 16  in the Appendix.
>
> > High-accuracy generalization is not being obtained in the 0%-noise setting for the harder datasets (Tiny ImageNet and CIFAR10). I recommend modifying models and training protocols as necessary to improve baseline generalization and then re-run the corruption experiments in that setting.
>
> Thank you for raising this question. There is a technical explanation, which we outline below.
> The seemingly poor performance of certain models arises from our deliberate choice not to apply regularization techniques such as dropout, except in the appendix where we have done so to study the effect of regularization. This decision was made to isolate the effects of explicit regularization on the generalization performance of the probes.
> This approach is common in work that investigates the internal mechanisms of memorized models. For example, Stephenson et al. (ICLR 2021) similarly omit regularization, and their baseline models consequently exhibit reduced performance.
>
> > Comparison with the baseline of VeLPIC or MASC applied to representations drawn from a randomly initialized network would also strengthen the argument that it is some generalization property of the representations rather than the strength of the probes.
>
> Thank you for this suggestion. We have added Section G in the appendix, along with Figure 17 , which compares the test performance of the model, MASC, and VeLPIC when applied to representations from randomly initialized and fully trained networks.
>
> In closing, we are thankful for your very careful and detailed comments. We have done our best in addressing many of the concerns raised, especially in the limited timeframe that was available. We would appreciate the opportunity to engage further with you during the discussion period.

---

> > ### Comment · Reviewer_YWzM · 2026-02-13
> >
> > My sincere thanks for taking my feedback into consideration so seriously. I am particularly appreciative of the effort that went into the battery of new experiments, which make it substantially easier to evaluate the paper's claims.
> >
> > Unfortunately, although these experiments represent a more rigorous evaluation, the narrative does not seem to hold up to this increased rigor. The ResNet experiments show MASC outperforming VeLPIC substantially in the earlier layers. The baseline experiments, where VeLPIC & MASC are applied to representations produced by randomly initialized networks, show that the accuracy barely degrades in the random network in some cases, which strongly suggests that the overall success of the probes is not due to some aspect of the training dynamics, but rather due to the model architecture, the data geometry, and the strength of the probes themselves. Again, the ResNet statistics are the most egregious examples of this, which is especially concerning since this is the "big" model tested.
> >
> > Overall, the point that study of the memorization regime is novel is well-taken and I believe this line of exploration could be fruitful. However, in my view, the empirical evidence is far too murky to support this paper's claims.

---

> > > ### Author Response · Authors · 2026-02-14
> > >
> > > We thank the reviewer for their detailed response. We address individual points below.
> > >
> > >
> > > > Unfortunately, although these experiments represent a more rigorous evaluation, the narrative does not seem to hold up to this increased rigor. The ResNet experiments show MASC outperforming VeLPIC substantially in the earlier layers.
> > >
> > >
> > > Firstly, we note that even with the ResNet-18 experiments, for higher degrees of corruption, VeLPIC on some layers does indeed beat model generalization later in training. While it is true that for early layers MASC does substantially beat VeLPIC, this does not imply that no additional latent generalization is linearly decodable. In short, we see this as an issue of difference in degree of latent generalization that is extracted linearly vs. by MASC. We also note that we have moderated our claims in light of the new experiments. For example, the last paragraph of the section detailing these results reads as follows.
> > >
> > > > These results indicate that the quadratic nature of MASC is not essential in extracting latent generalization from most models, where latent generalization is linearly decodable, although there exist models which manifest significantly better latent generalization with MASC, but not VeLPIC.
> > >
> > >
> > > > The baseline experiments, where VeLPIC & MASC are applied to representations produced by randomly initialized networks, show that the accuracy barely degrades in the random network in some cases, which strongly suggests that the overall success of the probes is not due to some aspect of the training dynamics, but rather due to the model architecture, the data geometry, and the strength of the probes themselves.
> > >
> > > These baseline experiments with MASC (but not VeLPIC) have already been largely reported in [Ketha & Ramaswamy, 2026] (see Fig 22, 24), so the degree of contribution of potentially the model architecture, data geometry and probe strength isn’t surprising here, although their relative contributions are as yet unclear. Even so, the probes on the trained model do perform markedly better, so it is clear that training does contribute to the phenomenon. Our goal with investigating the training dynamics here has been to compare latent generalization and model generalization over training, and this goal has been made amply clear from the beginning.
> > >
> > >
> > > > Overall, the point that study of the memorization regime is novel is well-taken and I believe this line of exploration could be fruitful. However, in my view, the empirical evidence is far too murky to support this paper's claims.
> > >
> > > We disagree with the characterization of the evidence being “murky”. There is definitely nuance and we believe that it is reflected in the revised claims made in conjunction with the reporting of the new experiments.

---

### Review · Reviewer_8TP4 · 2026-01-19

**Summary Of Contributions:**

The paper investigates the phenomenon of "latent generalization" within neural networks that have achieved high training accuracy on data with corrupted labels. The authors look into a few fundamental questions: the origin of this latent knowledge, the architecture of "probes" capable of extracting that knowledge, and the feasibility of "repairing" a corrupted model via weight transfer . Key findings include a demonstration that latent generalization is linearly decodable via a new probe (VeLPIC) and that replacing the pre-softmax weights of a model with these probe vectors can significantly recover test performance without additional training.

Strengths

S1: Geometrical perspective: The proof that MASC is a quadratic form serves as a good starting point to pivot toward VeLPIC, demonstrating that the latent "generalizable" signals in deep networks are not only extractable but are actually linearly decodable.

S2: Practical repair mechanism: Weight transfer offers a concrete mechanism for model recovery that requires no further training. By replacing the final layer's weights with vectors derived from latent probes, the authors show that a "memorized" model can be immediately repaired to utilize its internal representations.

S3: Dynamics analysis: The paper moves beyond static analysis by tracking the evolution of latent generalization throughout the training process. The authors provide an empirical study of the "early learning" window and the temporal point
where internal features and output heads diverge.

Weaknesses

Experimental

W1: Limited models: The authors perform experiments on a few simple architectures including small MLPs and CNN, and AlexNet. At this point in time, this cannot be considered sufficient coverage. I understand that access to advanced hardware may at times be limited, but looking into latent learning in ResNets/ DenseNets, and perhaps small ViTs should be quite possible.

W2: Limited tasks/ domains: The authors consider MNIST, CIFAR-10 and Tiny ImageNet. As with W2 - this is simply not enough to draw conclusions. Are the observed phenomena general - or a quirk of small models on small visual classification tasks? How would the dynamics behave in other visual domains? NLP? Regression tasks?

W3: Limited questions: Precisely because generalization is such an interesting topic, there is much that one would wish to see explored here. What happens as we move between layers (and here eg ResNets would be a great candidate)? What about noise other than corrupted labels? Does dropout impact the findings? Does an auxiliary semi/self-supervised mechanism? Are there forms of regularization that encourage latent generalization and make it more "retrievable"? Obviously any given paper cannot explore all the questions one might ask - but the list of questions that do get explored in this work really could be enriched.

W4: Limited benchmarks: The authors prove VeLPIC's efficacy mostly by comparing it to MASC. This is not entirely compelling - one would wish to see comparisons to alternative common approaches. There are entire multiple bodies of literature - masked autoencoders, semi-supervised learning, feature robustness, adversarial learning, and much more. MASC is not a widely used "gold standard benchmark", so just beating it cannot be considered strong evidence.

Theoretical

W5: Related literature: The paper does not engage with numerous fields and approaches that could shed much light on latent generalization. To name a few - PAC-Bayes theory could contribute to the discussion of linear vs quadratic classifiers, as well as clarify issues pertaining to sharp minima (for instance, does the proposed repair mechanism "move" the model into flatter, more robust areas in hypothesis space?). Work in gradient dynamics and implicit bias (e.g. [1], [2]) surely is relevant to the learning dynamics the paper explores. The Neural Tangent Kernel framework ([3]  seems like a good fit for the authors' kernel-based approach. Double descent ([4]) has findings that similarly challenge the "naive narrative" of classical ML. Learning with noisy labels (see [5] for a survey) cannot be ignored in this context. I want to stress here that I'm not merely complaining about the paper missing this or that favorite reference - ideally the paper would explore potential connections with some of the sub-fields above (or others) much more vigorously.

W6: Mathematical rigor: The paper is light on formal definitions and statements. To give a prominent example, the paper does not give a clear definition of latent generalization. When analyzing these phenomena, one should distinguish between the test-training performance gap (sometimes called the generalization gap), just the test error itself, and notions like the ability of the learned features to perform downstream tasks. All of those might be reasonably called generalization, and the claims sound different in each of these different contexts. Careful mathematical notation would be helpful to disambiguate.

W7: Mathematical "ambition": Viewed as a theoretical contribution, the paper does not attempt to establish any independently significant theoretical claims. The proof that MASC is quadratic is solid, but seems highly similar to existing SVM theory and QDA.

Stylistic

W8: Redundancy and imbalanced length: The abstract is substantially longer than is common, and importantly, contains much that is repeated (almost verbatim) in the introduction and discussion.

W9:MASC vs VeLPIC: Given that a key message is simply that VeLPIC is better than MASC, the amount of "real estate" given to MASC in the main body could perhaps be reduced. This include the proof MASC is quadratic (could be simply stated, with the proof relegated to the appendix) as well as the experimental setup (the MASC paper already provided this information, and is extensively cited and referred to in the current paper, and this is discussed in the appendix anyway).

W10: Motivation: the problem of learning with corrupted labels is presented "as-is", with no motivation, no discussion of real-life scenarios and the error regime that characterizes them, adversarial poisoning etc.

W11: Tone: On occasion, the paper uses language that this reviewer found too "dramatic" and rich in superlatives. Examples: MASC's "extraordinary ability" and the overstated (in my personal belief) emphasis on the surprising nature of latent generalization. It may or may not have been deeply surprising in 2017-18 (though anecdotally and personally I was deeply interested in such topics precisely then, and do not find the paper's description of the community consensus fully accurate), but since then we have had many results in that vein.

References:

[1] - Soudry et al., 2018, The Implicit Bias of Gradient Descent on Separable Data

[2] - Gunasekar et al., 2018, Characterizing Implicit Bias in Terms of Optimization Geometry

[3] - Jacot et al., 2018, Neural Tangent Kernel: Convergence and Generalization in Neural Networks

[4] - Belkin et al., 2018, Reconciling modern machine learning practice and the bias-variance trade-off

[5] - Song et al., 2020, Learning from Noisy Labels with Deep Neural Networks: A Survey

**Audience:**

No

**Audience Explanation:**

Readers who want to see strong empirical performance will need much more extensive experimental results. The more theoretically inclined members of the audience will not see a sufficiently rigorous or mathematically sophisticated investigation. Those who want to learn about the domain will not get a clear view of the state of literature. The most pragmatic readers in search of solutions to real-life problems will not see a discussion of noisy labels - or strong evidence that VeLPIC should be in their toolbox.

**Claims And Evidence:**

No

**Claims Explanation:**

As detailed above, the experimental evidence is limited in a number of ways (overly simple and outdated models, a narrow range of domains, a restricted set of investigated questions). The proposed repair method is not benchmarked against strong literature baselines. The engagement with most of the relevant up-to-date research literature that could corroborate the paper's approach is minimal. Mathematical work that would lend credence to the generality of the findings is missing.

**Requested Changes:**

I listed above what I see as key weaknesses in the paper. Not every single one of those *absolutely has* to be addressed - but the authors need to choose at least one aspect and strengthen it substantially. If it's to be an observational paper, then W1 and at least two out of the other three of W2-4 (and ideally all of them) would need to be greatly improved.

If the authors decide to emphasize concepts and theory, then a rigorous mathematical treatment (W6), as well as extensive work relating the paper to some of the existing bodies of research (W5) are "must-haves".

A more detailed coverage of related literature (W5) should be given however the paper evolves.

As for stylistic recommendations - I feel less strongly about those. Some form of motivational discussion (W10) does seem to be critical.
I feel confident that taking W8 and W9 into account will be beneficial, but would keep those as recommendations. Finally, the tone (W11) is a matter of personal taste.

---

> ### Author Response · Authors · 2026-02-07
> **Part-1 of author response**
>
> We thank the reviewer for their careful reading and constructive review.
>
>
> We address some of the comments individually below.
>
> > W1: Limited models: The authors perform experiments on a few simple architectures including small MLPs and CNN, and AlexNet. At this point in time, this cannot be considered sufficient coverage. I understand that access to advanced hardware may at times be limited, but looking into latent learning in ResNets/ DenseNets, and perhaps small ViTs should be quite possible.
>
> > W2: Limited tasks/ domains: The authors consider MNIST, CIFAR-10 and Tiny ImageNet. As with W2 - this is simply not enough to draw conclusions. Are the observed phenomena general - or a quirk of small models on small visual classification tasks? How would the dynamics behave in other visual domains? NLP? Regression tasks?
>
> Thank you. This point is well-taken. We have added experiments using a ResNet-18 model trained on CIFAR-10 and run MASC as well as VeLPIC probes on multiple layers, in response. These results appear in Figures 1,2,3, and 7 of the main text and Figures  10-12, and 16 in the Appendix. We have also compared the performance of MASC and VeLPIC with another baseline linear logistic regression probe for all models including ResNet-18 over epochs of training. Indeed, these experiments show interesting results and add important nuance to our findings.
>
> With NLP as well as regression tasks, we are not aware of a standard or straightforward way to port this definition of memorization to those settings. While there are notions called memorization especially in the NLP setting, those notions are not commensurate with the notions of memorization here and indeed what a canonical definition of memorization is, in the LLM setting is an active area of inquiry at the moment.
>
>
> > W3: Limited questions: Precisely because generalization is such an interesting topic, there is much that one would wish to see explored here. What happens as we move between layers (and here eg ResNets would be a great candidate)? What about noise other than corrupted labels? Does dropout impact the findings? Does an auxiliary semi/self-supervised mechanism? Are there forms of regularization that encourage latent generalization and make it more "retrievable"? Obviously any given paper cannot explore all the questions one might ask - but the list of questions that do get explored in this work really could be enriched.
>
> Thank you for this. We do explore the question of moving between layers with ResNet-18  and indeed as you predict, the results are quite interesting with MASC as well as VeLPIC. We have also run some preliminary experiments with Dropout. The MASC results in this case suggest no substantial difference in findings due to the introduction of Dropout. The corresponding VeLPIC experiments are running as we speak, and we will incorporate them in the next revision once they are completed. We have added the MASC results and a barebones description in Section I of the Appendix. Once the VeLPIC results are available we will incorporate them and include a detailed exposition on the results.
>
>
> > W4: Limited benchmarks: The authors prove VeLPIC's efficacy mostly by comparing it to MASC. This is not entirely compelling - one would wish to see comparisons to alternative common approaches. There are entire multiple bodies of literature - masked autoencoders, semi-supervised learning, feature robustness, adversarial learning, and much more. MASC is not a widely used "gold standard benchmark", so just beating it cannot be considered strong evidence.
>
> Thank you for this point. For alternative common approaches, we have added experiments using a linear probe trained with cross-entropy loss (logistic regression) across the layers of all networks, including ResNet-18 trained on CIFAR-10. These experiments are described in Section 7, with results shown in Figures 4 and 13. We further compare the logistic regression probe results with MASC in Figures 5 and 14, and with VeLPIC in Figures 6 and 15.
>
> As an aside, there is work [1] which suggests that MASC endows a certain degree of adversarial robustness to many models.
>
> [1] Simran Ketha, Nuthan Mummani, Niranjan Rajesh, Venkatakrishnan Ramaswamy, Adversarially-robust probes for Deep Networks, NeurIPS 2025 - Reliable ML Workshop.

---

> > ### Author Response · Authors · 2026-02-11
> >
> > > W3: Limited questions: Precisely because generalization is such an interesting topic, there is much that one would wish to see explored here. What happens as we move between layers (and here eg ResNets would be a great candidate)? What about noise other than corrupted labels? Does dropout impact the findings? Does an auxiliary semi/self-supervised mechanism? Are there forms of regularization that encourage latent generalization and make it more "retrievable"? Obviously any given paper cannot explore all the questions one might ask - but the list of questions that do get explored in this work really could be enriched.
> >
> >
> > > Thank you for this. We do explore the question of moving between layers with ResNet-18 and indeed as you predict, the results are quite interesting with MASC as well as VeLPIC. We have also run some preliminary experiments with Dropout. The MASC results in this case suggest no substantial difference in findings due to the introduction of Dropout. The corresponding VeLPIC experiments are running as we speak, and we will incorporate them in the next revision once they are completed. We have added the MASC results and a barebones description in Section I of the Appendix. Once the VeLPIC results are available we will incorporate them and include a detailed exposition on the results.
> >
> >
> > In the new revision, we have incorporated VeLPIC experiments and the exposition of the results in Section I of the Appendix.

---

> > > ### Comment · Reviewer_8TP4 · 2026-02-12
> > >
> > > Thank you for taking my concerns and recommendations to heart and for the substantial effort in providing multiple revisions.
> > >
> > > While the additional experiments and expanded bibliography have improved the paper's polish, several core issues regarding the coherence of the findings and the depth of the investigation remain. I have updated the status of the weaknesses as follows:
> > >
> > > W1 & W3 - Experimental Results and Coherence:
> > > These are mostly addressed in terms of scope, but the resolution comes at a price. With the ambiguous findings observed in the deeper layers of ResNet-18, where the quadratic MASC probe often outperforms the linear VeLPIC probe, it is no longer clear what the paper’s core claim is. The original narrative suggested that latent generalization is fundamentally linear; the new evidence contradicts this without providing a new synthesis. The exploration feels incomplete because we are left with an incoherent empirical picture rather than a clear discovery.
> > >
> > > W2 - Task and Domain Diversity:
> > > This remains unaddressed. Given the inconclusive evidence from the new ResNet experiments, the concern that these phenomena might be quirks of specific settings is amplified. I must disagree with the authors' rebuttal regarding NLP and regression; shuffling labels in sentiment classification or intent detection is a straightforward way to test the universality of these findings. Memorization and generalization are relevant across domains, and a lack of diversity here limits the paper's impact.
> > >
> > > W4 - Benchmarking:
> > > While the addition of a logistic regression probe is noted, it is a not a convincing baseline for 2026. The critical question remains: does this repair mechanism offer anything beyond what standard self-supervised learning features provide? Without a comparison to unsupervised/ semi-supervised manifold learning, we cannot determine if "latent generalization" is a unique byproduct of the training dynamics or simply a result of the model learning the data manifold despite the noise.
> > >
> > > W5 - Theoretical Engagement:
> > > The addition of citations is noted, but I believe that a deeper engagement with these frameworks, even if purely from an empirical perspective, could shed light on the inconsistent findings. For example, a discussion through the lens of neural collapse or implicit bias might explain why the linear decodability fails in certain architectures or layers.
> > >
> > > W6 - Clarity of Definitions:
> > > While I acknowledge the paper makes no formal theoretical claims, we cannot confidently interpret empirical findings without carefully distinguishing between different senses of generalization (e.g., test error, generalization gap, or feature transferability). The lack of an operational definition for "latent generalization" remains a hurdle for legibility and rigor.
> > >
> > > W7-W11:
> > > I consider these points addressed.
> > >
> > > In summary, the revision process has successfully expanded the data, but it has also revealed that the phenomenon may be less consistent than the original manuscript suggested. Without a clear empirical punchline, stronger modern baselines, or a definition that disambiguates the key quantities, I do not feel the paper is yet ready for publication.

---

> > > > ### Author Response · Authors · 2026-02-14
> > > > **Part-1 of author response**
> > > >
> > > > We thank the reviewer for their detailed response. We address individual points below.
> > > >
> > > >
> > > > > W6 - Clarity of Definitions: While I acknowledge the paper makes no formal theoretical claims, we cannot confidently interpret empirical findings without carefully distinguishing between different senses of generalization (e.g., test error, generalization gap, or feature transferability). The lack of an operational definition for "latent generalization" remains a hurdle for legibility and rigor.
> > > >
> > > > Thank you for this point. We have added a new subsection on Preliminaries, early in the paper, right after Related Work, where we provide precise operational definitions corresponding to our setting, including for latent generalization. We have also included a pointer from the Introduction section to this subsection.
> > > >
> > > >
> > > > > W1 & W3 - Experimental Results and Coherence: These are mostly addressed in terms of scope, but the resolution comes at a price. With the ambiguous findings observed in the deeper layers of ResNet-18, where the quadratic MASC probe often outperforms the linear VeLPIC probe, it is no longer clear what the paper’s core claim is. The original narrative suggested that latent generalization is fundamentally linear; the new evidence contradicts this without providing a new synthesis. The exploration feels incomplete because we are left with an incoherent empirical picture rather than a clear discovery.
> > > >
> > > >
> > > > We note that even with the new ResNet-18 experiments, for higher degrees of corruption, VeLPIC on some layers does indeed beat model generalization later in training. While it is true that for early layers MASC does substantially beat VeLPIC, this does not imply that no additional latent generalization is linearly decodable. In short, we see this as an issue of difference in degree of latent generalization that is extracted linearly vs. by MASC. We also note that we have moderated our claims in light of the new experiments. For example, the last paragraph of the section detailing these results reads as follows.
> > > >
> > > >
> > > > >These results indicate that the quadratic nature of MASC is not essential in extracting latent generalization from most models, where latent generalization is linearly decodable, although there exist models which manifest significantly better latent generalization with MASC, but not VeLPIC.
> > > >
> > > > More broadly, we want to say that empirical work in understanding the internals in Deep Learning is often messy and does not usually point to “clear discoveries”. E.g. A careful reading of the original Neural collapse paper immediately points to cases where particular models do not clearly exhibit some of the hallmarks of neural collapse. For progress, at this early stage of understanding these phenomena, we feel there is value to clearly-reported experiments, including those that may contradict a cleaner narrative, as long as the claims made are consistent with the findings, which we believe to be the case with our revision.
> > > >
> > > >
> > > > > W4 - Benchmarking: While the addition of a logistic regression probe is noted, it is a not a convincing baseline for 2026. The critical question remains: does this repair mechanism offer anything beyond what standard self-supervised learning features provide? Without a comparison to unsupervised/ semi-supervised manifold learning, we cannot determine if "latent generalization" is a unique byproduct of the training dynamics or simply a result of the model learning the data manifold despite the noise.
> > > >
> > > > The potential connections of latent generalization to phenomena such as unsupervised learning are important questions that will require substantial experimental work to investigate. We feel that it is outside the scope of the present work, whose goal was to investigate the training dynamics of latent generalization relative to model generalization, explore linear decodability of latent generalization and its direct transfer to the model.

---

> > > > > ### Author Response · Authors · 2026-02-14
> > > > > **Part-2 of author response**
> > > > >
> > > > > > W5 - Theoretical Engagement: The addition of citations is noted, but I believe that a deeper engagement with these frameworks, even if purely from an empirical perspective, could shed light on the inconsistent findings. For example, a discussion through the lens of neural collapse or implicit bias might explain why the linear decodability fails in certain architectures or layers.
> > > > >
> > > > > Thank you. In principle, the connection e.g. with neural collapse would be interesting to explore. However, we note that while we train our models typically to 100% training accuracy, neural collapse involves the terminal phase of training (TPT), where models are trained beyond this point. Therefore, studying this phenomenon would require substantial retraining of all our models. In short, we feel this is beyond the scope of the present work although we are intrigued by the possibility of interesting connections.
> > > > >
> > > > > > W2 - Task and Domain Diversity: This remains unaddressed. Given the inconclusive evidence from the new ResNet experiments, the concern that these phenomena might be quirks of specific settings is amplified. I must disagree with the authors' rebuttal regarding NLP and regression; shuffling labels in sentiment classification or intent detection is a straightforward way to test the universality of these findings. Memorization and generalization are relevant across domains, and a lack of diversity here limits the paper's impact.
> > > > >
> > > > > Thank you for this suggestion. We are considering training smaller 1-D CNN models on the sentiment analysis task, following your suggestion. We expect that this will require approximately 3-4 weeks of additional time to complete. Please let us know if doing so would be progress towards addressing this concern. We do note that we lack the compute budget at this time to train large transformer models on this task, since we would need to train multiple model instances, one for each corruption degree and cannot use pre-trained models, for our purposes.

---

> ### Author Response · Authors · 2026-02-07
> **Part-2 of author response**
>
> >  W5: Related literature: The paper does not engage with numerous fields and approaches that could shed much light on latent generalization. To name a few - PAC-Bayes theory could contribute to the discussion of linear vs quadratic classifiers, as well as clarify issues pertaining to sharp minima (for instance, does the proposed repair mechanism "move" the model into flatter, more robust areas in hypothesis space?). Work in gradient dynamics and implicit bias (e.g. [1], [2]) surely is relevant to the learning dynamics the paper explores. The Neural Tangent Kernel framework ([3] seems like a good fit for the authors' kernel-based approach. Double descent ([4]) has findings that similarly challenge the "naive narrative" of classical ML. Learning with noisy labels (see [5] for a survey) cannot be ignored in this context. I want to stress here that I'm not merely complaining about the paper missing this or that favorite reference - ideally the paper would explore potential connections with some of the sub-fields above (or others) much more vigorously.
>
> We thank the reviewer for this comment. Indeed, these are great points and it would be intriguing to investigate deeper connections to many of these other phenomena. We are of the opinion that this requires a deeper engagement with many of these topics, which we view as being beyond the scope of the present paper. That said, we have expanded the Related work section to include references to some of these topics such as with implicit bias and gradient dynamics, NTK, double descent, and learning with noisy labels.
>
> > W6: Mathematical rigor: The paper is light on formal definitions and statements. To give a prominent example, the paper does not give a clear definition of latent generalization. When analyzing these phenomena, one should distinguish between the test-training performance gap (sometimes called the generalization gap), just the test error itself, and notions like the ability of the learned features to perform downstream tasks. All of those might be reasonably called generalization, and the claims sound different in each of these different contexts. Careful mathematical notation would be helpful to disambiguate.
>
> > W7: Mathematical "ambition": Viewed as a theoretical contribution, the paper does not attempt to establish any independently significant theoretical claims. The proof that MASC is quadratic is solid, but seems highly similar to existing SVM theory and QDA.
>
> We do acknowledge that this is primarily an empirical paper and we believe we have made this fairly clear in the paper itself. We are of the opinion that latent generalization in memorization requires careful empirical analysis before getting down to the business of proffering a theoretical explanation. In short, this is a conscious choice.
>
> > W8: Redundancy and imbalanced length: The abstract is substantially longer than is common, and importantly, contains much that is repeated (almost verbatim) in the introduction and discussion.
>
> Thank you for this point. We have revised the abstract and the Introduction section to make them more concise, as suggested.
>
> > W9:MASC vs VeLPIC: Given that a key message is simply that VeLPIC is better than MASC, the amount of "real estate" given to MASC in the main body could perhaps be reduced. This include the proof MASC is quadratic (could be simply stated, with the proof relegated to the appendix) as well as the experimental setup (the MASC paper already provided this information, and is extensively cited and referred to in the current paper, and this is discussed in the appendix anyway).
>
> Thank you. While we see your point, our goal is also to keep this paper somewhat self-contained. Indeed, readers of a previous version of this manuscript, where much of this information had been relegated to the Appendix, expressed difficulty in having to refer to the Appendix to understand MASC. Likewise, another reviewer had questions about the quadratic nature of MASC that we showed, due to which we feel that in the interest of readability, there is value in keeping many of these parts of the paper in the main text.
>
> > W10: Motivation: the problem of learning with corrupted labels is presented "as-is", with no motivation, no discussion of real-life scenarios and the error regime that characterizes them, adversarial poisoning etc.
>
> Thank you for this comment. We have revised the introduction to better motivate the problem of learning with corrupted labels with reference to real-world scenarios and citations of corresponding references.

---

> ### Author Response · Authors · 2026-02-07
> **Part-3 of author response**
>
> > W11: Tone: On occasion, the paper uses language that this reviewer found too "dramatic" and rich in superlatives. Examples: MASC's "extraordinary ability" and the overstated (in my personal belief) emphasis on the surprising nature of latent generalization. It may or may not have been deeply surprising in 2017-18 (though anecdotally and personally I was deeply interested in such topics precisely then, and do not find the paper's description of the community consensus fully accurate), but since then we have had many results in that vein.
>
> Thanks. In response, we have replaced the two instances of the use of the word extraordinary in relation to MASC, to the word remarkable.
>
> In closing, we are thankful for your very careful and detailed comments. We have done our best in addressing many of the concerns raised, especially in the limited timeframe that was available. We would appreciate the opportunity to engage further with you during the discussion period.

---

### Review · Reviewer_8di8 · 2026-01-20

**Summary Of Contributions:**

The paper studies the topic of memorization -- it is focused around the data representations built by the networks which are trained on the partially corrupted datasets (part of the labels during training are shuffled). The authors confirm previous results [1] showing that despite poor (model) generalization when trained on corrupted data, the model still learns well-generalizing representations at earlier layers. The phenomenon called latent generalization is analysed using the probing technique (MASC) which is later developed by the authors and used for further experiments (VeLPIC).

The authors have three major contributions:
1. Confirm the results from previous work.
2. Develop a simplified probing mechanism VeLPIC that results in comparable or better generalization than MASC.
3. Based on VeLPIC, the authors propose a mechanism for improving the model's generalization.

**Audience:**

Yes

**Audience Explanation:**

The topic of generalization/memorization is still an unsolved problem in deep learning community and the work offers an interesting perspective on the topic which can be further improved by the authors with additonal results and increased clarity of the text.

**Broader Impact Concerns:**

No concerns.

**Claims And Evidence:**

Yes

**Claims Explanation:**

The experiments are clearly introduced and explained, the results confirm the findings of the authors. With that being said, **I would like to encourage the authors to broaden the scope of the experiments.** While I understand that the authors tried to recreate the experimental setup from the work [1], I'm afraid that the results might not hold on different architectures and datasets.

**Requested Changes:**

1. Editorial changes -- please consider rewriting abstract and contributions part (make them shorther!) You may consider this piece for an inspiration: https://www.easterbrook.ca/steve/2010/01/how-to-write-a-scientific-abstract-in-six-easy-steps/
2. Broader set of experiments -- as mentioned earlier, it would strengthen the work substantially.
3. Further experiments:
     3.1. Adding an analysis with the Linear Probes trained with Cross Entropy would be an informative experiment for the readers.
     3.2. Using VeLPIC vectors for pre-softmax layer as a warm-start to further training. I.e. Use these vectors as a trainable parameters and see if the model degrades the generalization or somehow the bias is strong enough so that the model keeps it?
4. Clarifications:
    4.1. I'm quite confused with the VeLPIC mechanism. As far as I understand, you compute $\mathcal{P}_m$ and $T_m$ on the **train dataset** according to the class-split with random labels? And then build $\mathcal{V}_m$ based on $\mathcal{P}_m$ and $T_m$ and use it to assign the label for the samples from the **test dataset** which have correct labeling. Then I'm not really sure how this should work. Imagine that the training dataset is 100% corrupted, but in a specific way where all the samples with true label 0 shift to label 1, all with label 1 shift to label 2 and so on. So in some sense, the training dataset works as a normal dataset with labels shifted by one. Do I understand correctly that given a test image with correct label 0, using the VeLPIC method we most probably would see it assigning label 1 due to the shift in the training data or am I missing something?

[1] https://arxiv.org/pdf/2501.14687

---

> ### Author Response · Authors · 2026-02-07
>
> We thank the reviewer for their careful reading and constructive review.
>
> We address each of the requested changes individually below.
>
> > 1. Editorial changes -- please consider rewriting abstract and contributions part (make them shorther!) You may consider this piece for an inspiration: https://www.easterbrook.ca/steve/2010/01/how-to-write-a-scientific-abstract-in-six-easy-steps/
>
> Thank you for the suggestion. We have revised the abstract and the contributions part in the Introduction section to make them more concise, as suggested.
>
>
> > 2. Broader set of experiments -- as mentioned earlier, it would strengthen the work substantially.
>
> Thanks. We have added experiments using a ResNet-18 model trained on CIFAR-10 and run MASC as well as VeLPIC probes on multiple layers, in response. These results appear in Figures 1,2,3, and 7 of the main text and Figures 10-12, and 16  in the Appendix. Indeed, these experiments show interesting results and add important nuance to the findings.
>
>
>
> > 3. Further experiments: 3.1. Adding an analysis with the Linear Probes trained with Cross Entropy would be an informative experiment for the readers. 3.2. Using VeLPIC vectors for pre-softmax layer as a warm-start to further training. I.e. Use these vectors as a trainable parameters and see if the model degrades the generalization or somehow the bias is strong enough so that the model keeps it?
>
> Thank you for these points. In response to 3.1 above, we have added a new section (Section 7 and Section E) that details new experiments with a linear logistic regression (LR) probe trained iteratively to minimize a crossentropy loss. The probe is trained by minimizing a crossentropy loss on snapshots of models taken during training, for individual layers of all models tested, including ResNet-18.  Furthermore, we have new plots that compare this LR probe with MASC as well as VeLPIC and we discuss the results in detail.
>
> For 3.2, we conducted weight intervention experiments during training, using VeLPIC vectors for the pre-softmax layer. Specifically, at epoch 40, we intervene by changing the pre-softmax layer weights with VeLPIC vectors and then continue standard training for the subsequent 60 epochs. The results of these experiments are presented in Section 9 of the main text and Section G of the appendix.
>
>
> > 4. Clarifications: 4.1. I'm quite confused with the VeLPIC mechanism. As far as I understand, you compute  and  on the train dataset according to the class-split with random labels? And then build  based on  and  and use it to assign the label for the samples from the test dataset which have correct labeling. Then I'm not really sure how this should work. Imagine that the training dataset is 100% corrupted, but in a specific way where all the samples with true label 0 shift to label 1, all with label 1 shift to label 2 and so on. So in some sense, the training dataset works as a normal dataset with labels shifted by one. Do I understand correctly that given a test image with correct label 0, using the VeLPIC method we most probably would see it assigning label 1 due to the shift in the training data or am I missing something?
>
> Indeed, in the case you describe, we expect that VeLPIC would most likely assign it a label 1, since all the labels are shifted by 1, and the geometry of representations is agnostic to the exact name of the label we assign.
>
> Also, we emphasize that, with VeLPIC (as well as MASC) the probes generalize nontrivially only for cases where there isn’t 100% corruption. When there is 100% corruption wherein relabeling happens uniformly at random, the model and the probes have roughly chance-level test accuracy; this has been demonstrated in Figures 10-12, 16-18. This is also implicit from [Ketha & Ramaswamy, 2026]
> In all corruption degrees that are not 100%, observe that the correct labeled examples in the training set outnumber the incorrect ones from each class, even if collectively there are more incorrectly labeled examples than correct ones. We hope this clarification is helpful.
>
>
> In closing, we are thankful for your very careful and detailed comments. We have done our best in addressing many of the concerns raised, especially in the limited timeframe that was available. We would appreciate the opportunity to engage further with you during the discussion period.

---

### Author Response · Authors · 2026-02-07
**Summary of the changes**

We thank the reviewers for their detailed and constructive comments and suggestions. In response, we have conducted multiple new experiments, and reviewed and expanded the exposition in multiple parts of the manuscript. Some of these new experiments have added additional perspective and nuance to our findings.

We list substantial changes made to the updated manuscript below.

1. Abstract: Abstract has been edited and shortened.

2. Section 1 Introduction

* Motivation with respect to e.g. real-life scenarios has been added.

* Our contributions are now listed more succinctly.

3. Section 2 Related work

*  Additional review of literature corresponding to topics such as double descent, gradient dynamics and implicit bias, NTK kernels have been added.

4. Section 4 Training dynamics of latent generalization using MASC

*  We trained ResNet-18 models on the CIFAR-10 dataset for multiple corruption degrees and performed MASC experiments on multiple layers.

* Figure 1 is reorganized to accommodate results from multiple models, some of which were previously in the Appendix, in addition to results pertaining to ResNet-18.

*  The section is rewritten to incorporate a more detailed exposition of our results, including the potential role of layerwise dimensionality, Epoch 0 accuracies and new results of ResNet-18.


5. Section 5 Non-linearity of MASC

* We have added exposition to contrast the implications of a probe being linear vs. non-linear and what a quadratic probe is.

6. Section 6 Vector Linear Probe Intermediate-layer Classifier (VeLPIC): A new linear probe

* 6.1 Training dynamics of the linear probe

     Figure 2 is reorganized to accommodate results of multiple models including ResNet-18, and more detailed exposition has been added.

* Figure 3 is now the difference in test accuracy (VeLPIC Accuracy - MASC Accuracy) during training of the network. This plot has been re-edited to accommodate more models and moved from the Appendix to main text, as suggested.

7. [New section] Section 7 Comparing MASC & VeLPIC with a baseline linear probe

* This section incorporates new experiments with a baseline logistic regression (LR) probe run on layers of all models tested over training. The probe is trained by minimizing a crossentropy loss on snapshots of models taken during training, for individual layers of all models tested, including ResNet-18.  Furthermore, we have new plots that compare this LR probe with MASC as well as VeLPIC and we discuss the results in detail.

8. Section 8 Transferring latent generalization to model generalization

* Figure 7 is reorganized  to accommodate results of multiple models including ResNet-18, to make the layout consistent with previous figures.

* We have discussed ResNet-18 results and added exposition to put them in perspective.

9. [New section] Section 9 Weight intervention during training using VeLPIC

* This new section incorporates new experiments involving a subset of the models, with weight intervention using VeLPIC midway during training, by updating the model weights at a specific epoch using VeLPIC. We then examine the dynamics of model test accuracy, as well as VeLPIC accuracy subsequently over training.

* Figure 8 and Figure 9 detailing the above results have been added.

10. Section 10 Discussion

* We have rewritten parts of the Discussion section, in light of the new experiments and inferences therein.

* We have added detailed exposition on the limitations of our work.


11. References
* Multiple new references have been added, reflecting a broader literature review in the Introduction and Related work sections. One of the references [Ketha & Ramaswamy, 2025], which was an arxiv preprint has now been published in TMLR; we have accordingly updated that reference as [Ketha & Ramaswamy, 2026].


12. Appendix

* Section A: Details of the ResNet-18 model and training details have been added.

* Section C, D, F have been updated to now have the results for 0% and 100% corruption degrees and the ResNet-18 model with 0% corruption degree.

* [New section] Section E: Results of Linear probe for  0% and 100% corruption are added.

* [New section] Section G: Results of weight intervention during training using VeLPIC for  0% and 100% corruption are added.

* [New section] Section H: Comparison between randomly initialized and trained model is added.

* [New section] Section I: Impact of Dropout as a Regularizer is added.

---

### Decision · Action_Editor_4H3r · 2026-02-20

**Recommendation:** Accept with minor revision

**Additional Comments:**

This was a bit tricky to make a decision on because of the split vote by reviewers. In addition, the two reviewers that spent the most time engaging with the authors and encouraging them to consider more complex examples were the ones who recommend not to accept. Their time and expertise should be taken seriously, and I am indeed grateful for their service.

The main rationale given by the reviewers not recommending accept is that the linear probe approach the authors develop does not convincingly outperform the MASC probe on ResNet-18. The authors point out cases where the linear probe does out-perform the MASC probe (later layers, more noise), but the larger point of there being a difference between the "simpler" networks considered by the authors and ResNet-18 I think stands. However, as the authors note, studies on phenomenology of DNN training/learning are often "messy" stories and - given TMLR's criterion for acceptance - this is not sufficient for rejection. I think as the paper stands, there is no reason to think the results are not correct (even if leading to a messy story) and they will be of interest to some TMLR reads. Therefore, I have decided to accept the paper.

All that being said, I do think there are two way the authors can strengthen their paper:

1. While the authors have tempered their claims about the universality of linear decoding in memorization networks (in-light of the ResNet-18 results), it is still phrased in a way that is misleading. In particular, in the Abstract, Introduction, and Discussion (and possibly elsewhere), the authors say that the linear probe worked "in most cases". While technically true, I think this sweeps under the rug the fact that it does not work specifically on the most complex network the authors analyze. Being more upfront about that is necessary.

2.  As a broader point, the authors have nicely titled this work "On the Dynamics & Transferability of Latent Generalization during Memorization" and motivated their work by discussing the surprising results of MASC and the need to answer the question of when/whether this generalized information can be extracted by linear probes. All of this is very general and frames the work as studying an interesting phenomenon in DNN training. However, because the initial results showed that all networks had the linear probe work as well, if not better, than MASC, the focus then moved towards linearity. This is the source of some of the concerns of the reviewers given the new results. Importantly though, the ResNet-18 results are not actually in conflict with the general goal of understanding when/whether linear probes can be used.
     Therefore, I think it would behoove the authors to discuss this difference between many of the networks they consider and ResNet-18 more earnestly (again, not sweeping it under the rug and saying "in most cases"). To me, this difference is quite interesting and new questions arise when considering it. Are more complex networks, like ResNet-18, different from earlier generation networks in the way in which they encode information (linearly vs non-linearly)? If so, what does this tell us about efforts to remove the effects of memorized labels on state-of-the-art networks?
     All of this is to say that I think the ResNet-18 results, while complicating the initial story, actually given this work more dimension and motivate interesting future work. The authors should embrace this.

These two points should be addressed in the camera ready version of the paper.

**Audience:**

Yes

**Audience Explanation:**

Again, there was a split between the reviewers. In reading through the paper myself, I felt that - while currently it is framed in a more niche way - the general topic of training dynamics and memorization are broadly of interest to TMLR readers. Opening up the framing to include these topics will make the paper of interest to the TMLR audience.

(see the additional comments section for more detail on my thoughts)

**Claims And Evidence:**

Yes

**Claims Explanation:**

There was a split between the reviewers on the claims being supported by clear evidence. Having read through the paper myself, I believe the majority of the claims are well supported, but - as some of the reviewers point out - the new results on ResNet-18 require a little more nuanced discussion. That being said, I think these are more minor edits and are easily achievable.

(I'm slightly putting the cart before the horse in these responses - see the Additional Comments section for more detail on my thoughts)

---

> ### Author Response · Authors · 2026-02-23
>
> Dear AE & Reviewers,
>
> We thank you once again for your time and service with this manuscript. We much appreciate the constructive review process.
>
> The AE’s revision suggestions are well taken and we are working on incorporating them in the camera-ready version, which we will upload in the coming days, with a brief summary of the changes made.
>
> Best,
>
> Authors

---

> ### Author Response · Authors · 2026-03-17
> **Summary of changes**
>
> We thank once again the AE and the reviewers for their time and service with this paper. In response to the AE’s suggestions, we have undertaken a minor revision and uploaded the revised version as the camera-ready version.
>
> Below, we summarize the substantive changes made in the minor revision, which are reflected in the camera-ready version of the paper:
>
> 1. Whereas previously, in the context of VeLPIC, we had framed it as a dichotomy of whether latent generalization was or was not linearly decodable by probes, we have now framed it (in the Abstract, Introduction, Section 6 and Discussion) as a question of the degree to which it may be linearly decodable.
>
> 2. We have stated prominently (in the Abstract, Introduction, Section 6 and Discussion)  that there are cases where latent generalization is linearly decodable, also noting important cases, e.g. with ResNet-18, where our linear probe underperforms MASC.
>
> 3. We have included a detailed discussion ion Section 6.1 of what factors might underlie underperformance of VeLPIC, in comparison to MASC. Ambient dimensionality appears to be such a factor, but we also discuss an exception.
>
> 4.  We have included a discussion of linear decodability or lack thereof of latent generalization, in the Discussion section.